



# Analogue earthquakes and seismic cycles: Experimental modelling across timescales

Matthias Rosenau[1], Fabio Corbi[2,3], Stephane Dominguez[3]

[1]Helmholtz Centre Potsdam – German Research Center for Geosciences, GFZ, Potsdam, D-14473, Germany
[2]University RomaTre, Rome, I- 00146, Italy
[3]University of Montpellier, UMR5243 Geosciences, Montpellier, F-34920, France

*Correspondence to: Matthias Rosenau (rosen@gfz-potsdam.de)*

**Abstract.** Since the formulation of Reid's elastic rebound theory 100 years ago laboratory mechanical models combining frictional and elastic elements have joined the forefront of the research on the dynamics of earthquakes. In the last decade,
with the advent of high resolution monitoring techniques and new rock analogue materials, laboratory earthquake experiments kept developing from simple spring-slider models to more sophisticated scaled analogue models. This evolution was accomplished by advances in seismology and geodesy which, along with a culmination of large earthquakes, have significantly increased the quality and quantity of relevant observations in nature. We here review the cornerstones of analogue earthquake model developments with a focus on scale models which are directly comparable to observational data
on short to long timescales. We revisit the basics of analogue modelling, namely scaling, materials and monitoring, as applied in earthquake modelling. An overview of applications highlights the contributions of analogue earthquake models in bridging timescales of observations including earthquake statistics, rupture dynamics, ground motion and seismic cycle deformation up to seismotectonic evolution. We finally discuss limits, challenges and links to numerical models.

## 1 Introduction

Earthquakes, the perceptible shaking of the ground, are the result of sudden release of stored elastic deformation energy in the earth due to a mechanical instability. Reid (1911) with his pioneering jelly experiments was the first to experimentally "simulate" earthquakes and formulate a theory, the elastic rebound theory, based on his laboratory and field observations following the 1908 San Francisco earthquake. Wondering "What forces could have produced such distortion and displacements in the rock mass of the region" (Wood, 1912) Reid postulated that only elastic forces can do so. He
hypothesized that the release of elastic deformation stored due to slow accumulation of strain in the earth occurred by fracturing "along an old fault line" (Reid, 1911) which caused vibration and rebound of the elastically strained rock mass in the vicinity of the fracture.

With the rise of the plate tectonic theory in the 1960's accompanied by a flouring of seismology and experimental rock mechanics, stick-slip instabilities along pre-existing discontinuities, i.e. seismogenic tectonic faulting has become the most
prominent earthquake mechanism (Brace and Byerlee, 1966a; Byerlee, 1970). Today we are aware that the largest



earthquakes, as measured by the seismic moment release energy, are exclusively a result of tectonic faulting in the brittle parts of the Earth's crust concentrated along plate boundaries. Smaller events might occur as a result of e.g. magmatic diking, hydraulic fracturing, landslides or nuclear tests. We here focus on (modelling of) tectonic earthquakes.

The study of earthquakes faces, however, several limiting factors related to the difficulty to access the deep source of
earthquake and to integrate the characteristic time scales of deformation processes that extend from seconds to thousands of years. As a consequence, seismic hazard mitigation is inevitably based on incomplete geological datasets and poorly constrained physical parameters, which affect notably their relevance.

New technological advances in seismology and geodesy have significantly improved our knowledge of the dynamics of deformation processes associated with earthquakes and, by substituting space for time, of all archetypical phases of the
seismic cycle including the coseismic, postseismic and interseismic phase (e.g. Klotz et al., 2001; Wang et al, 2012). However, if recent earthquakes are well documented, this is not the case for older events for which historical records are often too short and incomplete to be usable with the required precision. Beyond the last fifty years, only sparse seismological records and kinematic measurements are available, which is not sufficient to constrain seismic cycle dynamics extending over a time span of 100 to 1000 years. In particular, geodetic data (e.g., GPS) that only cover a short time span of the
interseismic period (< 30 years) are typically extrapolated over longer periods using steady-rate assumptions. However, numerous geophysical observations highlight the complexity of the interseismic period (e.g., slow earthquakes, periods of seismic quiescence or crisis) suggesting that the steady-state assumption may be an oversimplification. This raises the issue of a time constant in the earthquake cycle which is far larger than the duration of most scientific observations.

The scientific exploitation of available earthquake geophysical data is mainly based on analytical and numerical modelling
methods that allow obtaining complementary information about deformation processes, physical and mechanical properties of faults and boundary conditions by means of data inversion (e.g. slip at depth, contemporary stress accumulation, stress transfer). Although such approaches allow reproducing and analysing observed surface velocity fields, they face some limitations. On one hand, some important parameters, such as frictional conditions along the fault plane or stress history induced by past seismic cycles are not understood enough to be accurately considered. On the other hand, modelling all
seismic cycle phases using a single approach with different processes acting at different time and space scales is still a difficult goal to achieve numerically.

In summary, the low level of physical understanding of earthquakes prevents us from developing predictive numerical models. Moreover, only a short period of highly resolved natural occurrences is available while resolution decreases quickly as we go back in time. Respecting these limitations and gaps in understanding necessitates experimental approaches to
earthquake dynamics. While those experiments are necessarily reduced in their complexity and naturally limited in their direct applicability to larger scales, they provide long enough time series with a superb resolution. Moreover, analogue models used in experiments are physically self-consistent dynamic systems able to mimic phenomena such as emergence and self-organized criticality. Therefore they are an explorative simulation tool to understand the link between short-term and long-term deformation processes bridging the time scales from earthquake nucleation to tectonic evolution.



We here present an overview of the history and state-of-the art in analogue earthquake modelling. We distinguish existing experimental approaches into three categories of increasing complexity and increasing similarity with the natural prototype. The latest development, namely analogue scale models, has a central position here, as scale models have shown great potential for future developments in parallel to numerical simulations of the earthquake process. We elaborate on the scaling,

monitoring and material characterization and show a brief overview of applications of the various models. Where adequate we make links to numerical models and highlight perspectives and challenges.

## 2 Experimental approaches overview

In parallel to the development of analytical and numerical approaches, numerous experimental or analogue models have been developed to investigate the physics of earthquakes, seismic cycle dynamics and seismotectonic evolution. We here

categorize analogue earthquake models into three groups with decreasing level of abstraction (Fig. 1, Table 1): (1) "Spring-slider models" in which elastic and frictional elements are physically discrete components of the setup (Sect. 2.1); (2) "Fault block models" in which two elastic blocks, with similar or different elastic properties, are in frictional contacts (Sect. 2.2); (3) "Scale models" in which a distinct tectonic setting is realistically simulated at small scale and with boundary conditions mimicking as closely as possible the natural prototype (Sect. 2.3).

## 2.1 Spring-slider models

Following Reid's initial idea of earthquakes reflecting the release of elastic deformation energy stored in the earth by a mechanical instability rather simple spring slider models (Fig. 1a) have been employed both in seismology and experimental rock mechanics. The spring slider system is mathematically modelled using a single differential equation which describes the slider motion as a function of the relevant forces acting on it:

$$m\,a(t) + A\,v(t) + f(v,x,t,\dots) + k(x - vt) = 0, \tag{1}$$

where $m$ is slider mass, x, a and v are location, acceleration and velocity, t is time, A is damping factor, $f$ is friction force, $k$ is the spring constant (force over length). Numerous solutions, developments and applications exist in the literature (e.g. Burridge and Knopoff, 1967; Cao and Aki, 1984; Schmittbuhl et al., 1996; Carlson and Langer, 1998; Gu and Wong, 1991; Mori and Hikaru, 2006, 2008; Wang, 2012; Erickson et al., 2011; Abe et al., 2013; Aragon and Jagla, 2013) which will not

be discussed here in detail. Burridge and Knopoff (1967) developed the simple spring-slider model into one and two dimensional models, both experimentally and numerically. They established the concept of a chain of coupled spring-slider system being able to mimic realistically earthquake occurrence and mechanisms which revolutionized statistical seismology. Probably because of the simplicity of the math behind spring-slider models, few laboratory models strictly being spring-slider systems have been used to scientifically approach earthquake dynamics later on. One of the few is that of King (1991;

1994) who employed a circular chain of spring-sliders to study earthquake "predictability". Heslot et al. (1994) performed spring-slider experiments to illuminate the dependence of frictional stability on the fundamental parameters spring stiffness,





loading velocity and slider mass. More recent studies have been realized by Varamashvili et al. (2008) to study the effect of external forcing and by Popov et al. (2012) to study the onset of frictional instability.

Brace and Byerlee (1966) were the first amongst a number of other rock mechanics experimentalists of the late 1960's (see references in Brace, 1972) reproducing stick-slip instabilities by biaxial compression of cylindrical rock samples. The

loading machines used were usually designed as stiff as possible (in any case stiffer than the rock sample) but showed to be compliant enough to store and release elastic deformation energy. Therefore such tests can be considered as a type of spring-slider experiment. Both intact and pre-cut samples were used to generate stress-drops associated with stick-slip instability in the order of kilobars. In a large number of experiments, the role of mineralogy, porosity, pressure, water, temperature, gouge thickness and stiffness of the loading system on stick-slip has been established firstly (as summarized in Brace, 1972).

Since these pioneering experiments sticks-slip as an analogue of earthquakes has been studied using axial testing machines but also direct and rotary or ring shear tester (see respective references in Table 1). It is not the scope of this paper to review the large body of rock mechanics experimental work done on stick-slip deformation in the last half century. Here we focus on those approaches using analogue rock materials instead of rock samples. Knuth and Marone (2007) for example used rods of different materials in a double-direct shear device in different configurations to study the mechanisms sliding, rolling and

dilation as well as stick-slip. A large body of works exists on granular shear experiments using glass beads and other synthetic fault gouges in shear and axial compression apparatuses, some with a focus on stick-slip (e.g. Anthony and Marone, 2005; Mair et al., 2002, Alshibli et al. 2006, Schulze, 2003; Scuderi et al., 2015). These were accompanied by numerical models mainly using the discrete element method (e.g. Abe and Mair, 2009; Abe et al., 2006; Ferdowsi et al., 2013, 2014, 2015).

A special type of spring-slider setups may be called "deformable slider-spring" in which the slider is not rigid but plastic. While elasticity is still controlled by a separate spring, the frictional element can be replaced by different plastic rheologies, e.g. Bingham fluid (Reber et al., 2015).

The main limitation of the classical spring-slider setup stems from the general rigidity of the slider. A rigid slider distributes shear stress evenly across the frictional interface. Therefore both loading and release are unrealistically homogenous. Slip

distributions of earthquakes in nature usually show complexity with areas of high and low energy release (asperities and barriers, respectively, e.g. Aki, 1984). Such heterogeneity might be a stationary feature through subsequent seismic cycles or transient and related to variably frictional properties and/or slip history of the fault. In any case it also reflects heterogeneous loading as well as heterogeneous release.

## 2.2 Fault block models

Fault block models have been developed to dilute the strong assumption of uniform loading and release inherent spring-slider models. They allowed investigating different aspects of earthquake dynamics at the scale of an analogue fault plane. With respect to spring-slider models, where elasticity is provided to the system using a spring, in fault block models the elastic strain is stored within the sample volume (Fig. 1b) mimicking the behaviour of the fault bounded crustal blocks. The



first experiments by Reid (1911) were jelly block experiments used to demonstrate the distortion and displacement phenomena seen during the 1908 San Francisco earthquake. In contrast to spring-sliders this setup also allows small (partial) and large (complete) scale failures to occur bearing the potential to generate more realistic frequency size distributions.

Fault block models share the same common characteristic: they are composed by two blocks in relative motion or, equivalently, a slip surface embedded in an elastic solid. The slow motion imposed to the system mimics the tectonic loading. Two loading configurations can be differentiated: Shear (including direct shear, ring shear and Couette type) and (biaxial) compression. Earthquakes may nucleate spontaneously and repeatedly or by external forcing, i.e. impact (e.g. Xia et al., 2004).

The two blocks and the interface between them are the analogue of rock volume and an embedded fault of finite dimensions, respectively. Edge effects, artificial reflections and free surface effects are usually unavoidable in fault block models (e.g. Scholz et al., 1972). The two blocks may be of the same material (e.g., foam rubber: Brune 1973; rock: Lockner et al., 1991, Lei et al (2000), Zang et al. (2000), Thompson et al. (2005; 2006, 2009)) or materials with different compliances (e.g., gel sliding on glass: Baumberger et al., 2003; rubber on rough substrate: Hamilton and McCloskey, 1997, 1998; Schallamach, 1971). Different materials have been adopted depending on the desired rheological response of the system; ranging from purely elastic (e.g., rubber; Schallamach 1971, plexiglass, Rosakis et al., 2007 and references therein) to viscoelastic (e.g., Polyvinilalcool PVA; Namiki et al., 2014). The use of softer materials allows to slow down the rupture process and make it more accessible by means of monitoring.

### 2.3 Scale models

With the advent of high resolution strain monitoring (Adam et al., 2005) it became possible to measure small scale deformation increments (i.e. scaling to decimetres-meters in nature) corresponding to single earthquake displacements. This unlocked the possibility to realize analogue scale models of earthquakes (Fig. 1c). Modern scale models feature realistic non-linear frictional properties of materials mimicking the coseismic dynamic weakening that in nature happens for various reasons (e.g. frictional melting, thermal pressurization, chemical effects). A second feature is a properly scaled elasticity of the model. Classical analogue models of tectonic processes use sand or other rigid particles to study long-term fault kinematics in the brittle regime but their elastic moduli appear to be too high (GPa, e.g. Klinkmüller et al., 2016) to be used to simulate elastic deformation realistically. To model earthquakes and seismic cycles scaling rules impose to decrease the elastic moduli of the model by the several orders of magnitude. This can be achieved by adding elastic particles (e.g. rubber) or by using compliant solids (e.g. gelatine, foam). In contrast to fault block models scale models allow for a realistic depth-dependent pressurization of the faults (i.e. lithostatic pressure). Loading conditions mimic tectonic forcing is realized by applying e.g. pure or simple shear boundary conditions.

In the context of earthquakes and seismic cycles, scale models are used to study seismogenic fault behaviour over many orders of magnitude in timescales. They allow to simulate multiple seismic cycles in three-dimensional models with fully dynamic ruptures including the interaction between seismic and aseismic fault areas, off-fault deformation in the brittle



upper crust as well viscoelastic relaxation in the lower crust and mantle. Therefore, despite their simplicity, they are the preferred scientific tool to overcome the limitations in natural observations by means of sufficiently long and well enough resolved time series of deformation.

From an observational point of view, the main challenges of analysing analogue earthquake models are the elastic nature of deformation as well as the very small displacements on various time scales, especially with regards to the interseismic stage. Because earthquake cycles are dominated by elastic deformation, the deformation fields are characterized by velocity reversals and alternations between slow strain accumulation (loading) and fast release (earthquake). Due to the high variability in strain rates only a quasi-continuous or highly resolved incremental monitoring allows accurate quantification. Moreover, in scale models, incremental displacements of interest are necessarily very small as decimetres to meters in nature scale down to micrometres in labscale models. Finally, deformation increments occur on a wide range of time scales (seconds to thousands of years in nature, milliseconds to minutes in labscale models) resulting in strongly variable velocities which vary by more than 12 orders of magnitude in nature. Thanks to adaptive time scaling (see Sect. 3) the latter can be reduced to about three orders of magnitude in analogue models.

There have been few scale model approaches in the recent past (Table 1, Figure 2). Viscoelastic scale models (e.g. Corbi et al., 2013) were used to study rupture dynamics and elastic coseismic deformation of the forearc wedge. In these models, a gelatine wedge (analogue of the overriding plate) is underthrusted by a 10° dipping, planar and rigid aluminium plate (analogue of the subducting slab, Fig. 2a). Plate convergence is imposed kinematically. An UV light sheet is used to illuminate fluorescent markers along a central section of the model. The experiments are monitored in sideview using digital image correlation techniques. The subducting plate embeds a "seismogenic" velocity weakening zone limited by velocity strengthening areas at its updip- and downdip limits. Frictional instabilities (the analogue earthquakes) nucleate and propagate along the gelatine-aluminium interface. Viscoelastic models feature the following characteristics that make them useful to investigate different aspects of the seismic cycle with respect to elastoplastic models: a) the earthquake process lasts for few seconds, allowing to capture the nucleation process, the propagation and arrest of the rupture; and b) the rheology of the material can be tuned depending on the desired experimental requirements. The analogue models of Corbi et al. (2013) have been tested against numerical models using the finite element technique (van Dinther et al., (2013b).

A similar setup has been used by Rosenau et al. (2009) but with a granular, elastoplastic wedge on top of a rather stiff conveyer plate or belt (Fig. 2b). This setup allows simulation of seismotectonic evolution of subduction forearcs and thus bridging the short to long term processes of earthquakes and tectonic evolution, respectively. Because of the opacity of the models strain monitoring was restricted to side views through glass walls or stereoscopic top views of the surface deformation in 2D and 3D experiments, respectively. Seismogenic zones at the base of the wedge were defined by using velocity weakening materials (e.g. rice). Compared to viscoelastic wedges of Corbi et al. (2013) elastoplastic wedges were stiffer. Consequently the rupture velocity was higher while the deformations were smaller (Fig. 3a). From a scaling point of view this is more appropriate; however, it poses limitations on the observability of analogue earthquakes, i.e. a lower bound



of about M8 events. The analogue models of Rosenau et al. (2009, 2010) have been cross-validated using finite element models (Pipping et al., 2016; Rosenau et al., 2016).

Caniven et al. (2015) have developed an experimental approach allowing to simulate strike-slip fault earthquakes in a brittle-ductile crust and analyse hundreds of successive seismic cycles (Fig. 2c, 3b). One of the remarkable points is the use of a

multi-layered visco-elasto-plastic analogue material (silicone, PU foam, granular media) to take into account, at a first order, the crust rheology and its mechanical behaviour. This is a crucial point to allow for simulating brittle/ductile couplings, post-seismic deformation phase or also far field stress transfers. Following the approach of Caniven et al. (2015), a setup for 3D subduction megathrust experiments is being developed (Dominguez et al., 2015, Fig. 2d).

## 3 Scaling and similitude

Scaling labscale observation to the nature is a central issue in analogue earthquake modelling, especially if scale models are considered. This Section elaborates on the scaling of such models. To be representative of a natural system a small-scale model should share geometric, kinematic and dynamic similarity with its prototype (Hubbert 1937). This condition is termed similitude and requires that all lengths, time and forces scale down from the prototype in a consistent way dictated by scaling laws. The latter are derived either from an analytic approach (e.g. Weijermars et al., 1993) or from dimensional analysis and

the formulation of dimensionless numbers (Buckingham 1914, Table 2).

The large range of velocities to be captured in an analogue earthquake model pose both a practical challenge to conduct experiments in a reasonable time frame and a challenge in observing (monitoring): Either the total run time of an experiment simulating thousands to Millions of years in nature is reasonable (scale of hours) then the episodically recurring earthquakes are captured only with very high speed monitoring techniques (MHz). Alternatively the earthquakes are slowed down to a

reasonable speed (to be captured by 1-100 Hz monitoring) then a model run would take weeks.

However, from a scaling point of view the range in velocities in the model can be significantly reduced by using adaptive timescaling (corresponding to adaptive time stepping in numerical modelling). Rosenau et al. (2009) introduced such a "dyadic" timescale which recognizes two dynamically distinct regimes of the seismic cycle: the quasi-static interseismic regime, where inertial effects are negligible due to the slow deformation rates and the dynamic coseismic regime which is

controlled by inertial effects. Consequently two different temporal scales are applied. This way, the earthquake rupture can be virtually slowed down while the loading phase is sped up while keeping dynamic similarity in both stages. In a typical application 0.1 second may correspond to a quarter century of interseismic loading and about a minute of rupture time.

The transition from quasi-static to dynamic can be defined kinematically. Based on theoretical considerations using a spring-slider system Roy and Marone (1996) suggest the transition to occur at a critical velocity which is a function of extrinsic and

intrinsic frictional properties and mass. Latour et al. (2013) in contrast, based on empirical results and theory, equal the transition from quasi-static to dynamic with the transition from exponential growth to power law growth of the rupture length. They suggest that elastic and frictional properties control the transition. In practice a velocity threshold is defined



which also depends on the temporal resolution to differentiate between the quasi-static and dynamic regime. With better spatial and temporal resolution in future, this issue will certainly be re-visited in detail.

In the following the scaling for the short and long term, or inter- and coseismic phases are elaborated in Sect. 3.1 and 3.2, respectively. Dimensionless numbers and scaling relations are summarized in Tables 2 and 3.

## 3.1 Quasi-static regime: Scaling the interseismic phase

In the quasi-static regime of the interseismic phase scaling is identical to common scaling of long-term processes to the lab. For long-term tectonic studies two dimensionless numbers are of interest: First, the Smoluchowski Number

$$Sm = \frac{\rho g}{\sigma/l} = \frac{\text{gravitational force}}{\text{pressure force}},\tag{2}$$

where $\rho$ is density (kg/m³), $g$ gravitational acceleration (m/s²), $\sigma$ stress (Pa), $l$ a characteristic length and the Ramberg
Number

$$Ra = \frac{\rho g}{\eta v/l^2} = \frac{\text{gravitational force}}{\text{viscous force}},\tag{3}$$

with $\eta$ is viscosity (Pa s) and $v$ a characteristic velocity (Ramberg, 1967´; Brun, 2002; Pollard and Fletcher 2005). Note that the Ramberg Number has been derived from the Stokes Number Table 2) which characterizes more generally slow (small Reynolds Number $Re \ll 1$, Table 2) flow typical for tectonic considerations.

To achieve similitude these numbers have to be the same in the model as in the prototype. For a given length scale (usually suitably chosen for handling the model in a lab), $Sm$ and $Ra$ dictate the stress, length and time scaling in the brittle and viscous regimes, respectively.

By substituting cohesion for stress in Eq. (2) we find for a brittle sandbox model under normal gravity that the cohesion should scale according to the density and length scale following the scaling law:

$$C* = \rho* \, g* \, L*\tag{4}$$

where the asterisks denote the model/prototype ratios also known as the scaling factors (i.e. $C* = C_{model}/C_{prototype}$). Inserting typical numbers for the scaling factors for density and cohesion of a brittle analogue rock material like sand ($C* \sim 10^{-6}$, $\rho* \sim 0.6$; e.g. Klinkmüller et al., 2016) into Eq. (3) yields typical length scales $L*$ of $10^{-5}$ to $10^{-6}$, i.e. 1 cm in the model equals 1-10 km in nature. Regarding analogue earthquake models the typical scaling of length, here peak slip, is shown in Fig. 4a
using data from Rosenau et al. (2010). Accordingly, coseismic slip of few tens of micron scale to several meters in nature.

As cohesion and elastic moduli share the same dimension (Pa), elastic moduli should scale with the same scaling factor (e.g. $E* = E_{model}/E_{prototype} = C*$). For typical values of tens of GPa elastic moduli in nature analogue rock material should be typically rather soft, i.e. tens of kPa, like e.g. foam rubber (see Sect. 4.1.2).

For a viscous model under normal gravity we find that the stress should scale according to the viscosity scale and strain rate
scale following the scaling law:

$$\sigma* = \eta* \, (d\varepsilon/dt)*\tag{5}$$





Note that the time scale for long-term viscous models can be derived from Eq. (5). Inserting typical numbers for the scaling factors for viscosity of an analogue material like silicone (Sect. 4.3.2: $\eta^*$ ~10-15-10-16) and keeping the stress scaling factor the same as in the brittle model ($\sigma^*$ = 10-5-10-6) yields typical time scales $T^*$ of 10-9-10-10, i.e. 1 second in the model ranges to about 30-300 years in nature. Regarding analogue earthquake models the typical scaling of recurrence time is
shown in Fig. 4a. Accordingly, recurrence times of tens of seconds scale to hundreds to thousands of years in nature (Rosenau et al.; 2010; Caniven et al., 2015).

### 3.2 Dynamic regime: Scaling the coseismic phase

For the coseismic stage inertia controls the dynamics, so that the Froude Number can be used to reach dynamic similarity and find an appropriate short-term time-scaling (Rosenau et al., 2009):

$$\text{Fr} = v/\sqrt{gl} = \frac{\text{inertial force}}{\text{gravitational force}}. \tag{6}$$

Froude scaling sets the important constraint that the timescale of the model should be the square root of the length scale:

$$T* = \sqrt{L*}. \tag{7}$$

As a consequence, all accelerations are the same in the model as in the prototype. Regarding the stress scale in the dynamic regime the Cauchy number can be used (Rosenau et al., 2009):

$$\text{Ca} = \frac{\rho v^2}{K} = \frac{\text{inertial force}}{\text{elasticity}}. \tag{8}$$

where $K$ is the bulk modulus or an equivalent elastic moduli. Importantly, the stress scale derived by Cauchy scaling in the dynamic regime is coherent with stress scale derived by Smoluchowski scaling the in the quasi-static regime.

A theoretical conflict arises however when viscous forces in the dynamic regime are considered: These should be typically scaled using the Reynolds number. For various reasons the similarity requirements posed by the Froude and Reynolds
numbers can typically not be satisfied simultaneously. In our application, both the Reynolds and Froude number can be preserved simultaneously only if we use a viscous material which hardens dramatically (by three orders of magnitude or so) coseismically or assume that the mantle weakens accordingly. In practise, assuming that viscous flow does play no role in the coseismic stage which is dominated by elastic deformation non-conservation of the Reynolds Number through in the coseismic phase seems acceptable.

Apart from the dimensionless numbers introduced above all model parameters without a dimension should be preserved. Here belongs e.g. the Poisson ratio, the friction coefficient, the friction rate and state parameters and the stress exponent in the viscous regime. An exception to this general scale independence of dimensionless parameters is the moment magnitude $M_w$ which is related to the seismic moment (unit Nm) but defined dimensionless:

$$Mw = \frac{2}{3}\log(M0) - 10.7. \tag{9}$$

The scaling factor for seismic moment $M_0^*$ can be derived straight forward from the scaling rules for length and stress as it is defined as the product of rigidity of the earth, mean coseismic slip and rupture area. Accordingly, typical analogue





earthquake events with less than 1 Nm scale up by more than twenty orders of magnitude up to $10^{22}$ Nm in strike-slip experiments (Caniven et al., 2015) and $>10^{23}$ Nm in subduction megathrust experiments (Rosenau et al., 2009; Fig. 4a).

As an example to illustrate this massive scaling, the energy of an analogue earthquake is about the energy needed to illuminate a 10 W electric light bulb for the duration of an analogue event which is similar to an eye blink (ca. 100 ms). In contrast the energy of an $M_w$=9 earthquake released over several minutes equals to 100 years of the world's energy consumption (as of the year 2000).

As Eq. (9) is not a product of dimensional parameters but a sum of two terms one of which includes a logarithm of a dimensional parameter, the standard method of applying scaling rules for the involved dimensions fails. Consequently, a scaling factor s.s. for moment magnitude does not exist. We can however scale up analogue earthquake moment magnitude non-linearly by applying the scale factor of seismic moment $M_0$* and defining the following scaling rule:

$$Mwprototype = Mwmodel + \frac{2}{3}\log(M0*). \tag{10}$$

Typically this results in magnitudes of analogue earthquakes in the range of -6 to -7 which correspond to earthquakes of $M_w$ = 8-9 (Fig. 4b).

## 4 Analogue rock rheology

This Section reviews the history of analogue rock materials used in laboratory modelling of earthquakes and seismic cycles. They fall into three groups according to the dominant rheology: elastic, (frictional-)plastic and viscoelastic. They are accordingly described in Sect. 4.1, 4.2 and 4.3. The model materials fall additionally into soft and stiff materials in terms of everyday live experience. Stiff materials (e.g. Plexiglas, wood) are used mainly in the fault block model category to model earthquakes under near-natural pressures (MPa) in rock mechanics deformation rigs while for scale models the materials are generally soft or weak. This is because scaling laws dictate the models to deform in response to forces many orders of magnitude smaller than in nature. In particular, forces driving tectonic faults are in the order of TN per meter fault length while in the lab only few N should be enough to deform the material. The latter is typically realized by using bulk solids (e.g. loose sand), foam rubber or silly putty. Most of the materials (e.g. gelatine) inherit two or even all three rheologies under different conditions. Key material properties of the most commonly used rock analogues are summarized in Table 4.

### 4.1 Elasticity

### 4.1.1 Hooke's Law, crack growth and elastic dislocations

Rocks at low temperature, pressures and strains behave elastically, that is, they deform when a force is applied and return to their original shape when the force is released. In linear elastic solids, as in springs, elastic strain ε is generally linearly related to the applied stress σ in the same direction ("Hookes's Law", Fig. 5):

$$E = \frac{\Delta\sigma 1}{\Delta\epsilon 1}. \tag{12}$$





Moreover, elastic materials are conveniently treated as isotropic that is the mechanical property is independent of the direction and there are no preferred directions. Five elastic parameters exist, three of which have the dimension of stress (Young modulus $E$, Bulk modulus $K$, Shear modulus $G$) and two are dimensionless (Lamé constant $\lambda$ and Poisson ratio $\nu$). This set of parameters is dependent such that two needs to be defined to derive the others. The moduli E, K and G describe

the strain-stress relationship under different loading conditions, respectively, axial, volumetric and shear. The elastic moduli of rocks are in the order of 10 to 100 GPa (e.g. Turcotte and Schubert, 2002, Appendix 2, section F). The Poisson ratio, which describes the relation between axial and transverse strain, varies between 0.1 and 0.4 for rocks (ibid.).

Crack growth is intrinsically related to the elastic moduli. In the simplest case of a "penny shaped" crack in which the slip $D$ is driven by the uniform stress drop $\Delta\sigma$ along the fault slip ($x$-coordinate) is given by Eq. (13):

$$S(x) = \frac{24}{7\pi}\frac{\Delta\sigma}{G}\sqrt{r^2 - x^2},\tag{13}$$

in which $r$ is radius of the crack (Eshelby, 1957, Scholz, 2002). For a uniform initial shear stress slip on a penny shaped crack shows the typical elliptical shape in which slip at a location is inversely proportional to the shear modulus (Fig. 6a). The rupture velocity of cracks is also typically considered to be limited by the shear wave velocity $V_s$ which is given by Eq. (14):

$$Vs = \sqrt{G/\rho}.\tag{14}$$

Therefore the speed of the analogue earthquake process, which generally poses resolution limits on observations, is critically controlled by the stiffness of the rock analogue material used.

Elastic deformation in the solid surrounding the crack is usually described by elastic dislocation theory (e.g. Pollard and Segall, 1987). Since the earth's surface is considered mechanically a free surface, i.e. no shear and normal stresses are

transmitted across it (to the atmosphere), so-called "half space" models are applied. Additionally, because the characteristic lengths of static and dynamic deformations (e.g. stress shadows, seismic waves) associated with earthquakes are usually regional scale both small scale topography and large scale earth's curvature are often neglected and the surface modelled as a plane. Analytical solutions to the problem of shear crack induced surface and internal deformations in homogenous elastic half space are given by Okada (1985, 1992) and applied numerously (e.g. King et al., 1994; Toda and Stein, 2002; Lin and

Stein, 2004). A convenient Matlab© based tool ("Coulomb") based on these solutions has been developed by the USGS (Toda et al., 2011, Coulomb 3.3 Graphic-rich deformation and stress-change software for earthquake, tectonic, and volcano research and teaching—user guide: U.S. Geological Survey Open-File Report 2011–1060, 63 p., available at http://pubs.usgs.gov/of/2011/1060/).

Both crack growth as well as elastic dislocation predictions are superb benchmarks for scale models. Simplified versions of

the surface deformation induced by vertical strike-slip dislocations in elastic half space exist both for the case of interseismic and coseismic stages of the seismic cycle: Interseismic surface velocities $V_x$ parallel to a strike-slip fault as a function of fault-perpendicular distance y can be calculated using the following EQ. (15):

$$Vx(y) = \frac{V0}{\pi}\arctan(y/DL),\tag{15}$$





where $V_0$ is the far-field loading velocity, $D_L$ is the locking depth (Savage and Burford, 1973). Similarly Reid's coseismic rebound of the sidewalls of strike-fault can be calculated using Eq. (16):

$$Ux(y) = \frac{S}{\pi} \arctan(DS/y), \qquad\qquad (16)$$

where $U_x$ is the fault-parallel surface displacement and $D_S$ the depth to which slip extends (Chinnery, 1961). Fig. 6b and 6c
show the predictions for inter- and coseismic surface deformation across a strike-slip fault, respectively.

### 4.1.2 Elastic rock analogue materials

While in spring-slider setups, mechanical springs or the effective stiffness of the testing machine controls elasticity of the system, a variety of analogue rock materials have been used which can be classified linear, isotropic, elastic solids up to few percent of strain similar to the earth at relevant scales. Elastic rock analogues classify into relatively stiff (e.g. Plexiglas) and
soft (gelatine, rubber, foam). Stiff materials are used exclusively in spring-slider and block models while soft materials find also appliance in scale models for which scaling rules dictate their Young moduli in the order of 1-1000 kPa.

Examples of stiff elastic materials are "Homalite-100" and Polycarbonate as used in the studies shown by Rosakis et al. (2007). Those show enhanced photoelastic compared to other transparent stiff materials like PMMA (polymethyl methacrylate). Accordingly, they are characterized by a Young Modulus in the order of few GPa, a Poisson ratio of ca. 0.35-
0.4 and a shear wave speed of ca. 1000 km/s (Rosakis et al., 2007).

Examples of soft elastic materials include gelatine. Gelatine is the common name for animal and plant viscoelastic biopolymers that have been adopted for analogue modelling (see Di Giuseppe et al., 2009, Kavanangh, 2013, and van Otterloo and Cruden, 2016, for a complete rheological characterization of a wide range of gelatines). The shear modulus of gelatine is controlled by its concentration. Small concentration (< 10 %) gelatine has a Young modulus of few kPa while
high concentration (>10 %, "ballistic") gelatine has a Young modulus of few hundreds of kPa (Fig. 5). Because it consists mainly of water, gelatine has a density of around 1000 kg/m³ and a Poisson ratio of 0.45-0.5. The latter poses some limitations to the similarity of stress orientations in analogue models using gelatine which are, however, not yet well constraint. Gelatine can more completely be described as viscoelastic and is therefore described with regards to the respective properties in Sect. 4.3.2.
An often used soft elastic solid are foam "rubbers", i.e. foam polymers (e.. PE Polyurethan, PVC Polyvinyl chloride, Styrofoam, Polyimide, etc.) which come in a variety of densities and elasticities. Those used in analogue modelling are usually light weight (e.g. density ca. 20-40 kg/m³ used by Anooshepoor and Brune, 1999 and Caniven et al., 2015) and have a Young modulus ranging from 10-100 kPa. Poisson ratios range between 0.1 and 0.3 (Brune 1973; Caniven et al., 2015) and shear wave velocity are in the order of 10-100 m/s. Attenuation properties of foam correspond to a low Q of 10 similar to
shallow crustal layers (e.g. Brune and Anooshepoor, 1998). The damping cannot be controlled. This limits the use of foam in simulating seismic wave propagation to the near field. On the other hand it prohibits reflection from the model boundaries. The low density of foam poses additional limitations on its use in scale models as a realistic pressure gradient cannot be



established. The presence of material nonlinearities (both kinematic and constitutive) as well as the high friction coefficient of foam rubber-foam rubber interfaces raise some concerns about the scalability of theses models (Rosakis, 2007). Part of these limitations, i.e. the high fiction coefficient, has been overcome by Caniven et al. (2015) by coating the foam rubber with epoxy resin.

Rubber (e.g. EPDM, etyhlene-propylen-diene-monome) has been used both as a solid (Schallamach, 1971; Hamilton and McCloske, 1997, 1998) as well as in the form of pellets (Rosenau et al., 2009; 2010; Rosenau and Oncken, 2009). It comes in wide range of densities and elasticities. As a stiff solid it has a Young modulus of several MPa while a bulk of rather soft pellets is in the order of 0.1 MPa (Rosenau et al., 2009). Similar to gelatine it has a Poisson ration of 0.5 as a solid. Density ranges between 900 and >2000 kg/m³. Pellets are usually mixed with more rigid particles (e.g. sugar) to adapt elasticity (Fig.

10    5).

### 4.2 Frictional-plasticity

### 4.2.1 Mohr-Coulomb plasticity, frictional instability and rate-and-state friction

Once the forces acting on a rock sample exceed a certain threshold or yield strength, brittle fracture will occur at low confining pressures and temperature while ductile flow may occur at higher temperature and pressure. Both deformation

mechanisms cause permanent deformation, irreversible deformation to occur. Brittle rock deformation as it occurs at shallow to intermediate crustal levels is characterized by a cohesion and pressure-dependent frictional strength (Mohr-Coulomb type behaviour). The latter is in its simplest case described by a linear relation between applied normal load $\sigma_n$ and shear strength $\tau$ in a Mohr diagram, the slope of which is the friction coefficient $\mu$ and the *y*-axis intercept is the cohesion $C$ (Fig. 7a):

$$\tau = C + \frac{\Delta\tau}{\Delta\sigma n} * \sigma n. \tag{17}$$

Fault rocks at very shallow crustal levels ($\sigma_n$ < 100 MPa) rocks have virtually no cohesion and a relatively high friction coefficient of 0.85, while at deeper levels rock appears cohesive ($C \sim 50$ MPa) and has a friction coefficient of ca. 0.6 (Byerlee, 1978).

In the context of analogue earthquakes slip stability is key. Whether slip is stable or unstable depends on three parameters: the stiffness of the system, the dynamic weakening of the frictional interface (either proportional to slip or velocity) and the

applied normal load. Only if the system is soft enough to allow frictional strength to fall faster than the system can response, the force imbalance will cause slip instability to occur. This is described by the condition for slip instability following Eq. (18)

$$\frac{(\mu s - \mu d)\sigma n}{Dc} > K \tag{18}$$

(Scholz, 2002), where $\mu_s$ and $\mu_d$ are static and dynamic friction coefficients, $D_c$ is the characteristic distance over which

friction decreases (slip weakening), $\sigma_n$ is the normal load and $K$ is system stiffness. Accordingly, a stiff system tends to slip stably, while any weakening over short distances and high loads assist instability.



Fictional instability is described nowadays instead of static and dynamic friction rather in terms of the rate- and state-dependent friction (RSF) theory (Dieterich, 1972, 1978b; Ruina, 1983; summarized e.g. in Scholz, 1998, 2002). RSF constitutive laws consider time- and slip dependent restrengthening and variations in friction with slip rate and can reproduce the entire suite of slip phenomena.

RSF theory states that for slip instability to occur the frictional interface has to be velocity weakening, i.e. any increase in sliding velocity $\Delta V$ will reduce the dynamic fiction according to Eq. (19):

$$\Delta\mu d = (a-b)\,\Delta\log(V) \tag{19}$$

(Scholz, 2002), where $a\text{-}b$ is negative and the (dimensionless) parameter describing the net weakening. If $a\text{-}b$ is positive, the frictional interface is said to be velocity strengthening and slips stably. The parameters $a$ and $b$ can be independently derived

from velocity stepping tests and slide-hold-slide test respectively. In velocity stepping tests sliding velocity is changed systematically and the frictional response is measured (Fig. 7b). The data are then evaluated using the Eq. (20):

$$\frac{\Delta\mu d}{\Delta\log(V)} = a-b \tag{20}$$

(Scholz, 2002), which is the slope of the regression to data in Fig. 7b. Its value is in the order of few percent only up to subseismic sliding velocities of cm/s (shallower slope in Fig. 7b). At seismic speeds it may increase dramatically due to

dynamic effects like thermal pressurization or flash melting (Spagnuolo et al., 2016 and references therein).

In slide-hold-slide tests, deformation is halted for various periods (allowing the sample to heal) and then re-started. The static friction during reactivation, i.e. the peak friction to overcome when re-starting, is measured and scales in the preenec of healing with the length of the preceding hold period (Fig. 7c). The data are then evaluated using the following Eq. (21):

$$\frac{\Delta\mu s}{\Delta\log(t)} = b \tag{21}$$

(Scholz, 2002), which is the slope of the regression to data in Fig. 7c. Equation (18) can be rewritten in terms of RSF for a velocity weakening frictional material to Eq. (22):

$$\frac{(a-b)\sigma n}{Dc} > K \tag{22}$$

(Scholz, 1998).

This allows three stability regimes to be defined: The stable regime in which $a\text{-}b$ is positive (velocity strengthening) and slip

always aseismically; the unstable regime in which $a\text{-}b$ is negative (velocity weakening) and slip always seismically; and the conditionally stable regime in which $a\text{-}b$ is negative but Eq. (22) is not fulfilled either because the system is to stiff or the normal load to low such that slip is stable unless triggered by a velocity jump. Earthquakes can consequently nucleate only in the unstable regime but propagate into the conditionally stable regime and will stop in the stable regime. All three regimes have been realized in analogue models.





### 4.2.2 Frictional-plastic rock analogue materials

A large variety of materials both solids and bulk materials show the characteristic rate-and-state dependent frictional response to velocity steps and variable hold times in the respective tests (e.g. Dieterich and Kilgore, 1994; Schulze, 2003). Most granular material like quartz sand or glass beads have similar friction coefficients as rocks ($\mu \sim 0.4$-$0.6$) and a properly

scaled cohesion in the order of few tens to hundreds of Pa (e.g. Krantz, 1991, Lohrmann et al., 2003; Panien et al., 2006; Klinkmüller et al.2016; Ritter et al., 2016; Abdelmalak et al., 2016).

In contrast to sand which shows no measurable velocity-dependence of friction (Fig. 8a) many granular materials of organic origin show rate-and-state-dependent friction: Rice, salt, starch flour, polenta for example show velocity weakening while sugar shows velocity strengthening (Fig. 8a). Schulze (2003) showed velocity weakening behaviour for limestone powder

and wheat flour and velocity strengthening for PE (polyethylene) powder. Healing (i.e. strengthening in static contact) of frictional interfaces is also minor in sand but evident for several materials including wheat flower, cacao (Fig. 8b), PE powder and limestone powder (e.g. Schulze, 2003).

Gelatine and foam both show stick-slip behaviour along pre-cut surfaces controlled by rate-and-state dependent friction. Foam on foam contacts show unrealistically high friction coefficients ($\mu > 1$, Brune and Anooshepoor, 1997). However

coated foams show values similar to rock ($\mu = 0.6$) and a velocity-weakening behaviour (Caniven et al. 2015). Gelatine (pig skin 2.5 %) shows velocity weakening in contact with sandpaper while velocity strengthening when in contact with a plastic sheet (Corbi et al., 2013). In both cases the gelatine contact shows a small friction coefficient ($\mu \sim 0.1$) similar to rock interfaces at high fluid pressures.

The *a-b* values of rock analogue materials are in the order of few percent per decade change in slip velocity similar to rock

(e.g. Scholz, 1998; Dieterich, 2007). As a consequence of velocity weakening, stick-slip occurs in analogue rock materials. Figure 9 shows regular stick-slip behavior as exemplified by various granular rock analogue materials in a Schulze ring shear tester (Schulze, 1994). The latter allows the simulation of faulting in granular materials at loads and velocities typical of analogue models (Klinkmüller et al., 2016; Ritter et al., 2016; Panien et al., 2006; Lohrmann et al., 2003). Accordingly, stick-slip events in rock analogue materials are characterized by partial (10-30 %) stress drops of few kPa which increase

consistently in size and recurrence period with loading rate.

### 4.3 Viscoelasticity

### 4.3.1 Newtonian vs. non-Newtonian and viscoelastic models

Rocks at higher temperature and pressure deform in a ductile manner, that is elasto(frictional)plasticity is replaced by viscoelasticity. Depending on the timescale of applied forces and strain rate the deformation is dominantly elastic (at short

time scales, e.g. coseismic) or viscous (at long time scales, e.g. interseismic).

At long time scales viscoelastic materials show a strain-rate dependent strength $\tau(d\varepsilon/dt)$, the ratio between the two is defined as the viscosity $\eta$ (unit Pa s):




$$\eta = \tau / (\tfrac{\delta \varepsilon}{\delta t}). \tag{23}$$

If the relation is strictly linear, the material is said to be Newtonian. If it is non-linear, the material is said to be Non-Newtonian (either strain-rate thickening or thinning). With respect to ductile rock deformation, Newtonian and shear-rate thinning behaviour is relevant: At low strain rates and high temperatures the dominant crystal plastic deformation

mechanism is diffusion creep (Newtonian) while at higher strain rates and/or lower temperatures dislocation creep (Non-Newtonian) occurs (Bürgmann and Dresen, 2008). The latter is usually described by a power-law constitutive law where the power-law (or stress) exponent $n$ describes the decrease of viscosity with strain rate (Pollard and Fletcher 2005). Typical values for $n$ in the earth are 3-4. Such a pronounced shear-rate thinning behaviour is believed to control strain localization in the ductile regime. High strain rates are typically reached in co- to postseismic phases where shear rate thinning then

overlaps with elastic effects.

At short time scales viscoelastic materials show a delayed, time-dependent, response when stress is applied and/or removed. The classical example is that of a sample where deformation is recoverable but strain accumulation and release are delayed due to the coexistence of both elastic and viscous behaviour simultaneously. The rheological behaviour of viscoelastic material is therefore commonly described using the analogy to physical models of a spring (responsible of the elastic

behaviour) and a dashpot (responsible of the viscous behaviour).

The simplest rheological models describing viscoelastic behaviour are obtained combining spring and dashpot elements in parallel (Kelvin-Voigt model) or in serial configuration (Maxwell model). A way to distinguish between the two rheological models is performing a series of low-stress creep-recovery tests with a rheometer. The test consists in applying a constant shear stress to the sample for a given time interval. The instrument records the strain in the loading phase and in the

following recovery phase. Different shapes of the strain-time curve are then observable depending on the rheological model of the sample (Fig. 10).

The deformation of a sample that follows the Maxwell model shows an instantaneous elastic response followed by linearly viscous flow. When the load is removed the elastic component is recovered instantaneously while a fraction of the deformation linked to the dashpot is not recoverable (Fig. 10a). The constitutive equation for the Maxwell model can be

expressed as follow:

$$\frac{d\varepsilon}{dt} = \frac{\sigma}{\eta} + \frac{1}{E}\frac{d\sigma}{dt}. \tag{24}$$

The deformation of a sample that follows the Kelvin-Voigt model is slowed down by the piston component both in the loading phase and when the load is removed (Fig. 10b). Such slow down effect is highlighted by the curved path of strain as a function of time in both the loading and in the recovery phases. Sample deformation is fully recoverable when the load is

removed. The constitutive equation for the Kelvin-Voigt model is expressed as follow:

$$\sigma = E\varepsilon + \eta \frac{d\varepsilon}{dt}, \tag{25}$$

where $\varepsilon$ is strain, $\sigma$ is the stress, $\eta$ is the viscosity of the material, $E$ is the shear modulus of the material and $t$ is time.



In earth science the Maxwell model is considered the more relevant compared to the Kelvin-Voigt model. A Maxwell body possesses a relaxation time $T_m$, defined as the time required for the viscous strain to become equal to the elastic strain. $T_m$ is given by the ratio of viscosity and shear modulus $G$:

$$Tm = \eta/G. \tag{26}$$

This allows differentiating between relatively short timescales dominated by elasticity and relatively long time scales dominated by viscosity. Applied to the earth, taking as typical values $\eta \sim 10^{19} - 10^{20}$ Pa s and $G \sim$ 10-100 GPa one obtains $T_m \sim$ 3-300 years.

A more elaborate viscoelastic rheology is the Burgers model which shows a share of the responses of Kelvin-Voigt and Maxwell model (Fig. 10c). In particular it features the instantaneous elastic response as well as the transient creep of the

Kelvin-Voigt model. It recently found application in earthquake studies because it allows fitting time-series of postseismic deformation with a single set of parameters. Thanks to the increasing number of geodetic studies of convergent margins it has been pointed out that Earth's mantle response after large earthquakes is characterized by two timescales, a shorter one for the transient viscosity and a longer one for the steady-state viscosity (Wang et al., 2012).

### 4.3.2 Viscoelastic rock analogue materials

Most viscous materials used in analogue modelling of seismotectonic processes (silicones, honey etc.) can be described by the Maxwell model at least under certain conditions. The proper rheological model as well as the constitutive parameters like viscosity, elasticity and the Maxwell relaxation time are inferred from a series of oscillatory and rotational tests in a rheometer (e.g. Rudolf et al., 2016; Di Giueseppe et al., 2009; ten Grotenhuis et al., 2002, Boutelier et al. 2008, 2016; Otterloo and Cruden, 2016).

Polydimethylsiloxane (PDMS), mostly referred to as silicone or silicone, is one of the most common viscoelastic material used in analogue modelling. The rheology of PDMS can be described by a Maxwell model including linear Newtonian viscosity of about $10^4$ Pa s at low strain rates ($< 10^{-2}$ s$^{-1}$), shear rate thinning (stress exponent $n$ = 1-2) above and a Maxwell relaxation time $T_m$ of 0.1-0.2 seconds (e.g. Rudolf et al., 2016). Considering the scaling laws for viscosity and timescales above $T_m$ of silicone is scaling to tens to hundreds of years in nature. These are typical relaxation times in nature and silicone

is therefore a suitable material to simulate seismic cycles, especially the postseismic phase (e.g. Caniven et al, 2015). Hydrogels or suspensions, made of aqueous solutions of polymers used for thickening and stabilization of viscous fluids in the cosmetic and food industry, found widespread applications in analogue modelling. Gelatine has been used in analogue earthquake models, both in fault block models (Corbi et al., 2011) and scale models (Corbi et al., 2013). Gelatine rheology varies as a function of composition, concentration, temperature and ageing. At concentrations > 3 %t the viscoelastic

behaviour of gelatine can be described by a Maxwell model ($T_m$ = 0.1-1 s) while for < 3 % concentrations the rheology of gelatine can be described by a "bi-viscous" model combining a Kelvin-Voigt element in series with a second viscous element (van Otterloo and Cruden, 2016). The rheological properties of gelatine can be modified by adding electrolytes,



phosphate, and non-electrolytes. In particular, the addition of NaCl has been shown to weaken the gelatinestructure (Brizzi et al., 2016).

Hydrogels made of Natrosol, a cellulose polymer, similarly shows a Burgers rheology and Maxwell relaxation times in the order of seconds (Boutelier et al., 2016). Another polymeric viscoelastic material which found application in analogue

modelling of earthquakes by means of deformable slider-spring models is Carbopol ® (Reber et al., 2015). Its rheology depends similar to gelatine on the concentration but additionally on pH. It is very shear thinning and has a yield strength of up to a few hundreds of Pa (Di Giuseppe et al., 2015). It is consequently rheologically modelled as a Herschel-Buckley fluid. It can more generally be described as a brittle-ductile material. The Maxwell relaxation time of Carbopol ® is in the order of 0.1 second.

While hydrogels have complex rheologies with a high potential in analogue modelling, care must be taken during preparation and experimenting. This is because they are generally very sensitive to temperature and concentration which requires careful handling, following strict protocols, and rigorous characterization of the individual rheology. Also, storage is usually limited due to a pronounced sample aging.

Wet Kaolin has recently been recovered as a suitable analogue material with some potential in modelling short and long-term

deformation. It shows the more complex Burgers rheology controlled by the water content (Cooke and van der Helst, 2012). With viscosities in the range of $10^6$-$10^7$ Pa s and elasticities in the order of 10 kPa, relaxation times are rather long (up to 15 min) compared to the previously discussed materials limiting the applicability in scale models of seismic cycles.

## 5 Analogue earthquake monitoring techniques

Advances in analogue rock material characterization have been paralleled by the development of new monitoring techniques

allowing high resolution quantitative measurements of the deformation of analogue models. Monitoring techniques as applied in analogue earthquake models can be grouped into local (at a point in space), regional (mapping an area) and global (integrating over an area or volume) techniques. They are decribed accordingly in Sect. 5.1, 5.2 and 5.3. They differ in their temporal and spatial resolution as well as the coverage. They may further be differentiated into direct and indirect observation methods. The main monitoring techniques used in analogue earthquale modelling are summarized in Table 5.

Local monitoring techniques (Sect. 5.1) provide time-series of point measurements and include quasi-seismological and quasi-geodetic techniques. They use e.g. accelerometers, acoustic sensors, strain gauges or laser interferometry which provide temporally high resolution time series of displacement at a single location. Most of these techniques can be considered indirect as they do not observe the process directly but inversion techniques are required to describe the analogue earthquake source. In contrast regional techniques (Sect. 5.2) surfaces are mapped directly by means of deformation.

Regional techniques are sometimes called "full field" techniques. They include photoelastic and digital image correlation techniques and allow, respectively, high coverage, full field, stress and strain monitoring of the model surface and fault at high spatial but generally lower temporal resolution compared to local and global monitoring techniques. Global methods



(see Sect. 5.3) are those providing a kinematic or dynamic measurement of an average value integrated across a surface area or volume, e.g. the motion of one side of a sample or the loading stress. Those measurements are necessarily indirect but can usually be inverted easily using geometric tools to a direct measure of interest (e.g. fault slip, stress drop).

## 5.1 Local monitoring techniques

Johnson et al. (1973), Wu et al. (1972) and Hamilton and McCloskey (1998) used an array of strain gauges to monitor model motion at up to few hundreds of Hz. Brune et al. (1990) in his foam block models used an instrumentation of digital velocity transducers and accelerometers as well as microphones embedded in the foam block and along its surface. Absolute stress and stress drop has been measured using an in line hydraulic pressure gauge. All embedded sensors were designed with a low mass and high dynamic range to allow measuring acceleration up to hundreds of $g$ as expected in the foam models.

Brune and Anooshehpoor (1998, 1999) used ultralight accelerometers with a dynamic range of +/- 1000 $g$ and a flat response between 1 Hz and 20 kHz. To additionally reduce mass loading effects, they mounted the accelerometers on Styrofoam disks.

Optical techniques exploiting brightness changes between successive images of a target were also developed since the beginning of analogue earthquake modelling. Deformation along the analogue fault in foam were detected by Brune et al.

(1973) using a photocell focussing on a black-and-white target along the analogue fault line. Hartzell and Achuleta (1979) developed a new optical monitoring technique using a light sensitive field effect transistor and an analogue to digital recorder to measure particle motions in the near and far-field of an analogue fault embedded in foam block. Brune and Anooshehpoor (1998, 1999) experimented with a telescopic, 2-axis position-sensing detector which was focused on a small light emitting diode (LED) embedded in the foam. They report a resolution of 0.1 mm.

Nowadays digital displacement and force sensors as well as ultralight microelectromechanical systems (MEMS) are available. Sampling rates are typically kHz to MHz. For example, Dieterich (1978a) and Ohnaka and Kuwahara (1990) used semiconductor strain gauges to monitor analogue earthquakes in granite in a block-model setup. Pressure gauges have been used by Niewland et al. (2000) to measure stress "in situ" in analogue models potentially useful for analogue earthquake models in future.

Acoustic sensors have been widely applied to study analogue earthquakes: Johnson et al. (1973); Wu et al. (1972), Okubo and Dieterich (1984) used piezoelectric transducers to estimate slip  rate and rupture speed in stick-slip experiments on precut rock and rock analogue materials. Lockner et al., 1991, Zang et al. (2000) and Thompson et al. (2005; 2006, 2009) and used acoustic emissions to monitor localization, precursory phenomena and stick-slip ruptures in rock specimens. Varamashvili et al. (2008) and Zigone et al. (2011) use acoustic emission to characterize the stick-slip process in a spring-

slider and salt-slider setup, respectively. Zang et al. (1998), Kwiatek et al. (2014) and Stierle et al. (2016) used acoustic emissions to further constrain the source of laboratory earthquakes in loaded rock specimen by means of seismic moment tensor and b-value. Acoustic sensors usually have high sampling rates (kHz) and work for accelerations up to several g. A



thorough review of the large body of literature on acoustic emission as a seismological tool in laboratory earthquake studies is given by Lei and Ma (2014).

Most recently, laser velocimetry based on interferometric techniques has been used to get displacement time series at selected points on the surface of the specimen (e.g. Lykotrafitis et al., 2006; Rubino et al., 2015; Caniven et al., 2015). The
instruments record a specific component of the velocity field at up to 10 m/s at picosecond temporal resolution. Usually a set of instruments is distributed across the surface of interest.

## 5.2 Regional monitoring techniques

### 5.2.1 Photoelasticity

In many earthquake studies using fault block models (Rubio and Galeano, 1994; Rosakis et al., 1999; Xia et al., 2004; Lu et
al., 2009, 2010; Mello et al., 2010, Schubnel et al., 2011, Nielsen et al., 2010) photoelasticity combined with high speed photography is used to monitor the transient deformation and stresses associated with earthquake-like slip events. Photoelasticity provides not only a visualization of small scale deformation but a direct and quantitative measurement of stress in suitable materials (e.g. Jessop and Harris,1960).

Photoelasticity is physically based on the fact that when polarized light passes through a stressed birefringent material, the
light separates into two wavefronts traveling at different velocities, each oriented parallel to a direction of principal stress in the material, but perpendicular to each other. Different values of the refraction index are assigned to two component that are out of phase when leaving the birefringent material. This difference in optical path can be measured by interferometry and visualized using a second polarizer. The resulting fringe patterns correspond to isocontours of maximum shear stress. This assembly forms the base of so-called "polariscopes".

In analogue earthquake studies light sources like laser beams or floodlights are used to illuminate the transparent model made e.g. of homalite, polycarbonate or gelatine. A pair of linear or circular polarizers, one in front and one behind the model, forms the basis of the experimental polariscope assembly. Usually the light path and therefore the viewing perspective is parallel to the fault plane and normal to the rupture direction.

Photoelasticity is able to monitor the distribution of maximum shear stress in the model at full coverage and high resolution.
However, the absolute values of the principal stress components remain unknown. The temporal resolution is only limited by the sampling rate of the employed digital cameras which is generally flexible and can be adapted to the expected rupture velocity. I.e., while rupture monitoring in rigid materials require high speed cameras are needed (kHz imaging), commercial video cameras with 25 Hz imaging do the job in soft gelatine model approaches as the rupture velocity is drastically reduced. Photoelasticity works best in quasi two dimensional models providing plane strain deformation fields. A thorough review of
dynamic photoelastic applications in fault block models is given by Rosakis et al. (2007).



### 5.2.2 Image correlation techniques

Image correlation techniques aim at retrieving the shape and deformation of a surface or volume from digital images (e.g. Sutton et al., 2009). In the framework of experimental deformation monitoring usually successive optical images are analysed to quantify incremental displacements, from which strain rates can be calculated (e.g. Adam et al., 2005; 2013).

Elastic components of the strain field can be converted into stresses using the constitutive properties of the sample (e.g. Hild and Roux, 2006; Rubino et al., 2015).

A variety of digital image correlation (DIC) algorithms exists. They generally make use of successive monochromatic digital images in which pattern of few pixels can be tracked at sub-pixel accuracy. Given modern image resolutions of up to 30 MPx and 16 bit monochromatic color depth allows tracking millimeter sized features at the micrometer displacement scale.

In combination with high-speed cameras, this technique provides dynamic deformation monitoring options of unprecedented accuracy and precision. Commercial and non-commercial software packages are on the market including LaVision's Strainmaster®, Correlated Solutions Inc.'s VIC™, open-source software MicMac (Galland et al., 2016), MATLAB®-based open toolboxes MatPIV and TecPIV (Boutelier, 2016) and COSI-Corr (Leprince et al., 2007).

The latest developments in strain monitoring using image correlation techniques included a coupling of strain monitoring of

the experiment with analytical or numerical elastic dislocation modelling (EDM). For example Rosenau et al. (2009, 2010) used elastic EDM to differentiate between elastic and plastic deformation inherent their elastoplastic models. Rubino et al. (2015) used EDM for invert strain for stress applying a linear constitutive behaviour. Caniven et al. (2015) used EDM to invert surface deformation for fault slip distribution and depth of locking.

### 5.3 Global monitoring techniques

Brune et al. (1990) used a pen attached to one foam block of their fault block models moving over a strip chart recorder to derive the displacement time function of one side of the fault. Similar data were obtained by Rosenau et al. (2009) using a high-resolution electronic odometer allowing to monitor motion at micrometer scale and kHz sampling rate to derive the displacement time function of the rigid basal plate simulating subduction.

Force sensors at sampling rates up to kHz are routinely used to monitor the forces acting on one side or across an area of a

sample in all deformation apparatuses (e.g. in a Schulze ring shear tester, data in Fig. 9) including spring-slider and fault block setups (e.g. Corbi et al., 2011).

## 6 Applications

### 6.1 EQ statistics

Earthquake statistics deals with the probabilistic treatment of size and frequency of earthquakes by means of frequency-size

distributions, probability distribution functions (pdf), b-values and coefficients of variation.



The iconographic Gutenberg-Richter distribution is by far the most prominent result of earthquake statistics. It is a cumulative frequency plot of earthquakes occurring generally in a large area over a long period. It shows a negative loglinear correlation with a slope ("b-value") of -1, i.e. a power law distribution. It is considered evidence for the self-similarity of earthquakes. The b-value, describing the relative amount of large versus small events, might change however depending on

fault orientation and stressing level (e.g. Schorlemmer et al., 2005).

The original Burridge-Knopoff model (Burrdige and Knopoff, 1967) was able to mimic the self-similarity of earthquakes as well. Accordingly, two types of events can be distinguished: Local events which "smooth" existing stress heterogeneities and which obey a Gutenberg-Richter distribution and system-sized events which recur more regularly. Similarly, Hamilton and McCloskey (1997), when investigating the frequency-size distribution in a simple fault block model, they found a power law

behavior up to analogue earthquakes approximately the size of the smallest dimension of the setup. Larger events occurred more often than predicted by extrapolation of the power law. They concluded that a break in slope of the Gutenberg-Richter distribution is due to the change in rupture mechanism from truly two-dimensional to quasi one-dimensional once the earthquake ruptured the whole seismogenic width.

A simple measure of periodicity is the coefficient of variation (CV) of recurrence intervals. It is defined the standard

deviation divided by the mean recurrence interval. Recurrent events with a CV<50 % can be considered quasi-periodic as their frequencies follow a normal pdf. CV>50 % is considered aperiodic. Aperiodic events may follow an exponential pdf (CV=1) indicating random occurrence or a long tail pdf (e.g. gamma pdf with CV=2) indicating clustering. Brune et al. (1990) in their foam models found a high periodicity of recurring characteristic stick-slip events with a CV~10 %. Similarly elastic sliders by Corbi et al. (2011) or stick-slip of granular materials as shown in Fig. 9 generally display regular stick-slip

with minimal variability in size and frequency given that extrinsic factors (load, loading rate) are kept constant. This regularity can be understood by means of system-sized characteristic events, which rupture the whole fault area resulting in very homogenous stress release. Slightly more complexity is introduced in scale models allowing the rupture to propagate more freely and introducing spatially more heterogeneous stress drops. The subduction earthquake models by Rosenau et al. (2009) and Corbi et al. (2013) for example generate sequences with a CV of 20-30 %. Such a quasi-periodic behavior and its

breakdown controlled by the degree of freedom by means of the seismogenic patch size has also been numerically simulated and studied more systematically by e.g. Nielsen et al., (2000) and Herrendörfer et al. (2015).

Allowing earthquake interactions by means of static stress coupling or off-fault plasticity seems critical in controlling earthquake recurrence behavior. In particular, static stress transfer between two seismogenic patches results in a switch from periodic to random behavior, rarely synchronized (Sugiura et al., 2014, Varamashvili et al. 2008) as suggested by numerical

models (Kaneko et al. 2010) and preliminary experimental results from the Rosenau et al. (2010) setup shown in Fig. 11. This is consistent with simple spring-slider experiments (e.g. Burridge and Knopoff, 1967; King 1991; 1994) and fault block models (e.g. Rubio and Galeano, 1994) where complexity emerges naturally.

Interaction with plastic deformation, i.e. faulting, of the hanging wall in the subduction earthquake models of Rosenau and Oncken (2009) resulted similarly in a more randomized recurrence of analogue earthquakes. Similarly with viscoelastic





wedge models: While pure gelatine models (Corbi et al., 2013) display a very regular stick-slip (characteristic earthquakes) modified gelatine models tend to show more random behaviour (Brizzi et al., 2016). The rheological properties of gelatine in the latter have been modified by adding NaCl which caused an increase of viscoelastic behaviour. This increase in turn affected analogue earthquake statistics and widened the range of earthquake magnitudes, recurrence times and rupture

durations by a factor of two.

Quasi-periodic events can be potentially described by slip-predictable and time-predictable recurrence models (e.g. Weldon et al., 2004): In slip predictable models the amount of slip depends on the duration of the previous interseismic periodic while in time-predictable models the duration of the interseismic period depends on the size of the last event. However, no indication for such a predictability has been found in nature nor in analogue earthquake models (e.g Rubinstein et al.,

2012a,b). Nevertheless, a distinctive bimodal distribution in models by Hamilton and Closkey (1998) as well as in the models of Rosenau et al. (2009) emerges where smaller (but still large) events follow a distinctly different, though well-defined frequency distribution than larger events. In contrast spring-slider models by Burridge and Knopoff (1967) and King (1994) as well as some fault block models (e.g. Rubio and Galeano; 1994) show a more random behavior.

## 6.2 Seismic versus aseismic faulting

Tectonic faults are known to accumulate slip unsteadily on a wide range of rates: from sudden, seismic wave releasing slip instability at speeds of m/s to slow aseismic creep around the tectonic loading rate at mm/yr. Only if active faults become partially locked, that is no or very little slip (less than loading rate) occurs across the fault interface, elastic energy is re-stored in fault bounded blocks effectively. We are now aware that stick-slip in only one endmember of the cyclic storage and release of elastic energy known as "seismic cycles" while continuous (secular) creep at loading rate is the other endmember.

A wide variety of slip transients (slow or silent earthquakes, non-volcanic tremors, very low frequency earthquakes) occur in between these endmembers (e.g. Peng and Gomberg. 2010). Understanding the physical foundations that govern the transition between the two dynamical regimes is pivotal for seismic hazard assessment as it may allow distinguishing between quiet faults and potentially dangerous ones.

Sliding in spring-slider and fault block models may occur both through a typical see-saw profile of stress reflecting phases of

accumulation (i.e., stick phase) alternated by sudden drops (i.e., slip phase) or through smooth and continuous motion. The first regime, also known as "stick-slip", represents the basic physical model for the seismic cycle while the second, also known as "stable-sliding", is the analogue of creeping.

Deformable slider-spring setup of Reber et al., (2015) is used to study transients in the brittle-ductile regime. The latter is defined as a two-mineral-phase regime where one phase deforms in a brittle manner while the other is ductile (e.g. feldspar

vs. quartz). In the experiments by Reber et al. (2015) a viscoelastic material is used to induce both creep and fracture. The observation of slip transients at various speeds in such experiments may be equivalent to tremors and slow slip phenomena (Peng and Gomberg. 2010). Similar variability of slip styles can be simulated using e.g. rice in a ring shear tester setup (Fig. 12). Here shear and normal stress seem to be the controlling factors.



Fault block models have been specifically designed to investigate frictional dynamics as function of the system loading rate, material rheology and interplate roughness. In general, a bifurcation from stick slip to stable-sliding is observed with increasing the system loading rate (e.g., Baumberger et al., 1994). A similar transition from potentially seismic to aseismic behaviour have been speculatively applied to subduction megathrusts, where the observed earthquake magnitude decrease

with depth and the subsequent switch off at the downdip limit of the seismogenic zone may be explained by a progressive decrease of the viscosity of the upper plate (Namiki et al., 2014) or by the progressive smoothing of the interplate roughness (Voisin et al., 2008; Corbi et al., 2011).

## 6.3 Rupture dynamics

Rupture dynamics is by far the widest field of application including the study earthquake nucleation, the transition to

dynamic rupturing and arrest. We can give here only a small overview of the vast amount of existing knowledge highlighting the experimental contributions using analogue earthquake models. The latter include mainly fault block models where a precut surface in rock or rock analogue material is stressed by application of far-field compressive or shear forces.

The nucleation of an earthquake, i.e. the onset of frictional instability has been investigated with a variety of analogue models. It was studied experimentally using fault block models using pre-cut rock (e.g. Dieterich, 1978a; Okubo and

Dieterich, 1984; Ohnaka and Shen, 1999, McLaskey and Kilgore, 2013, McLaskey and Glaser, 2011, McLaskey et al., 2012) as well as rock analogues, e.g. polycarbonate (e.g. Nielsen et al., 2010; Kaneko and Ampuero, 2011). Accordingly, the onset of frictional instability is characterized by quasi-static creep up to loading velocity, acceleration and dynamic propagation. Based on theoretical considerations using a spring-slider system Roy and Marone (1996) suggest the transition to occur at a critical velocity which is a function of extrinsic and intrinsic frictional properties and mass. Latour et al. (2013) in contrast,

based on empirical results using a fault block model and theory, equal the transition from quasi-static to dynamic with the transition from exponential growth to power law growth of the rupture length. They suggest that elastic frictional properties control the transition.

Regarding rupture propagation, two main mechanisms can be distinguished depending on slip duration at a single point along the fault with respect to total rupture duration. In the "crack model" slip at a point is continuous for about the entire

rupture duration while in the "pulse model" it slips only for a small fraction of the rupture duration (e.g. Heaton, 1990). Understanding what governs the duration of slip at a point is crucial for earthquake hazard assessment because the two models predict different degree of strong motions with distance from the nucleation site (Marone and Richardson, 2006).

Brune et al. (1993) were amongst the first to find slip pulses travelling along interfaces of foam and relate them to earthquake dynamics. They argued that normal vibrations reduce the load on the fault at the rupture tip and thereby allows

the rupture to propagate in a self-sustained, wrinkle-like manner and slip to occur at very low friction. Similarly, Schallamach (1971) and Rubinstein et al (2004) reported detachment waves in experiments using rubber on hard ground and between PMMA blocks, respectively. Slip pulses were also found as the main rupture mechanism by later studies in different materials (e.g. Lykotrafitis et al., 2006; Nielsen et al., 2010). Lu et al (2010) suggested that a low stress level along faults



may support pulse like behavior. The role of slip pulses as an earthquake mechanism was studied more systematically using foam block models in order to explain the heat flow paradox associated to the San Andreas Fault (Anooshepoor and Brune, 1994).

Using the same experimental technique Anooshepoor and Brune (1999) verified theoretical predictions of Weertman (1980) and Andrews and Ben-Zion (1997) regarding the directivity and speed of slip pulses travelling along contact interfaces between differentially compliant media. Key findings were that slip pulses propagate into the direction of the particle motion in the more compliant medium at a rupture velocity close to the shear wave velocity of the more compliant medium. Similar results were found by Xia et al. (2005) using much stiffer, bimaterial interfaces (Homalite). The role of the bimaterial character of fault interfaces were studied in depth numerically in recent times (e.g, Ma and Beroza, 2008, Ampuero and Rubin, 2008; Ampuero and Ben Zion, 2008; Brietzke et al., 2007, 2009).

Consistent with the above and with Rosakis et al (1999), who found that cracks can move at velocities faster than shear wave speed ("supershear" ruptures), Lykotrafitis et al. (2006) found that pulses are generally characterized by a slower propagation velocity than cracks. Accordingly, the origin of the two different types of rupture modes depends on the strength of the initial forcing. Similarly, Xia et al. (2004) found that in their experimental setup, the sub-shear to supershear transition transition depends on the dynamic loading conditions.

The control of other parameters than rupture mechanism on rupture velocity has been studied by a variety of approaches: Wu et al. (1972) found, using precut Columbia Resin, that propagation velocity can range in general from sub-shear to 110 % of the shear wave velocity. Johnson et al. (1973) found, using precut rock specimens, that particle velocity and rupture speed increases with stress drop consistent with theoretical predictions. Okubo and Dieterich (1984) showed that rupture velocities along a simulated fault in granite are lower on rough faults than those on smooth fault. Fault block models have also been developed to investigate how different configurations of roughness affect the rupture propagation. It has been found that a single linear barrier may both accelerate and decelerate a rupture while a large and distributed barrier slow down the rupture (Latour et al., 2013). Rousseau and Rosakis (2009) investigated the effect of more complex fault geometries including kinking and branching on rupture propagation in a Homalite material. In parallel, Templeton et al. (2009) were able to simulate experimental results numerically. The control on rupture velocity in general and supershear ruptures in specific is a very active field in analogue earthquake studies (e.g Lu et al., 2009; Schubnel et al., 2011; Mello et al., 2010, 2014, 2016; Passelegue et al 2013, 2016).

Recently, scale models became available which allow studying rupture dynamics in a subduction setting (Corbi et al., 2013). Because of the slowness of the earthquake process in viscoelastic gelatine models, rupture dynamics can be studied at high resolution. Key characteristics of earthquake ruptures in viscoelastic subduction zone models of Corbi et al. (2013) regarding rupture nucleation, directivity and mechanism are as follows:

(1) Hypocenters concentrate near the base of the seismogenic zone (Fig. 13). This is consistent with numerical simulations (Das and Scholz 1983; van Dinther et al., 2013a,b, Pipping et al., 2016). In nature, the spatial relation between the hypocenter and the rupture area is less clear: The hypocenter of the 2004 M9.2 Sumatra and 2010 M8.8 Chile earthquakes





were located in the deepest part of the rupture (Rhie et al., 2007; Moreno et al., 2010) probably aided by the lithostatic pressure and locking gradient. Mai et al. (2005) based on observations of 12 finite-source rupture models of megathrust earthquakes, suggested that hypocenters tend to concentrate in the along-dip center of the fault. Similarly, the hypocenter of the 2011 M9.0 Tohoku-Oki earthquake was located in the along-dip center of the rupture (Lee et al., 2011).

(2) Ruptures propagate bilaterally with preference in the updip direction (Fig. 13) in viscoelastic models of Corbi et al. (2013). This behavior is consistent with previous analogue models of interplate seismicity performed with foam rubber (Brune, 1996) and elastoplastic materials (Rosenau et al., 2009) as well as numerical simulations (van Dither et al., 2013 a.b, Pipping et al. 2016). The most likely explanation for the preferential earthquake migration to shallow levels is that the rupture follows the lithostatic pressure gradient that results from the thrust geometry (Das and Scholz, 1983). Also the

material compliancy difference between gelatine and aluminium favours the upward migration direction. Such bimaterial contrast may be active also in nature where the overriding plate is expected to be the more compliant than the subducting one (e.g., Ma and Beroza, 2008).

(3) The majority of ruptures are crack-like as they display a minimum duration of slip at a point larger than 1/10 of the entire rupture duration. The spatio-temporal cumulative slip distribution also seems to support the idea of growth of the rupture as a

crack (Fig. 13). A detailed study based on numerical modelling highlights the occurrence of both cracks and pulses and that the largest stress drops are associated with cracks (Herrendorfer et al., 2015).

The characteristics shown by the viscoelastic scale models by Corbi et al. (2013) are consistent with observations in experiments with elastoplastic models of Rosenau et al. (2009). However, in the latter the rupture process is far less well resolved as the models were stiffer speeding up the process while the monitoring resolution was limited.

**6.4 Ground motion and site effects**

Foam rubber models were also amongst the first to explain the strong asymmetry in particle motion and associated ground motion across dipping faults. Brune (1996) investigated the dynamics of seismogenic thrusting using a wedge shaped foam block. He found a pronounced amplification of particle and ground motion in the hanging wall. He explained this by considering static and dynamic effects: the free surface effect, as predicted by analytical dislocation models, allows higher

static particle motions in the hanging wall because of the possibility of the material to lift up. Additionally, seismic energy is reflected by the fault as well as the free surface and becomes trapped in the hanging wall wedge increasing its coseismic motion. Shi et al. (1998) and Shi and Brune (2005) were able to reproduce and refine the experimental results numerically. Several numerical studies confirmed their results (Oglesby et al., 1998, 2000a,b; Ma and Beroza, 2008, Nielsen, 1998). Gabuchian et al. (2014) most recently revisited this issue by means of experiments using Homalite as a rock analogue.

Beside the symmetric effects they focussed on rupture velocity as a controlling factor for ground motion.

Brune and Anooshepoor (1999) showed the dynamic effect of normal fault geometry and low stress level at shallow level. They found systematically lower accelerations of the model surface near the normal faults when compared to strike-slip faults. Similar results have been obtained with numerical models (Shi et al., 2003; Oglesby et al., 1998, 2000a,b;). The effect





of a shallow weak and creeping zone on ground motions from strike-slip earthquakes has been studied quantitatively using foam models by Brune and Anooshepoor (1998).

A cogenetic though more engineering-type approach has been used to study site effects due to topography (Anooshepoor and Brune, 1989) and sedimentary basins (King and Brune, 1981) or the response of constructions to earthquakes (e.g. Brune and
Anooshepoor, 1991a; Anooshepoor and Brune, 1989). Finally, Brune and Anooshepoor (1991b) simulated a large scale seismic experiment in order to help interpreting the seismic data obtained in nature. In such models, excitation has been not by stick-slip events but by forced vibrations.

## 6.5 Seismic cycle deformation

Since Reid's formulation of the elastic rebound theory following the 1908 San Francisco earthquake (Reid 1911) seismic
cycles in various settings are seen as recurring, more or less sudden release of stress or elastic strain energy which slowly accumulated in the period before. The term cycle by no means implies regularity of the recurring events but rather describes the succession of the archetypical stages. Accordingly, a full seismic cycle consist primarily of the interseismic period (years to Millenia) and the coseismic rupture (seconds). Precursors, postseismic relaxation as well as interseismic transients may complete the seismic cycle. Traditionally, the seismic cycle has been considered as purely elastic (e.g. Klotz et al., 2001) and
modelled accordingly using elastic models (e.g. Fig. 6b, C). The recognition of inter- and postseismic viscoelastic relaxation phenomena in the ductile lower crust (e.g. Wang et al., 2012) and mantle as well as possibly universal precursor activity (e.g. Bouchon et al., 2013) led to continuous refinement of the seismic cycle concept. Finally, in recent years plasticity theory has been formulated in the framework of seismic cycles allowing for the accumulation of permanent (i.e. tectonic) deformation through seismic cycles (Wang and Hu, 2006). Observation both in nature (e.g. Wesson et al. 2015) as well as in experiments
using scale models (e.g. Rosenau and Oncken, 2010) corroborate this new view on elastoplastic seismic cycle deformation.

Seismic cycle deformation using scale models have been realized using elastic (foam, Caniven et al., 2015), viscoelastic (gelatin, Corbi et al., 2013) and elastoplastic rheologies (rubber mix, Rosenau et al., 2009; 2010; Rosenau and Oncken 2009). Scale models were able to reproduce the basic pattern of seismic cycles in subduction zones and strike-slip zones with alternating phases of stress build up (analogue of the interseismic stage) and stress release (analogue of the coseismic stage)
due to coseismic slip associated with uplift and subsidence in the order few micrometres (decimetres to meters if scaled to nature).

Caniven et al. (2015) developed crustal scale three-layer brittle-ductile models by coupling frictional-plastic, elastic and viscoelastic layers in a strike-slip setting. These models intended to study the mechanical coupling between the layers with respect to seismic cycle deformation. The models consist of a 30 mm thick basal layer constituted of viscoelastic silicone
(PDMS), simulating the ductile behaviour of the lower crust. A 50 mm thick elastic middle layer, simulating the seismogenic upper crust, is composed of resilience polyurethane foam which tends to stick-slip along cuts (faults). The brittle/ductile transition corresponds to a 2-3 mm thick impregnation of the silicone into the base of the polyurethane foam, insuring a strong coupling depending on model velocity boundary conditions. Finally, a 5 mm thick upper layer, simulating the



shallow, aseismic crust, is constituted by a frictional-plastic material (mixture of silica and PVC plastic powder). The model is loaded by applying both horizontal compression and shear at velocities in the order of μm/s. Model kinematic evolution is monitored using a high performance optical system, based on sub-pixel correlation of HD digital images, enabling very accurate measurements of model deformations with a spatial resolution ranging from 1 to 5 mm, an accuracy of a few

microns (equivalent to a 1 km spacing dense permanent GPS network) and a sample rate of 0.2 Hz.

Caniven et al. (2015) used an average length scale of $L^* = 4\cdot10^{-6}$ such that 1 cm in the model equals 2-3 km in nature. This scale was mainly imposed by the maximum size of the experimental box (1 m) and the dimensions of the studied geological prototype, i.e. a 200-300 km long strike-slip fault. The calculated stress scaling factor is very close to the length scaling factor and was estimated in the range of $\sigma^* = 5\cdot10^{-6}$ (taking into account that the normal stress was increased to compensate

the low density of the foam and an shear modulus scaling factor $G^*$ of $5*10^{-6}$). The silicone oil used to model the ductile lower crust has a viscosity of $\eta = 3 - 5\cdot10^4$ Pa s at $T = 20°C$ for a mean strain rate of $d\varepsilon/dt = 10^{-4}$ s$^{-1}$ (Ten Grotenhuis et al., 2002; Rudolf et al., 2016). Consequently, the viscosity scaling factor ranges between $\eta* = 5\cdot10^{14}$ and $5\cdot10^{16}$ considering a viscosity of $10^{18}$ to $10^{20}$ Pa s for nature. Altogether, the resulting interseismic time scaling factor should be in the range between $10^{-8}$ to $10^{-10}$ which corresponds to 1 second is 3 to 300 years.

This experimental set-up by Caniven et al. (2015) succeeded in simulating realistic long-term elastic loading phases (interseismic) followed by almost instantaneous fault slip events (coseismic rupture) and low amplitude slow deformation phases (postseismic relaxation) whose kinematics can be directly compared with geodetic measurements on natural cases (Fig. 14). For each experiments, the evolution of the interseismic strain field is recorded semi-automatically by; measuring surface deformation, calculating the evolution of the locking depth, quantifying the amount and location of aseismic creep,

analysing the spatial and temporal distribution of coseismic ruptures (surface rupture dimensions and geometries, coseismic slip profiles, earthquake magnitude, return period) and the post-seismic relaxation phase (surface deformation kinematics, decay of micro-earthquake activity). The model results compare to numerical simulations of strike-slip fault earthquakes (e.g. Ben-Zion and Rice, 1997; Lapusta and Rice, 2003, Tullis et al., 2012a, b)

### 6.6 Seismotectonic evolution of subduction zone forearcs: Linking short and long timescales

Lithospheric scale elastoplastic models were used to study seismotectonic evolution of subduction zone forearcs (Rosenau et al., 2009; Rosenau and Oncken, 2009). Such models have helped in understanding the relationship between earthquakes along the subduction megathrust and the structure and topography of the forearc wedge. Thereby they are an valuable tool in understanding the links between short-term and long-term deformation processes, i.e bridging the time scales from single eartquakes to tectonic evolution.

The models of Rosenau et al. (2009) consist of a 200 mm thick granular wedge representing the brittle forearc lithosphere (< 60 km depth) made of a mixture of rubber pellets and sugar, embedded into which is a seismogenic zone of rice grains. The whole model sits on top of a rigid conveyer plate (2d) or a stiff conveyer belt made of rubber (3D) driven at few mm/min to simulate convergence. Kinematic monitoring occurred by the particle image velocimetry method able to detect



displacements down to tens of microns at 10 Hz resolution. While the resolution was good enough to monitor seismic cycle deformation, it was too low to image the rupture process which occurred within 0.1-0.2 seconds. Nevertheless, they succeeded in simulating the main stages of the seismic cycle, namely the co-, post, and interseismic stage. The key issue in using this setup was to study the accumulation of permanent (plastic, tectonic) deformation over several seismic cycles.

Differentiating between elastic and plastic deformation at seismic cycle scale has been done by using elastic dislocation modelling to substract the elastic deformation from the elastoplastic deformation seen.

According to 2d models few percent of plate convergence is converted into permanent across-strike shortening of the forearc wedge over several seismic cycles. Shortening localizes both at the updip and downdip limit of the seismogenic areas along the megathrust (Fig. 15a). At the updip limit of the seismogenic zone, coseismic compression is relaxed postseismically by

internal shortening accommodated by a splay fault in the models of Rosenau et al. (2009, 2010). This is consistent with theoretical predictions (Wang and Hu, 2006) and observations in nature (e.g. Lieser et al., 2014). Vice versa, during the interseismic period compression occurs at the downdip limit of the seismogenic zone and may lead to uplift of the coast over multiple seismic cycles.

Preliminary results from 3D experiments in the Rosenau et al. (2009) setup suggest that a similar mechanism is active along

strike causing permanent shortening and uplift of coastal regions overlying aseismically slipping zones (barriers) along the megathrust (Fig. 15b, c).

In Summary, analogue models suggested that permanent shortening localizes at the periphery of repeating great earthquakes (M8-M9) in subduction zones. In particular, a tectonically stable basin or platform evolves on top of the seismogenic areas along the megathrust. This leads to a morphotectonic segmentation of the wedge which directly reflects the pattern of

seismic and aseismic slip along the megathrust. This mechanism could be responsible for the correlation of basins with source areas of Megathrust earthquakes (Mogi, 1969; Nishenko and McCann, 1979; Wells et al., 2003) and peninsulars with barriers (e.g. Victor et al., 2011; Schurr et al., 2012).

Rosenau and Oncken (2009) moreover suggest a feedback between forearc deformation and seismogenesis along the megathrust: Accordingly, because the stable wedge part overlying the seismogenic zone in segmented forearcs deforms

quasi-elastically, characteristic great earthquakes tend to occur rather periodically as in simple spring-slider experiments and numerical simulations of the experiments (Pipping et al., 2016). In contrast, less segmented subduction zone forearcs have been predicted to show more random earthquake occurrence. This is in line with observations (Tormann et al., 2015) and numerical predictions (Fuller et al., 2006).

## 8 Conclusions and perspectives

We presented an overview of experimental approaches to model earthquakes and seismic cycles. Existing analogue earthquake models can be categorized into spring-slider models, fault block models and scale models according to their complexity and similarity to the natural prototype. Scale models have been developed very recently exploiting advances in



material characterization and monitoring techniques. Materials used in analogue earthquake studies included elastic, frictional plastic and viscoelastic rheologies. Monitoring techniques exist which allow accurate and precise deformation monitoring in the lab at high spatial and temporal resolution. Analogue models provided original insights into earthquake statistics, rupture dynamics, ground motion, seismic cycle deformation and seismotectonic evolution.

The key challenges and development we see are:

1.  New materials remain to be explored. Especially non-linear rheologies both in brittle and viscoselastic regimes will contribute to more realistic analogue models in future. A rigorous material characterization is prerequisite.

2.  Monitoring techniques continuously are developed towards higher resolution both in space and time. A key future challenge is handling the growing amount of image data. Adaptive imaging equivalent to adaptive time stepping in

numerical modelling is a way to be explored in this context.

3.  Coupling of analogue models with numerical models help to overcome the respective limitations. More rigourous cross-validation and benchmarking will contribute to better exploitation of the respective potentials.

Tackling the above challenges will enable us to develop more complex scenarios in terms of structure and rheology. Higher resolutions will shift the detection threshold for analogue earthquakes (i.e. the magnitude of completeness) further down

from currently ca. M7-8 which will allow simulating transient processes more completely including fore- and aftershocks. Finally, longer and more complete experimental time-series will allow further insight especially into the link between short-term and long-term deformation processes. New questions might arise e.g. about the link between megathrust seismogenesis and long-term accretion and erosion processes in subduction zones or about the link between strike-slip seismogenesis and fault growth and linkage on million year time scale.

**Data availability**

Original data underlying the here presented material are published in an open access data set Rosenau et al. (2016).

**Acknowledgements**

FC received funding from the European Union's Horizon 2020 research and innovation programme under the Marie Sklodowska-Curie grant agreement No. 658034 (AspSync). We thank Malte Ritter and Michael Rudolf for sharing data. We

thank Kirsten Elger and GFZ Data Service for publishing the data.

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

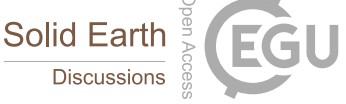



**Figures**

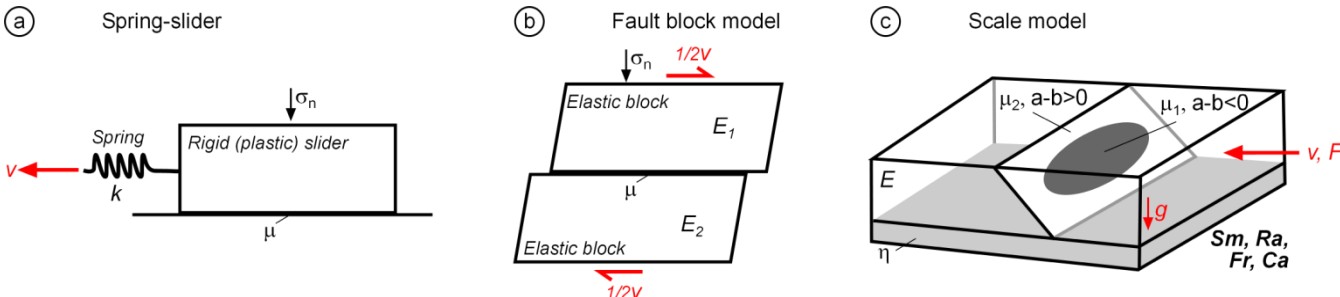

10     **Figure 1:** Analogue model categories: (a) Spring-slider model (*v* is velocity, *k* is spring constant, σv is normal load, μ is friction coefficient), (b) Fault block model (*E* is Young modulus), (c) Scale model *a-b* is rate-state parameter, η is visosity, *g* is gravitational acceleration, *F* is force, *Sm, Ra, Fr* and *Ca* are dimensionless numbers used for scaling, (see Table 2).







**Figure 2**: Scale model setups: (a) Viscoelastic gelatine wedge (RomaTre, Corbi et al. 2013); (b) Elastoplastic granular wedge setup (GFZ Potsdam, Rosenau et al., 2009); (c) Elastic-viscoelastic strike-slip (Caniven et al., 2015) and (d) subduction setup (Dominguez et al., 2015; Univ. Montpellier).





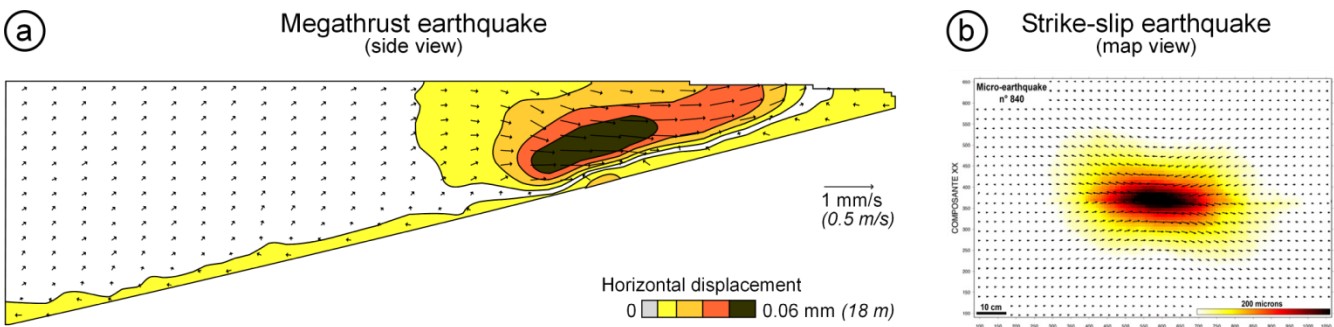

**Figure 3:** Examples of experimental deformation pattern in analogue earthquake models: (a) subduction megathrust earthquake (side-view, Rosenau et al. 2009); (b) strike-slip setup (map view, Caniven et al. , 2015).



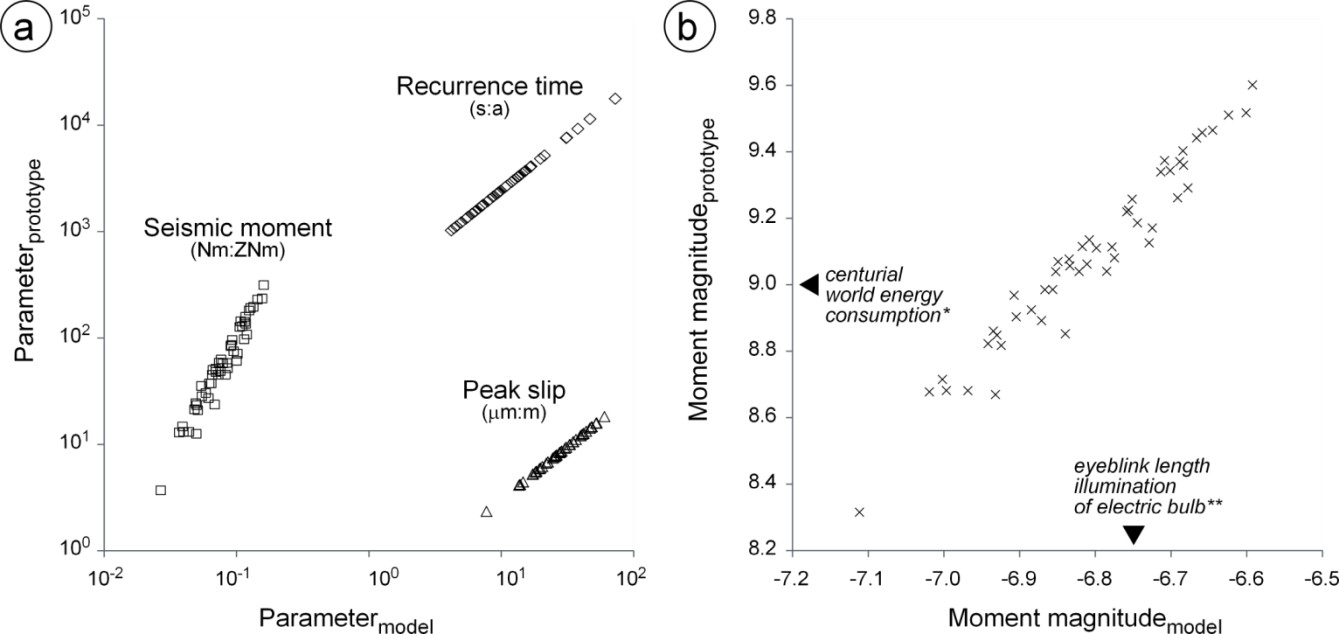

**Figure 4**: Scaling of laboratory scale values to nature (prototype scale) for specified parameters. (a) Scaling for peak slip, recurrence time
5   and seismic moment (the unit relation given in parentheses correspond to the dimensional mapping of the respective axis); (b) Scaling for
moment magnitude (*as of the year 2000; **10 W bulb, ca. 0.1 seconds). Data and methodological details are published open access in
Rosenau et al. (2016).



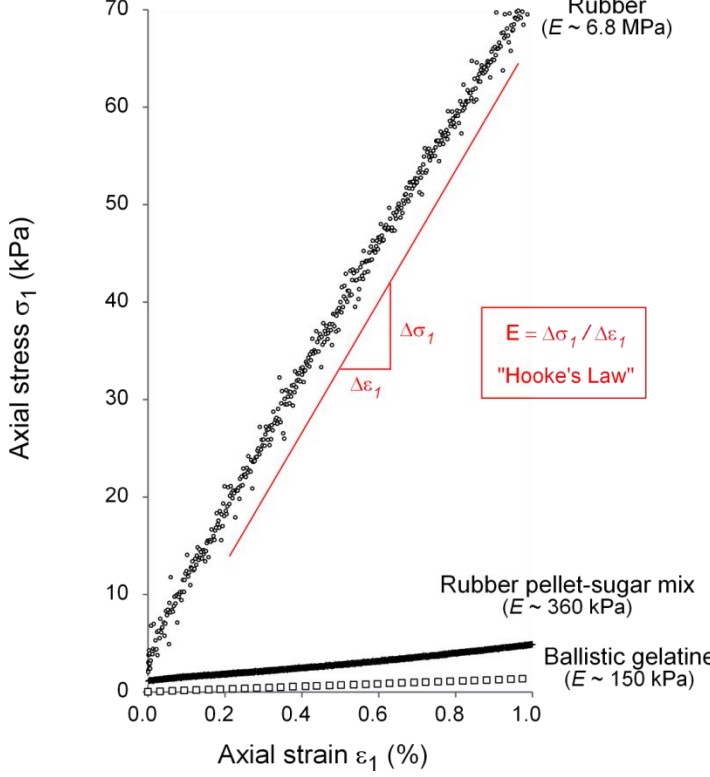

**Figure 5:** Elastic moduli of selected rock analogue materials as measured in an axial tester. Data and methodological details are published open access in Rosenau et al. (2016).





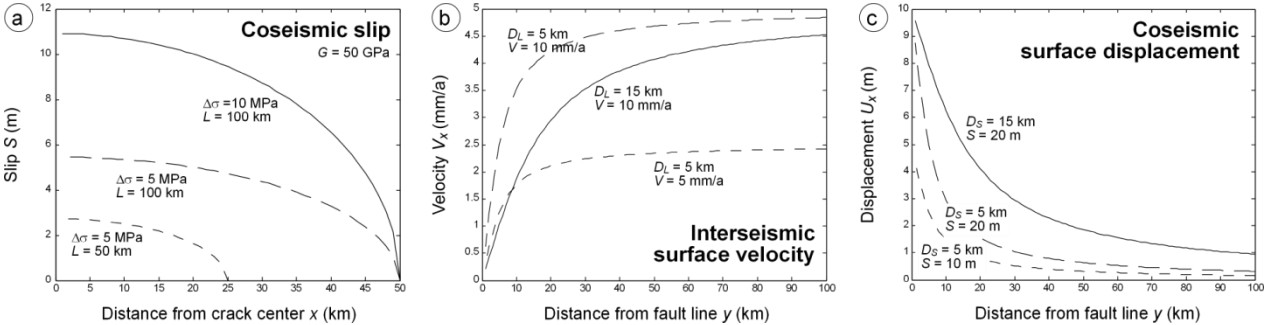

**Figure 6:** Predictions from analytical elastic models: (a) coseismic slip $S(x)$ along a penny shaped crack for different crack length $L$ (50, and 100 km) and stress drop $\Delta\sigma$ (5 and 10 MPa) according Eq. (13); (b) surface velocity $V_x(y)$ away from a locked fault represented by a deep dislocation simulating interseismic loading velocity $V$ (5 and 10 mm/a) of a fault locked to depth $D_L$ (5 and 15 km) according Eq. (15); (c) surface displacement $U_x(y)$ away from a slipped fault simulated by a shallow dislocation with slip $S$ (10 and 20 m) down to a depth $D$ (5 and 15 km) according Eq. (16). Note that only one half of the mirror-symmetric solutions are shown.





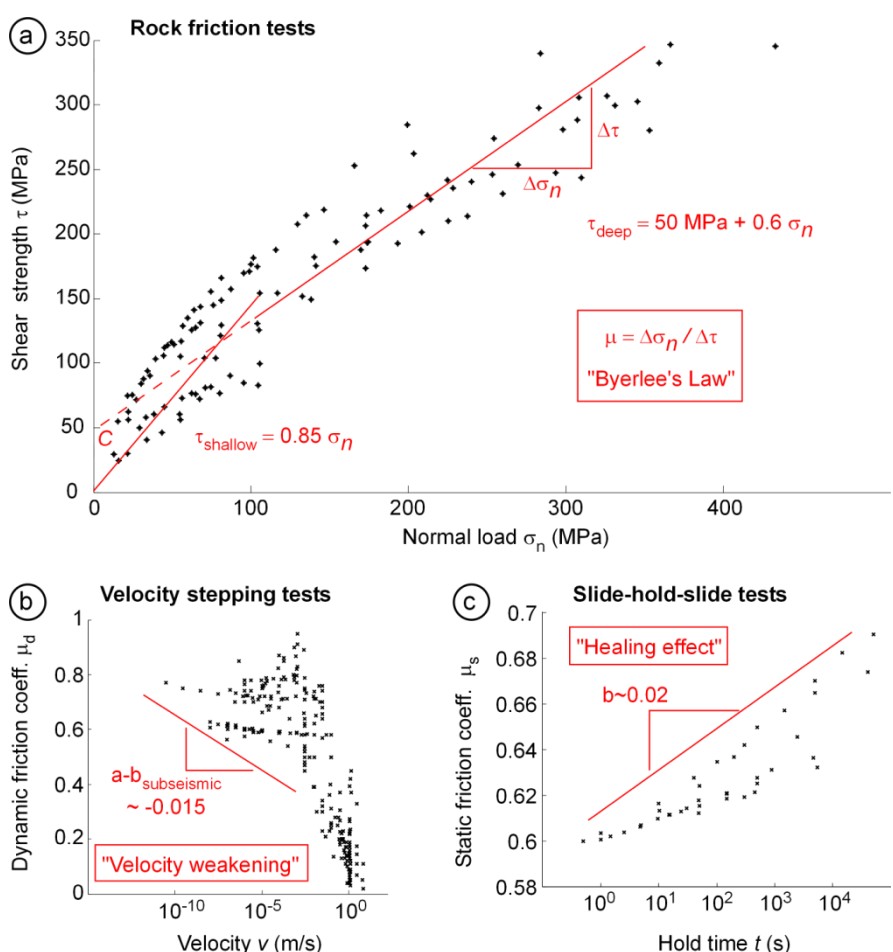

**Figure 7**: Rock friction data: (a) Data from rock friction tests in a Mohr-Coulomb plot showing the pressure dependence of frictional strength ("Byerlee's law", data as compiled by Ritter et al., 2016, references therein). The slope corresponds to the friction coefficient, the y-axis intercept to the cohesion of the rock and rock interfaces. (b) Data from velocity stepping tests (as compiled by Marone, 1998, and Spagnulo et al., 2016, references therein) indicating the velocity weakening behaviour of rock interfaces. The slope corresponds to the parameter (a-b) in the RSF friction law. (c) Data from Slide-hold-slide test (data as compiled by Marone, 1998, references therein) demonstrating healing of rock interfaces. The slope corresponds to the parameter b in the RSF law. Data plotted here are published open access in Rosenau et al. (2016).



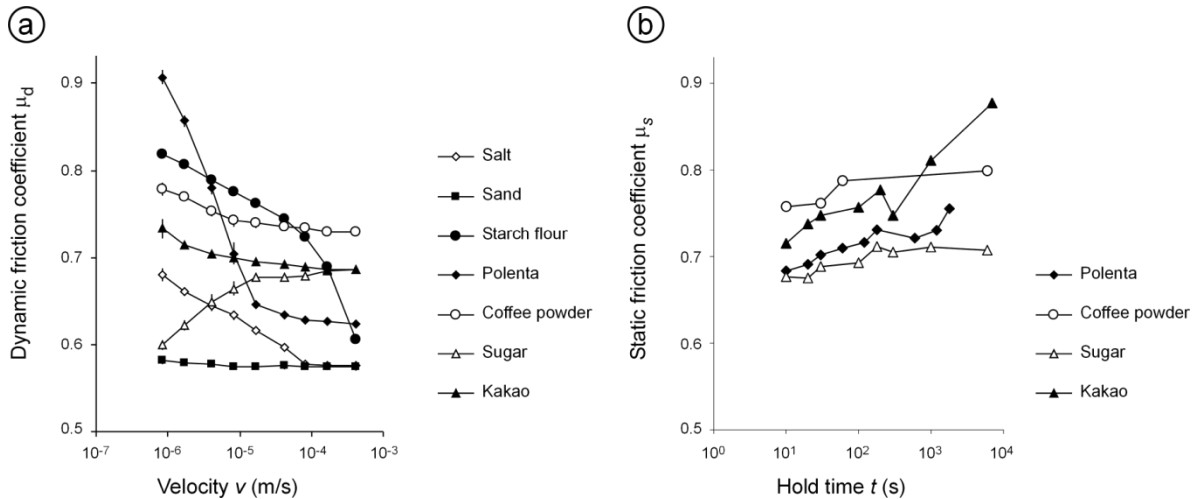

**Figure 8:** Rate-and-state effects on friction of selected analogue rock materials: (a) rate effect (v-strengthening versus v- weakening), (b) state effect (healing). Preliminary data from ring shear tests unsing a Schulze Ring shear tester RST-01 (Schulze, 1994). Data and methodological details are published open access in Rosenau et al. (2016).





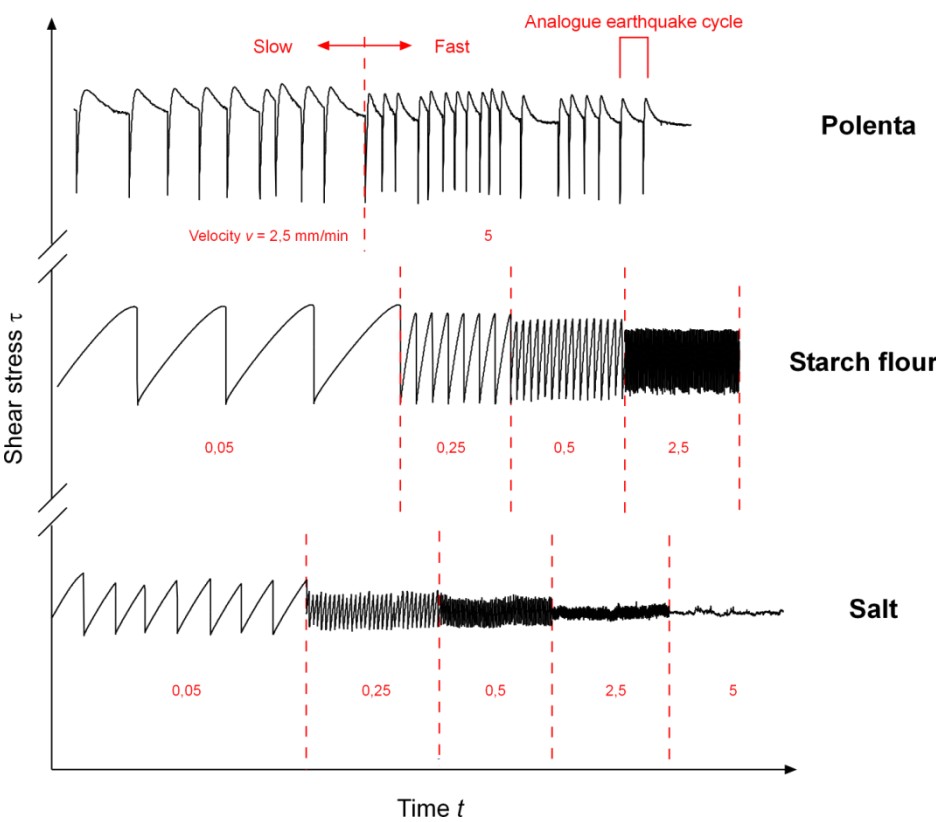

**Figure 9:** Systematics of stick-slip in selected analogue rock materials. Preliminary data from ring shear tests using a Schulze Ring shear
tester RST-01 (Schulze, 1994). Data and methodological details are published open access in Rosenau et al. (2016).





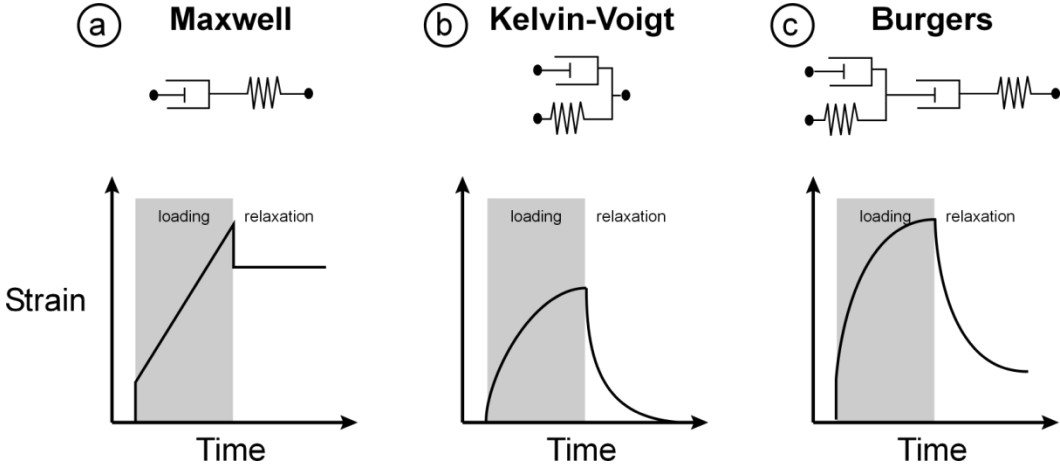

**Figure 10:** Rheological models represented as spring and dashpots under laoding and relaxation with their relative strain-time curve: (A) MAxwell model, (B) Kelvin-Voigt model, (C) Burgers model. See text for discussion.





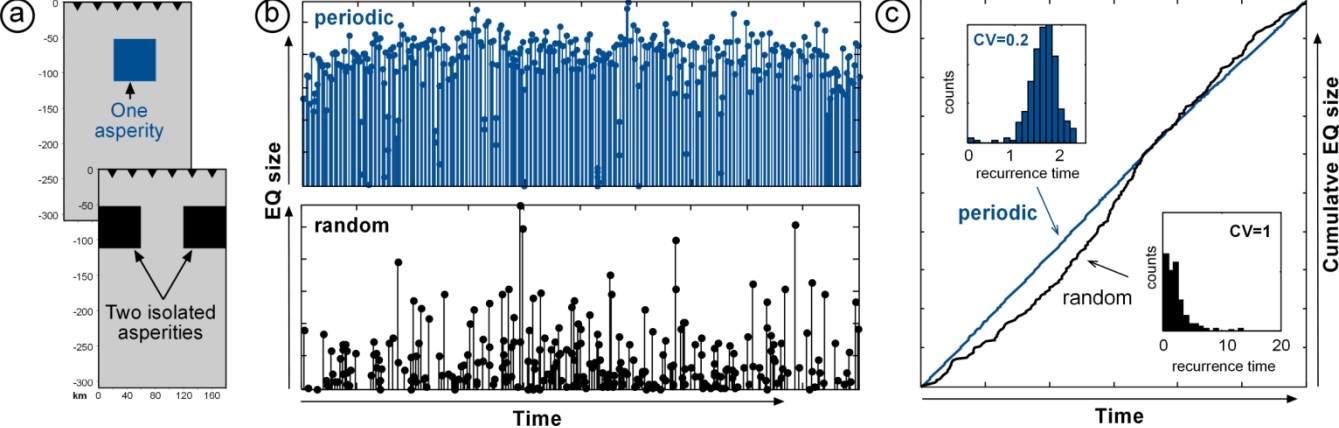

**Figure 11:** Periodic vs. random rupture behaviour as exemplified by scale models: (a) model setup: blue – one asperity, black m two asperities; (b) analogue earthquake catalogues generated by blue and black models; (c) cumulative plot of analogue earthquakes sequences and histograms of frequencies. Data and methodological details are published open access in Rosenau et al. (2016).



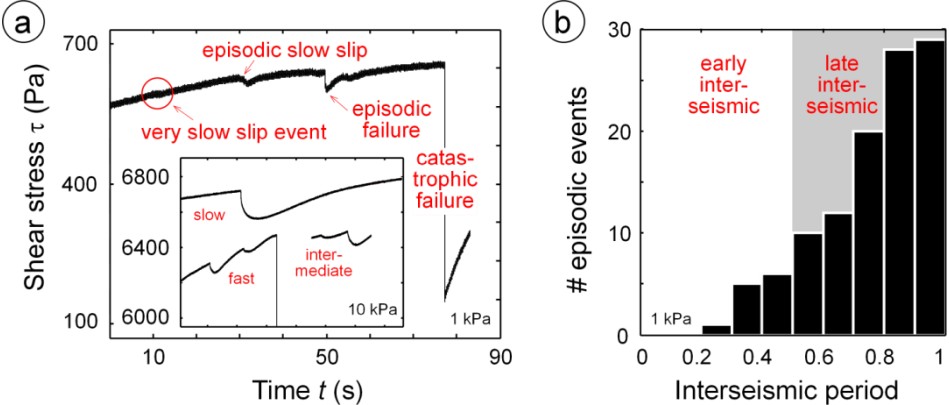

Figure 12: Seismic vs. aseismic slip in a ring shear test using rice. (a) Slip styles ranging from creep to transient slip to "earthquake" slip. (b) Systematic increase of transient slip events towards the end of an analogue seismic cycle. Preliminary data from ring shear tests using a Schulze Ring-shear tester RST-01 (Schulze, 1994). Data and methodological details are published open access in Rosenau et al. (2016).





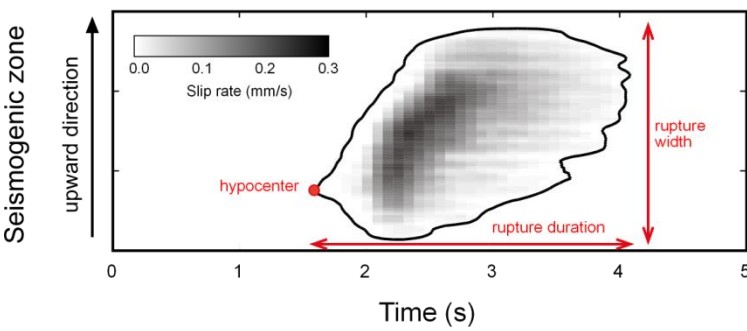

**Figure 13:** Rupture dynamics as observed in the scale models of Corbi et al. (2013): Slip evolution of a crack-like analogue subduction megathrust earthquake propagating up- and downwards the seisogenic zone.





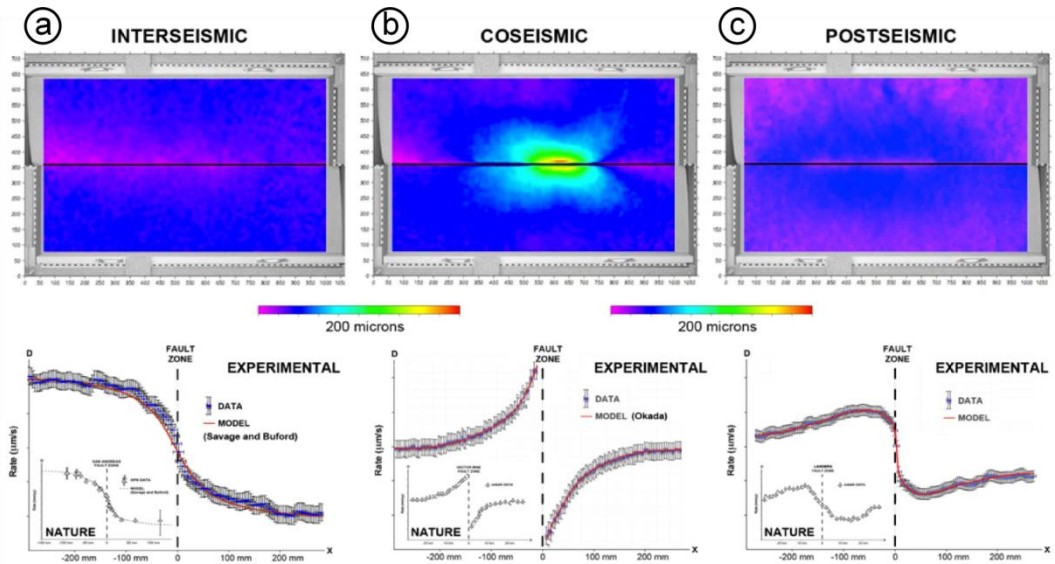

**Figure 14**: Seismic cycle deformation as shown by elastic-viscoelastic models by Caniven et al. (2015): (a) interseismic, (b) coseismic, (c) postseismic phase.





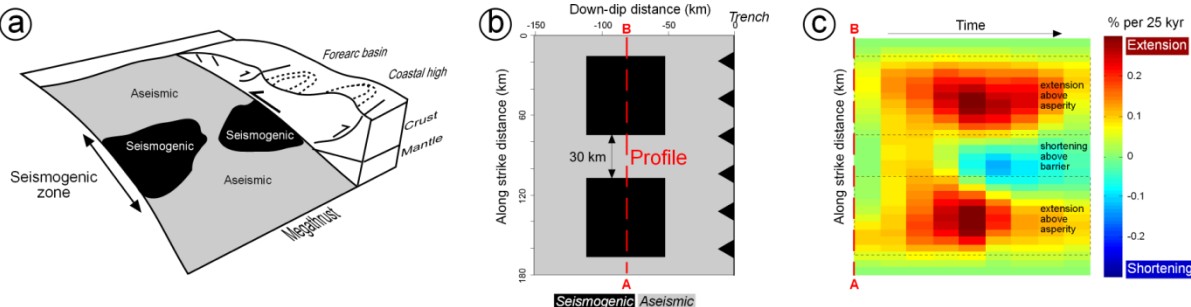

**Figure 15:** Seismotectonic evolution of subduction zone forearcs as suggested by elastoplastic scale models: (a) Spatial correlation
10   between forearc topography and the seismogenic zone along subduction megathrust suggested by the models of Rosenau and Oncken
(2009). (b) Setup of scale model with two seismogenic patches. (c) Evolution of along strike strain emerging from the model shown in B.
Note the correlation between along strike shortening and extension with aseismic and seismic areas along the megathrust.




## Tables

| Category | Setup | Materials | References | Numerical pendants | Scope |
|---|---|---|---|---|---|
| **Spring-slider models** (elastic and frictional elements seperate) | Spring-slider s.s. | | *Burridge and Knopoff (1967), King (1991,1994), Heslot et al. (1994), Popov et al. (2012), Varamashvili et al. (2008)* | *Burridge and Knopoff (1967), Erickson et al. (2011), Thomas (1930); Cao and Aki (1984), Schmittbuhl et al. (1996), Carlson and Langer (1998), Carlson et al. (1991), Gu et al. (1991),Wang (2012), Abe et al. (2013), Mori and Kawamura (2006, 2008), Narkounskaia and Turcotte (1992), Saito and Matsukawa (2007)* | Statistics, scaling |
| | Direct-shear / ring-shear | Glassbeads, Sand, Gouge, Pasta, Wooden rods, Steel bars, rice, etc. | *Leeman et al. (2016), Knuth and Marone (2007), Anthony and Marone (2005), Mair et al. (2002), Alshibli et al. (2006), Schulze (2003), Scuderi et al. (2005), Niejmeier et al. (2008, 2010), Daniels et al. (2008), Nasuno et al. (1998), Jonhson et al. (2013), Schulze (2003); Rosenau et al. (2009)* | *Abe and Mair (2009); Abe et al. (2006), Ferdowsi et al. (2013, 2014, 2015)* | Slip stability, rate/state effects, triggering |
| | Deformable slider-spring | Carbopol® | *Reber et al. (2015)* | | Slip stability, brittle-ductile deformation |
| **Fault block models** (elastic solid(s) in frictional contact) | Shear | Foam | *Brune (1973), Brune et al. (1990, 1993) Anooshepoor and Brune (1994, 1999), Hartzell and Achuleta (1978)* | Weertman (1980), Andrews and Ben-Zion (1997) | Rupture dynamics |
| | Compression | Plexiglass-like (Homalite, resin, polycarbonate etc.) | *Lu et al. (2009,2010); Rosakis et al.(1999, 2007), Rousseau and Rosakis (2009), Cocker et al. (2005), Gabuchian et al. (2004), Rubinstein et al. (2014), Lykotrafitis et al. (2006, 2009); Nielsen et al. (2010), Wu et al. (1972), Xia et al. (2004, 005), Rubino et al. (2015), Schubnel et al. (2014), Mello et al. (2010, 2014, 2016), Kaneko and Ampuero (2011)* | Ma and Beroza (2008), Ampuero and Rubin (2008), Ampuero and Ben-Zion (2008), Brietzke et al. (2007, 2009), Templeton et al. (2009), Kaneko and Ampuero (2011) | Rupture dynamics |
| | Compression | Rock | *Zang et al. (1998), Kwiatek et al. (2014), Stierle et al. (2016), Lockner et al., 1991, Lei et al (2000), Zang et al. (2000), Thompson et al. (2009), Passelegue et al. (2013, 2016), Brace and Byerlee (1966), Brace (1972), McLaskey and Glaser (2011), McLaskey et al. (2012), Blanpied et al. (1991,1995); McLaseky and Kilgory (2013), Lei and Ma (2014), Dieterich (1978a), Okubo and Dieterich (1984), Ohnaka and Shen (1999), Johnson et al. (1973)* | *Kaneko et al. (2008), Ben-Zion and Rice (1997), Lapusta and Liu (2009), Yoshida et al. (2004), Ben-Zion (2001)* | Rupture dynamics |
| | Shear | Rubber | *Hamilton and McCloskey (1997, 1998), Schallamach (1971)* | | Rupture dynamics |
| | Shear | Salt | *Voisin et al. (2007, 2008), Zigone et al. (2011), Mair et al. (2006), Renard et al. (2012)* | | Slip instability |
| | Shear | Hydrogels | *Corbi et al. (2011), Rubio and Galeano (1994), Baumberger et al. (1994, 2003), Namiki et al. (2014), Latour et al. (2013)* | | Slip instability |

**Table 1**: Categories of analogue earthquake models (see text for discussion).





| Category | Setup | Materials | References | Numerical pendants | Scope |
|---|---|---|---|---|---|
| **Scale models** (elasto-visco-plastic solids and fluids with similarity to prototype) | Megathrust | Gelatine | *Corbi et al. (2013)* | *Van Dinther et al. (2013), Herrendörfer et al. (2015), Kaneko et al. (2010)* | Rupture dynamics, seismic cycle |
| | Megathrust | Rubber-granular-silicone | *Rosenau et al. (2009, 2010), Rosenau and Oncken (2009)* | *Pipping et al. (2016), Kaneko et al. (2010)* | Seismotectonic evolution, seismic cycle, statistics, tsunamigenesis |
| | Megathrust | Foam-silicone | *Dominguez et al. (2015)* | *Pipping et al. (2016), Kaneko et al. (2010)* | Seismic cycle |
| | Thrust/normal fault | Foam | *Brune (1996), Brune and Anooshepoor (1999)* | *Shi et al. (1998, 2005), Oglesby et al. (1998, 2000a,b); Ma and Beroza (2000), Nielsen (1998)* | Rupture dynamics, ground motion |
| | Strike-slip | Foam | *Caniven et al. (2015), Brune and Anooshepoor (1998)* | *Ben-Zion and Rice (1997), Lapusta and Rice (2003), Tullis et al. (2012 a,b)* | Seismic cycle |

**Table 1 (continued):** Categories of analogue earthquake models (see text for discussion).





| Regime | Number | Equation | Meaning |
|---|---|---|---|
| **Quasi-static** | *Smoluchowski* | $Sm = \rho g l / \Delta p$ | Gravity-pressure relation |
| | *Ramberg* | $Ra = \rho g l^2 / \eta v$ | Gravity-viscosity relation |
| | *Stokes* | $St = \Delta p / \eta v$ | Pressure-viscosity relation |
| **Dynamic** | *Reynolds* | $Re = \rho v l / \eta$ | Inertia - viscosity relation |
| | *Froude* | $Fr = v / (g l)^{1/2}$ | Inertia-gravity relation |
| | *Cauchy* | $Ca = \rho v^2 / K$ | Inertia-elasticity |

**Table 2:** Dimensionless numbers used in analogue earthquake models (see text for discussion).





| Regime | Parameter | Symbol | Dimension | Model value | Prototype value | Scaling relation | Scaling factor |
|---|---|---|---|---|---|---|---|
| **Interseismic** (quasi-static regime) | Length | $l$ | L | mm-dm | 1-1000 km | $L^* = L_{model}/L_{prototype}$ | $10^{-5}$-$10^{-6}$ |
| | Time | $t$ | T | sec-hours | Years-Ma | $T^* = T_{model}/T_{prototype}$ | $10^{-10}$ |
| | Density | ρ | M/T³ | 1000-2000 kg/m³ | 2500-3000 kg/m³ | $\rho^* = \rho_{model}/\rho_{prototype}$ | $1/3…1$ |
| | Mass | $m$ | M | kg | Tera-Petatons | $M^* = \rho^* L^{*3}$ (for $g = 1$) | $10^{-15}$-$10^{-18}$ |
| | Gravity | $g$ | L/T² | 9.81 m/s² | 9.81 m/s² | $g_{model} = g_{prototype}$ | 1 |
| | Recurrence time | $Trec$ | T | sec | centuries | $T^* = T_{model}/T_{prototype}$ | $10^{-10}$ |
| | Maxwell relaxation time | $Tm$ | T | 0.1 sec | decades | $T^* = T_{model}/T_{prototype}$ | $10^{-10}$ |
| | Tectonic loading velocity | $vtec$ | L/T | mm/sec | mm/years | $V^* = L^*/T^*$ | $10^4$-$10^5$ |
| **Coseismic** (dynamic regime) | Length | $l$ | L | μm-mm | cm-m | $L^* = L_{model}/L_{prototype}$ | $10^{-5}$-$10^{-6}$ |
| | Time | $t$ | T | sec | Sec-min | $T^* = (L^*)^{1/2}$ | $10^{-3}$ |
| | Mass | $m$ | M | kg | Tera-Petatons | $M^* = L^{*3}$ (for $g = 1$) | $10^{-15}$-$10^{-18}$ |
| | Acceleration | $a$ | L/T² | 1 m/s² | 1 m/s² | $a_{model} = a_{prototype}$ | 1 |
| | EQ duration | $Teq$ | T | ms-sec | sec-min | $T^* = (L^*)^{1/2}$ | $10^{-3}$ |
| | Rupture velocity | $Vs$ | L/T | m/s | km/s | $V^* = T^* = (L^*)^{1/2}$ | $10^{-3}$ |
| | Slip velocity | $V$ | L/T | mm/s | m/s | $V^* = T^* = (L^*)^{1/2}$ | $10^{-3}$ |
| | Seismic moment | $M0$ | ML²/T² | 0.1-1 Nm | $10^{21}$-$10^{23}$ Nm | $M0^* = M^* L^*$ (for $g = 1$) | $10^{-21}$-$10^{-23}$ |
| | Moment magnitude | $Mw$ | - | -7- -5 | 8-9.5 | $M_{w\ prototype} = M_{w\ model} + 2/3\log(M_0^*)$ | n.a. |

**Table 3:** Typical scales, scaling relations and factors in analogue earthquake models (see text for discussion).




| Rheology | Material | Source | Rheological model | Constitutive parameters | | | |
|---|---|---|---|---|---|---|---|
| | | | | *E* (kPa) | ν | Vs (m/s) | ρ (kg/m³) |
| **Elastic** | Homalite-100 | *[1]* | linear elastic | 3860*10³ | 0.35 | 1200 | 1230 |
| | Polycarbonate | *[1]* | linear elastic | 2480*10³ | 0.38 | 960 | 1129 |
| | Foam | *[2]* | linear elastic | 10-100 | 0.1-0.3 | 10-100 | 40 |
| | Rubber (EPDM) | *[3]* | linear elastic | 100-10.000 | 0.5 | 1-100 | *1600* |
| | Gelatine (<10%) | *[4][5]* | linear elastic | 1-10 | 0.45-0.5 | 0.5-2 | *1000-1100* |
| | | | | *C* (Pa) | μ | a-b | ρ (kg/m³) |
| **Frictional-plastic** | Rice | *[6]* | high friction, velocity-weakening | 10-100 | 0.7 | -0.015 | 900 |
| | Sugar | *[6]* | high friction, velocity-strenghtening | 10-100 | 0.7 | +0.015 | 900 |
| | Foam (coated) | *[2]* | high friction, velocity-weakening | *not specified* | 0.65 | -0.017 | 40 |
| | Gelatine-on-sandpaper | *[7]* | low friction, velocity-weakening | *not specified* | 0.15 | -0.028 | 1000 |
| | Gelatine-on-plastic | *[7]* | low friction, velocity-strenghtening | *not specified* | 0.05 | +0.027 | 1000 |
| | | | | η (kPas) | *n* | *Tm* (s) | ρ (kg/m³) |
| **Viscoelastic** | Silicone oil (PDMS) | *[8]* | Maxwell | 30 | 1-2 | 0.1-0.2 | 970 |
| | Silicone-plasticine (4:1) | *[9]* | Maxwell | 6.000*10³ | 2.8 | 30 | 1000 |
| | Gelatine (<10%) | *[5][10]* | Maxwell (>3%)/ bi-viscous (<3%) | 0.05-10 | 5 | 0.1-1 | 1000-1100 |
| | Natrosol HH | *[11]* | Burgers | 0.02-3 | 1-2 | 1-12 | 1000-1010 |
| | Carbopol® | *[12]* | Herschel-Buckley | 0.01-10 | 1.6-3.4 | 0.1 | 1000-1030 |
| | Kaolin (wet) | *[13]* | Burgers | 1-10*10³ | *not specified* | 100-1000 | 1600-1700 |

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

**Table 4:** Mechanical properties of selected rock analogue materials at typical laboratory conditions (see text for discussion).





| Category | Technique | Observables | Indirect observable | Precision | Spatial resolution | Temporal resolution |
|---|---|---|---|---|---|---|
| **Local** | Strain gage | strain | rupture kinematics | medium | low | 100 Hz |
| | Velocity transducer | velocity | rupture kinematics | high | low | kHz |
| | Accelerometer | acceleration | rupture kinematics | high | low | kHz |
| | Microphone | acceleration | rupture kinematics | medium | low | kHZ |
| | Pressure gauge | pressure | absolute stress, stress drop | medium | low | kHz |
| | Photocell | velocity | rupture kinematics | medium | low | kHz |
| | Acoustic emission | acceleration | rupture kinematics | high | low | $10^2$ kHz |
| | Laser velocimetry | velocity | rupture kinematics | high | low | $10^9$ kHz |
| **Regional** | Photoelasticity | stress | strain, rupture kinematics | medium | ca. 100 px² | kHz |
| | Digital Image Correlation | velocity, strain rate | stress | $10^{-3}$ px | ca. 10 px² | kHz |
| **Global** | Strip chart recorder | displacement | fault slip | low | low | continuous |
| | Odometer | displacement | fault slip | micrometre | low | kHz |
| | Force sensor | force, stress | absolute stress, stress drop | mN | low | $10^2$ kHz |

**Table 5**: Monitoring techniques used in analogue earthquake models (see text for dicsussion).

