# Peer review of "Analogue earthquakes and seismic cycles: Experimental modelling across timescales"

_Solid Earth, 2016_

## Referee Comment (RC1) · M. Cooke (Referee) · 6 Jan 2017

M. Cooke (referee)
cooke@geo.umass.edu

**General Comments:**

This discussion paper summarizes research on analog modeling of earthquakes and seismic cycles and provides a comprehensive resource for workers in this field. The paper nicely classifies and describes various experimental approaches, materials used and observational methods.  In addition to covering a wide range of experimental results, the paper provides useful reviews of the fundamentals of scaling, rheology, fault friction, and earthquake statistics in order to provide context for the analog studies. I learned much from reading the paper and appreciate the authors' vast and deep knowledge of the subject.  This paper serves as an excellent reference for experimentalists interested in earthquake and seismic cycle processes. I offer suggestions for the authors to take opportunity to improve the paper's utility.

**Specific Comments:**

A. The third classification of experiments as 'scale' models might more accurately called scaled crustal models.  Some fault block models are scaled and so the term scale models doesn't fully describe the models of this classification. I recommend replacing with the term 'scaled crustal models'.

B. Page 2, line 3; Why not consider all earthquakes? While the scaled crustal models primarily pertain to tectonic earthquakes, the spring slider and fault block model certainly can be applied to any earthquake, regardless of setting.  The mechanisms presented in the paper are broad enough to consider all earthquakes.

C.  The paper misses an opportunity to promote the benefit of experimental results over numerical simulations.  The discussion mentions that numerical simulations can be used to understand the experimental results and a reader could be left with the impression that one could dispense with experiments entirely and go right to numerical models.   Should we not bother with the challenge of scaling the crustal experiments to both short and long term processes and just develop numerical models?

D.  The review is comprehensive but its utility could be improved by additional presentation of particular configurations, rheology etc that would be adept at capturing particular processes.  For example, in the scaling discussion, the appropriate scaling to use in the model depends on the questions of interest. If you are more interested in the directivity and details of rupture evolution you will

probably scale the experiment differently then if you are interested in the statistics of thousands of rupture events. Examples of this could be very helpful.

E. The implementation of equation (4) is based on the assumption that the Coulomb failure dominates within the brittle regime.  Since it is the creation of new faults and not the sliding along existing faults that is the process of interest, the scaled parameter should be inherent shear strength rather than cohesion, which describes the strength of existing faults.   For dry sand, this distinction is blurry because sand has many surface, grain boundaries, along which to slide and there is no explicit material failure.  For this reason, many people have used the term cohesion for scaling of experiments but since this discussion paper goes beyond dry sand, the formulation should be clear that the assumption is failure strength of the material. This also applies to section 4.2.1 on Mohr-Coulomb plasticity. The parameter of interest there should also be inherent shear strength, So, which for dry sand happens to be the same as cohesion.

F. Please explain why the characteristic length scale for the quasi-static model should be peak slip. The peak slip may a consequence of dynamic processes in the model. If the dynamics aren't properly scaled then the peak slip might not scale regardless of whether the quasi-static regime is properly scaled. For some models the more appropriate length scale for the quasi-static regime would be thickness of the brittle material, which should scale to locking depth of the crustal system.

G. For the dynamic regime, the scaling of Dc, slip-weakening distance, should be discussed. If the Dc of the material is artificially high, this can change the nature of dynamic rupture.  This parameter is challenging to scale within numerical models and should be addressed within the scaling of the dynamic regime. Dc is mentioned in equation 18 but its scaling is not discussed.

H. The discussion of the incompatibility of scaling the Froude and Reynold's numbers is very interesting and relates to the issue that the most important aspect of scaling is to ensure that you scale the processes important for the questions asked. Some scaled models may aim to capture all of the processes acting within the crust but many very useful models will investigate a subset of processes.  For most models it will be critical to scale some but maybe not all of the crustal processes. The scaling section of this paper should make note of the importance of matching your scaling to the processes of investigation.

I. The text on rate and state friction within section 4.2.1 overly relies on the textbook of Scholz.  As a review paper, this manuscript should cite additional resource. Even Marone, 1998, which is cited in the figure caption, does not appear in the text.

J. The end of section 5.2 .2 mentions that numerical models can deepen our understanding of the experimental strain fields. This doesn't really belong in the image correlation techniques section. Numerical simulations of experiments have

*Solid Earth Discussions doi:10.50194/se-2016-165-RC1*

great potential and could warrant a separate sub section within the techniques or applications sections.

K. The paper misses an opportunity in the discussion of seismic versus aseismic faulting to highlight the simple block slider experiments that are used in teaching classrooms. These simple brick on sandpaper (or variations) demonstrations very effectively convey the concepts of stick slip to students and the public.  Some apparati include accelerometers, acoustic emissions, force data etc for students to analyze various earthquake properties.

L.  There is a rich literature of stick slip experiments with glass beads that is not included in this review (e.g. Savage and Marone 2007 JGR). Since glass beads are analogs for rock they should probably be considered within this review paper.

M. Section 6.4 may need a different title or become more broad in scope. For many researchers, site effects include the very local effect of the geotechnical layer on ground shaking.  For example, some civil engineers care only about the attenuation and amplification within the geotechnical layer.

**Technical Corrections:**

- Page 1 line 11 kept developing -> developed
- Several occurrences of 'which' should be 'that'.  Page 1, line 14; Page 6 line 12; Page 7 line 23; Page 8 Line 1; Page 11, Line 23;
- When using 'which', it should always be part of a phrase that is bracketed by commas. First comma before 'which' and second comma at the end of the phrase (unless the phrase ends the sentence).
- Page 1 line 23 The San Francisco earthquake was in 1906 (revise throughout paper)
- Page 1 Line 28: flouring -> flourishing (probably there is a better word to us)
- Page 1 Line 30: Add 'e.g.' before reference
- Page 2 Line 3: anthropogenic pumping can also produce earthquakes
- Page 2 Line 11: if -> while
- Page 2 Line 19-20: …mainly utilizes analytical and numerical modeling methods in order to constrain complementary…
- Page 3 Line 28-29: awkward sentence needs rewriting
- Page 4 Line 20-28: Since the Reber system is a deformable slider-spring, maybe it should be presented after the paragraph that starts at line 23 about the rigidity of the classic spring-slider system.
- Page 5 Line 19: …strain monitoring, such as digital image correlation (e.g. Adam et al., 2005), it became…
- Page 5 Line 25: awkward sentence needs rewriting
- Page 5 Line 28: add comma after models,

- Page 7 Line 16-17: sentence has awkward structure.. maybe -> challenge to conduct.. and to record deformation..
- Page 7 Line 17: Either -> If
- Page 7 Line 21: Do not start a sentence with 'however'. This is a linking word to be used after a semi colon
- Page 7 Line 21: comma after view,
- Page 7 Line 24: comma after regime,
- Page 7 Line 25: comma after Consequently,
- Page 7 Line 29: comma after system,
- Page 7 Line 29: add 'that' after suggest
- Page 7 Line 29: comma after velocity,
- Page 8 Line 3: Remove 'In the following'
- Equation 3. The Ramberg number doesn't seem to be utilized in this paper. Please explain its utility.
- Page 8 Line 12: remove '
- Page 8 Line 13: comma after ),
- Page 8 Line 24: comma after model,
- Superscripts are not formatted correctly: Page 8, lines 2 and 3
- Page 9 Line 22: does play no -> plays limited
- Page 9 Line 23: remove 'in'
- Page 9, Line 24, commas before which and after deformation,
- Page 9 Line 25: comma after above,
- Page 9 Line 25: the rate and state parameter $D_c$ has dimension and should be scaled.
- Page 10 Line 9: what is s.s?
- Page 10 Line 9: I may have missed something but I didn't see where Mo* was defined.
- Page 10 Line 23: inherent -> exhibit
- Half-space should be hyphenated Page 11 lines 20 24, 30and elsewhere
- End of section 4.1.1. You could point out that in the case where DL equals Ds the sum of the interseismic deformation and the coseismic deformation produce a set function of the tectonic velocity.
- Page 12 Line 13: Photoelasticity
- Page 12 Line 14: remove 'accordingly'
- Page 12 Line 24: constraint -> constrained
- Page 12 Line 31: add undesired before reflection
- Page 13 Line 7-9: Awkward sentences
- Page 13 Line 14: pressures is plural in one instance and singular in the other
- Page 13 Line 15: remove first 'deformation'
- Page 13 Line 17 commas after is, and case, (to bracket the phrase)
- Page 14 Line 1: Researchers are still using both static and dynamic friction as well as the a-b values for rate and state friction. It really isn't a matter of one or the other, both sets of parameters are involved in the empirical formulations, which is evident in equations 19-21.

- Page 14 Line 27-28 Consequently, earthquakes nucleate only in the unstable regime but can propagate…
- Page 15 Line 22: The latter -> This apparatus
- Page 15 Line 23: 'Accordingly' doesn't seem right here
- Page 15 Line 29: add comma after strain rate,
- Page 16 Line 6 and 7: add e.g. before citations
- Page 17 Line 1:  Please explain why the Maxwell model is considered more relevant than the Kelvin-Voigt model.
- Page 17 Line 10: add 'has' after It
- Page 17 Line 13: add e.g. before citation
- Page 18 Line 15:  Helst -> Elst
- Page 19 Line 8: in-line hydraulic pressure gauge needs a citation
- Page 19 Line 23: Can Tchalenko, 1970 GSA Bull, v 81 pm 162501640. A very cool paper that like Niewland et al. (2000) shows the pressure changes associated with faulting.
- Page 19 Line 27:  Paul Young has also done a lot with AE for precursory failure.
- Page 20 Lines 3-6: What are the drawbacks and benefits of laser techqniues?
- Page 20 Line 10: Karen Daniels has some very nice granular experiments using photoelastic materials that should be included.
- Page 20 Line 14: awkward sentence
- Page 21 Line 13: PIVLAB is another used by some groups
- Page 21 Line 5: This mentioned of stresses from constitutive laws is out  of place as it is an application rather than an experimental technique.
- Page 21 Line 26: Add references to stick slip experiments of Reber et al 2014 GRL and Reber et al 2015.
- Page 22 Line 1: iconographic -> iconic
- Page 22 Line 7: different spelling of Burridge Burrdige
- Page 22 Line 7-8:  remove 'as well'
- Page 22 Line 25: unclear sentence
- Page 22 Line 26: Numerical earthquake simulators often show stress transfer between  patches (see Tullis et al 2012 SRL).
- Page 23 Line 10: … distribution of slip events in…
- Page 23 Line 18: in -> is
- Page 23 Line 23: The danger of faults
- Page 24 Line 16: Add Rosakis references here.
- Page 27 Line 19-20: Can add citation to Kaj Johnson 2013 JGR
- Page 27 Line 29: constituted -> consists of
- Page 28 Line 18: experiment
- Page 28 Line 18: remove semi colon
- Page 28 Line 27: thereby -> therefor

---

## Referee Comment (RC2) · Anonymous Referee #2 · 13 Jan 2017

GENERAL COMMENTS

This manuscript provides a review mainly of techniques used in the lab for studying earthquakes via analogue modelling. As someone who is not directly involved in that branch of research, I found it difficult to muster much interest during most of the paper. I am curious about the topic, but I find the review too inwardly focused and disorganized. Many laboratory studies are mentioned, but the review only lists the technical aspects of these studies, ignoring (except in section 6) the insight they (hopefully) provided. To me, the best reviews summarize that insight so that not only the people directly involved in this line of research but also researchers in ancillary fields learn something by reading it.

Perhaps the key issue I have with the paper is lack of focus. The title promises a review of experimental modeling "across timescales". I would have expected the paper to address what these time scales are and how the analogue models inform our knowledge of them. I am left wanting. Instead, Section 1 and 2 summarize many of the approaches available to analogue modelling, section 3 introduces scaling, section 4 summarizes rheologies, and section 5 monitoring techniques. All these sections are useful to learn how to build a model, but what do they tell us about the timescales of earthquakes and seismic cycles? They give the tools, but no the results. Section 6 is the only one where results are summarized. The abstract mentions a review of "cornerstones" of development, which actually does describe the content OK (except we don't know why each study mentioned is important, as much of the paper is a just a list of works) but if that's the motivation for the paper, the title is misleading.

Section 6 is the closest to what I was expecting in this review. Specific studies are mentioned. In a few cases, details of the setup are included, but, importantly, the results are mentioned. There is still vagueness and room for improvement, though. For example, Section 6.5 concludes the description of the Caniven et al. (2015) study with "The model results compare to numerical simulations of strike-slip fault earthquakes", leaving me wondering how they compared (well, I assume, but I can't be sure), to what specific aspects of these simulation the experiments can be compared, and, crucially, what new insight has been gathered from the experiment. I suppose it's not just a confirmation to earlier studies, but there is no way to tell, based solely on this manuscript. Similar vagueness pervades the paper (e.g. Page 6, line 25).

Finally, I found the paper difficult to read due to imperfections of language. It needs to be thoroughly edited. This may be a stylistic choice, but the authors seem to avoid commas at all cost and that makes many sentences long and confusing. On the other hand, they love "i.e." and "e.g." whereas I find it best to avoid abbreviations. I find many twisted sentences that, although possibly not incorrect from a grammatical standpoint, are certainly not the clearest way to present the information. In writing, as in modeling, simplicity leads to clarity.

SPECIFIC COMMENTS

1) The abstract promises to discuss "limits, challenges and links to numerical models". I don't see that in the paper, expect for the occasional statement interspaced with general presentations. It's certainly not a focus of the paper. The stated focus on "scale models that are directly comparable to observational data on short and long timescales" is lost in the more general and occasionally very basic sections on modeling in general, rheology, and monitoring, which are imperfectly linked to observations and models.

2) The introduction introduces the "issue" that the time constraint of the earthquake cycle is unknown. How then can it be argued that the analogue systems are properly scaled? Doing this requires that we know and understand the relation between the various timescales (e.g. nucleation stage, repeat time, postseismic duration). The paper doesn't make the point that these relations are well understood, quite on the contrary, whether in the lab or in nature.

3) The distinction between "fault block" models and "scale" models seems arbitrary to me: fault block models are scaled as well as the "scale" models, even though the scaling may not always be rigorous. The schematics of Figure 1 imply that fault block models are in a strike-slip configuration with elastic blocks only whereas the scale models would be in a thrust configuration with both elastic and viscous layers. I don't see why one would be scaled and not the other. Elastic moduli and friction properties are relevant in the all the models. Several of the scale models of Figure 2 do not have the kind of layering in Figure 1c and one is not in a thrust setting. I agree that there is likely a difference in the rigor of scaling between the models built recently in the authors' labs and earlier efforts, but I don't see them as forming an entirely new category of models. If the classification is based on the complexity of the loading system (rigid blocks, elastic blocks, visco-elasto-plastic blocks) then the name of the proposed categories is misleading.

4) Section 2 is essentially a list of works. It shows that people do different things but doesn't explain why these different approaches were adopted (why this material vs. that material), what problem or question new developments are trying to address, and what we learned as models grew in sophistication. It is as if an architect was describing a monument by listing stones that were used without telling us why there is such a variety of material and what the final building looks like.

5) I'm amazed that there is summary of what controls frictional sliding more recent than Brace (1972) (page 5, line 9). As much as I like that paper and respect its historical value, it might be good to mention some of the developments from the last 45 years... Later on (page 5, line 13), you mention you want to focus on studies using analogue rock materials instead of rock samples. Why? And where are the results? You mention "a large body of work" twice in that paragraph, but only list them. What did they see? What is the key point of these papers if the context of the present review? By the way, I don't see why this discussion belongs in the section on spring-slider models, as the loading machine acts as a deformable loading block (fault-block model). Also at the end of section 2.2: what did all these studies using blocks of different materials see?

6) The issue that a rigid slider distribute stress uniformly (Page 5, line 23-25) is exactly why people developed models with a network of springs (King 1994, Heslot et al., 1994, which were mentioned earlier in the section). What did they see? What did they learn about earthquakes from these models?

7) Page 6, line 6-17: why is a rigid plate appropriate to model the slab? Slabs are also elastic. Even though the wedge above the slab is generally softer than the slab due to its elevated temperature, thermal conduction implies that there is no actual temperature jump across the subduction interface. Therefore, the footwall is as deformable as the head wall at least over some length scale.

8) I don't understand the analogy from adaptive time scaling in experiments and adaptive time stepping as a numerical method. The numerical strategy involves changing

time step so that the solution becomes more stable or accurate. The solution itself is modified. The physical time scales, lengths scales, and other scales of the modelled system are not modified. The adaptive time scaling does the opposite: the solution is unchanged, but different scales are used when extrapolating different phenomena to natural conditions. This analogy baffles me.

9) The section on scaling is important and starts to address the issue of multiple time scales. Once again, unfortunately, it mostly states what is done without presenting many results. In addition, there is quite a bit of confusion there as some quantities are either incompletely defined, or substitutions occurs without justification. I feel the authors could do a better job linking the non-dimensional numbers and the scaling relations. To start with, in Equation 2, please define the measure or component of stress that are you using. Is it an invariant, a shear stress, or a normal stress? The words in the equation say "pressure force", which is weird, as pressure is a force (over unit area) and isotropic, whereas I suspect that shear stress is used here. Neither rho*g or sigma/l have units of force. (same issue in equation 3). The definition of Sm in Table 2 is different from Eq. 2, and that table includes a Stokes number whose importance was not discussed in the text. I am confused, in line 15-16, how Sm and Ra can dictate, among other things, length scaling, when the statement is "for a given length scale". Note also that no brittle scaling has been defined. What if the model is not viscous? Can there not be a number equivalent to Ra but using, for example, inertial forces? I see Ra as a special case of Sm when the stress is controlled by viscous processes ($\sigma = \eta * v/l$). Why are they treated as different numbers? Why did you switch from v/l in Eq. 3 to $d\epsilon/dt$ in equation 5? Page 9, line 18: why restrict the scaling to "typically"? How other than with Re would you scale dynamical effects? The final paragraph of Section 3.2 belongs earlier, as that scaling is used in the analogy of moments at the top of page 10.

10) Section 4 on rheology is written as a level that doesn't help with the topic of the review. It is also not really "historical" as it doesn't describe how ideas and approaches

have changed over time, just a portion of current understanding. It would be appropriate for a textbook, but defining all the possible rheologies seems a waste of space. In addition, these definitions are not rigorous. For example, Hooke was referring in 1676 to "The power of any springy body is in the same proportion with the extension.", which gives F=kx. It is not equivalent to Eq. 12, which is a differential form that allows for residual strength or strain. The diagram of Fig. 5 shows non-linearity and possibility residual strength, and while this is more realistic than F=kx, it is also not Hooke's law. Also, Byerlee's law (Fig. 7a) is not a generic linear relation but refers to specific sets of parameters (those next to the line fits, but not in the label for $\mu = \Delta\sigma_n / \Delta\tau$) To save time and space, I will not give details of typos, unclear statements for this section (suffices it to say it needs as much editing work as sections 2 and 3) as I think it first needs to be reworked to focus on what is truly needed to understand the time scales of seismic phenomena in the lab. The section on slip models, for example, is entirely irrelevant. I do need to point out that unlike what is written at the top of P.17, I find that Burger's body is considered to be more relevant then either the Maxwell or Kelvin-Voigt models in recent studies and that the presence of multiple time scales of postseismic relaxation was seen in many studies long before Wang et al. (2012), e.g., Savage and Svarc (JGR, 1997), Nishimura et al. (Tectonophysics, 2000), Kenner et al. (JGR 2000) and others.

11) The Schultze ring shear apparatus plays a prominent role in the collection of the mechanical data presented. Yet its description is minimal (page 15, line 21-22). Please describe in more detail what that apparatus is and how it works. Maybe include a schematic of this apparatus?

12) Section 5 (monitoring techniques) reads like a long list of approaches. As before, I'd like the authors to maybe compare more explicitly what can be learned from using these techniques. Looking back at all these works, what would you recommend using to answer different questions?

13) I found section 6 to be much better written than the rest of the paper and more

useful, in that it details not just what was done, but also what was learned from these experiments. It finally explains something about seismic phenomena and reveals the usefulness of (a few of) the experiments mentioned earlier. There is room from improvement, though. For example, b-values are mentioned page 21 line 30, before the concept was introduced in page 22 line 7. In page 23, stick-slip is discussed line 18 but defined line 25. My other comments on this section are minor.

14) At the end of section 6.4, we are presented again with a technical aspect (how Brune and Anooshepoor excited their models) without being told what they learned in that study.

15) Check the references. Several are missing elements. A few are using all-caps for the journal. Sometimes, the first name appears first (A. Alshibi)

16) What exactly are the "Nature example" shown in Figure 14? Neither the caption nor the text give us this information.

TECHNICAL COMMENTS This list is not exhaustive, as I do not have the time nor the qualifications necessary to pick up every grammatical or stylistic issue with this paper.

Awkward expression, twisted sentences, etc.

_ Page 1, line 8: "joined the forefront of the research" is just weird...

_ Page 1, line 13: "We here review the cornerstones" ("here" should be after "review"; cornerstone implies that the study will change the direction of science but that case is never made explicit, except for Reid (1911)".

_ Page 1, lines 29-30: It's trivial that "seismogenic" faulting should be a mechanism for earthquakes. That's what the name means!

_ Page 2, line 3: "we focus here tectonic earthquake modelling" or "we focus here on the modelling of tectonic earthquakes".

_ Page 2, line 8: "which affect notably their relevance". Isn't that redundant with what

is above?

_ Page 2, Line 13: "which is". There's no clear subject in agreement with the singular "is".

_ Page 2, paragraph starting with "New technological advances". The topic of the paragraph changes from "new advances" to "the issue of a time constant". This kind of change must be avoided to produce clear writing.

_ Page 3, line 1: "we present here".

_ Page 3, line 9-10: "Here, we categorize...".

_ Page 4, Line 30: "to dilute". To circumvent?

_ Page 5, line 32: "allow to simulate". They make it possible to simulate (?) they allow the simulation (?)

_ Page 6, line 10, 11: "Laboratory scale, not "labscale". Also at page 7, line 10.

_ Page 7, line 4: one of the remarkable points OF THIS STUDY is the

_ Page 7, line 10: the ending of the sentence "especially if scale models are considered" is trivial: the whole sentence is about scaling, which can't be done if you are not considering a scale model!

_ Page 7, line 18: replace "then" by "in which case". Also in line 20.

_ Page 7, line 19: replace "Alternatively" by ", or" (to go with "Either" in line 17, or remove that "Either". Also, be more explicitly by what you mean with "(MHz)" (that's a unit of measurement, but you include it as a modifier for "techniques". It's not a technique).

_ Page 8, line 9, replace "and the Ramberg Number ..." by ". Second, the Ramberg Number...".

_ Page 9, line 22, at the end write "does not play a role" (play no role is too colloquial)

_ Page 9 line 26: "Here belongs" is very awkward (and should be plural anyway as several things are listed next).

_ Page 10, line 18: in English, no "live" in "everyday experience".

_ Page 18, line 29: I don't understand "In contrast regional tectonics surfaces"

_ Figure 4 caption: I suggest rewriting the beginning "Scaling of parameter values from laboratory (model) to nature (prototype). (a)..."

_ The caption of Figure 12 mentioned "creep" and "transient slip" but that nomenclature is different from what is in the figure itself.

Improper terms

_ Page 1, Line 12: "culmination"

_ Page 1, Line 20: "perceptible shaking". Not every earthquake leads to shaking that can be felt, which is the meaning of perceptible.

_ Page 4, line 31: "They allow investigating..." (use present tense as this hasn't changed).

_ Page 6. Line 14: "several", not "few" scale models (few implies that there should be more, and it's quite negative. I think you want to impress upon the reader that models actually exist, so "several" is more appropriate).

_ Page 7, line 21: delete "such" (dyadic has not been explained before)

_ Page 8, line 14: replace the ending with "typical of tectonic applications"

_ Page 9, line 15: Replace "coherent" with "consistent", missing word "with THE stress scale"

_ Page 10 line 23: "inherit" is wrong (there's no passing of the characteristic from one level to the next).

_ Page 22, line 8: "regularly" implies a recurring phenomenon, following a pattern, especially if it occurs at constant intervals. I don't think we know that is indeed what happens for "system-size" events, especially in nature, due to our very limited dataset.

_ Page 24, line 9: that first sentence is just odd. Maybe "Rupture dynamics, which includes the study of earthquake nucleating, the transition to dynamic rupture, and its arrest, has by far the broadest range of applications of the phenomena that can be studied by analogue experiments".

_ Page 28, line 27: therefore, not thereby

_ Page 28 line 32, also page 29, line 7: does "2d" stand for "2D" (two-dimensional)? If so, use a capital (as for 3D). Also, I don't understand why having a rigid conveyer plate makes the model 2D. Does that change the width of the model? _ Page 29, line 7: not sure if you mean "few percent", which means "not a lot of percent", or "a few percent".

_ Figure 4: The blink of an eye (not an eyeblink) is a duration, not length.

Typos

_ Page 1, line 28: replace "flouring" with "flourishing".

_ Page 1, line 28; Page 4 line 3: No apostrophe in "1960's" (it's not a possessive)

_ Page 4, line 4: "to reproduce stick slip instabilities".

_ Page 4, line 5: "designed TO BE as stiff... but compliant enough..."

_ Page 7, line 7: "Scaling laboratory scale observations to nature"

_ Page 7, line 11: no capital S in "This section".

_ Page 7, line 18: no capital in "millions".

_ Page 8, line 13: missing parenthesis before "Table 2".

_ For consistency, please decide whether to use "micron", "micrometer", or "micrometre". All three are used at different part of the text.

_ Page 8, line 25: scaleS (it's used as a verb here).

_ In general: please format the exponents in the powers of ten correctly as superscripts (e.g. missed in Page 9, lines 2-3). Use consistently italics for symbols (e.g. missed in Eqs. 7, 8) and subscripts when needed (missed in Eqs. 9, 10)

_ Page 9, line 30: fix spaces and hyphens "... derived straightforward from the scaling rules..."

_ Page 19, line 8: no space in "inline"

_ Page 19, line 27: add parentheses around 1991.

_ Page 19, line 28: delete "and" at the beginning of the line?

_ Page 20, line 27: Don't start sentence by "i.e.".

_ Page 22, line 9: Delete "they".

_ Page 22, line 13: "It is defined AS the standard deviation..."

_ Page 23, line 28: "THE deformable...".

_ Page 24, line 4-5 "... magnitude decreaseS with depth ..."

_ Page 25,line 19: no s in "increase".

_ Page 26, line 5: "... in THE viscoelastic models..."

_ Page 27, line13: no capital but two n in millennia.

_ Page 27, line15: no capital in Fig. 6b, c

_ Page 28, line 29: missing h in earthquakes

_ Page 29, line 31: "into which is embedded ..."
_ Page 29, line 17: no capital S in "In summary".

_ Page 30, line 1: present tense "include" (the materials have not finished including these rheologies).

_ Page 30, line 21: "the material presented here".

_ Acknowledgment: please specify the kind of data. It's too vague as it is.

_ Figure 8: label "Cacao", not "Kakao".

_ Figure 9 is referred to in the text (section 5.3) after Fig. 10 (Section 4.3.1).

_ Caption of Figure 10: no capital A in Maxwell.

_ Caption of Figure 13: "upwards and downwards in the seismogenic zone".

_ Whether references are in italics or not in table 1 is inconsistent.

_ Table 5: "gauge", not "gage"

---

## Author Comment (AC1) · 1 Mar 2017

**Comments of Reviewer 1 (Cooke)**
*Replies of Authors*
*Changes in manuscript*

**A. The third classification of experiments as 'scale' models might more accurately called scaled crustal models. Some fault block models are scaled and so the term scale models doesn't fully describe the models of this classification. I recommend replacing with the term scaled crustal models.**

*We used "scale model" as a standing term (e.g. in title of Hubbert's seminal paper), here applied to earthquakes and tectonics.*
*Fault block models are labscale but not "scaled" in the sense of Hubbert's scaling theory, i.e. showing geometric, kinematic and dynamic similarity. Fault block models do rarely show geometric similarity (instead of representing a certain structural setting at a given geometric scale they represent an arbitrarily oriented and scaled fault interface), typically there is no defined length scale and the deformation is highly exaggerated. Fault block models show no kinematic similarity (which would mean loading rate and rupture speed etc. scale down consistently) and consequently cannot, by definition, be considered dynamically similar. Observations from those models cannot be directly scaled up to nature but only extrapolated qualitatively.*
*This is why we suggest to "seismotectonic scale model" to better describe these models. This also better describes the multiscale nature of the approaches.*

*We changed the name of the third category to "seismotectonic scale models" in order to emphasize the multiscale nature and more quantitative applicability of these models.*

*We changed the text in Sect. 2 to make clear the difference also in terms of scaling:*

*"Here, we categorize analogue earthquake models into three groups with decreasing level of abstraction and applicability (Fig. 1, Table 1): (1) "Spring-slider models" in which elastic and frictional elements are physically discrete components of the setup (Sect. 2.1). **These models can only be applied conceptually to nature**. (2) "Fault block models" in which two elastic blocks, with similar or different elastic properties, are in frictional contacts (Sect. 2.2). **Observations from these models can be qualitatively extrapolated to nature**. (3) "Seismotectonic scale models" in which a distinct tectonic setting is realistically simulated at small scale and with boundary conditions mimicking as closely as possible the natural prototype (Sect. 2.3). **These models can be directly and often quantitatively upscaled to nature**."*

**B. Page 2, line 3; Why not consider all earthquakes? While the scaled crustal models primarily pertain to tectonic earthquakes, the spring slider and fault block model certainly can be applied to any earthquake, regardless of setting. The mechanisms presented in the paper are broad enough to consider all earthquakes.**

*Earthquakes are the result of a variety of source mechanism, only some of which involve the here considered Mode II crack-like displacement in rocks (i.e. tectonic earthquakes). Many non-tectonic earthquakes involve very different kinematics (e.g. hydrofractures = Mode I crack, landslides = non-double couple, explosions = Mogi source) or mechanisms (e.g. resonance in volcanic conduits). We therefore don't think that we can consider all earthquakes.*

*No changes in manuscript*

**C. The paper misses an opportunity to promote the benefit of experimental results over numerical simulations. The discussion mentions that numerical simulations can be used to understand the experimental results and a reader could be left with the impression that one could dispense with experiments entirely and go right to numerical models. Should we not bother with the challenge of scaling the crustal experiments to both short and long term processes and just develop numerical models?**

*We hope this impression is wrong! We agree that there are pros and cons but also benefits on both sides.*

*Numerical models are an approach to describe nature mathematically and since the physical laws are not known (at best described by simplistic models or empirical laws) any simulation is a somehow arbitrary and possibly biased solution of a highly non-unique problem. The fact that numerical models replicate natural observations does not mean they are a proper physical description. Given the many parameters to be tuned, any result can be obtained by a numerical model. Experiments on the other side are physically correct in itself; however, extrapolation to nature might not always be valid. Additionally, not all quantities necessary to infer the physical laws from experiments might be observable directly but they might be inferred from the experiment using numerical models. Finally, numerical models can implement more complex scenarios, be it in terms of geometry or rheology, and a wider parameter spaces. However, computational limitations often limit the spatial and temporal resolution or reduce the model to 2D. We think it is here were analogue and numerical models should complement, rather than compete, with each other.*

*Experiments suffer from variability while numerical models always give the "right" answers; however, precision should not be confused with accuracy. In fact, variability is present in nature as well and we might learn a lot from analyzing it in the models. Experiments are carried out in a space-time continuum, while numerical models are discretized. Especially in terms of cross-scale modelling, this may be a source of bias.*

*However, since the paper does not explicitly deal with numerical models we feel uncomfortable with adding this general debate to the manuscript. Also we do not know if this paper is right place to discuss this at length. We hope the current version advertises experiments enough and leaves it to the reader to judge whether experiments are needed or not.*

*We added however some discussion on this in the intro & conclusions and more details on the comparisons mentioned to illustrate the interplay between experiment and simulation:*

*Intro:*
*"Deformation of the earth involves timescales ranging from nanoseconds (processes at the rupture tip) to hundreds of Millions of years (Wilson cycle) and all spatial scales from atoms to the earth itself (e.g. Ben-Zion, 2008). Such a multiscale process pose major challenges to observation in nature as the instrumental and historical records are too short to capture a significant amount of the evolution. Simulation is a way to overcome such limitations. However, our knowledge of the physics of earthquakes and earth deformation in general is incomplete and does not allow to setup realistic numerical scenarios. Non-uniqueness of numerical solutions for typical problems of earth deformation at various time scales (e.g. inversion of rupture kinematics or mantle rheology from co- and postseismic observations, respectively) is another limitation of computer models. Experimental*

approaches have been traditionally used to address physical problems like earthquakes and seismic cycles and, more recently, bridging the gap between short-term instrumental/historical observations and long-term paleoseismological/geological observations."

Ch. 2.3:
"The analogue models of Rosenau et al. (2009, 2010) have been cross-validated using finite element models (Pipping et al., 2016;).The numerical model of Pipping et al. (2016) replicated the laboratory results by means of the general deformation pattern of the wedge through seismic cycles, the recurrence behaviour (recurrence times and periodicity) as well as principal source parameters (slip distribution). Using numerical simulation, the frictional properties of the analogue model material were validated and augmented if not measured physically. Moreover the numerical model provided a highly-resolved image of the rupture dynamics beyond what was observed in the analogue model (see e.g. animation of analogue vs. numerical subduction earthquake cycles in Rosenau et al., 2016). Vice versa, the laboratory example served as an object for testing the general performance of the numerical modelling scheme and for the verification of the effectiveness and reliability of the algorithm for frictional contact modelling introduced there."

"The analogue models of Corbi et al. (2013) have been tested against numerical models using the finite element technique (van Dinther et al., (2013a, b). An extensive benchmark has been carried out analysing a set of parameters that are characteristic of both the interseismic and coseismic stages (i.e., recurrence time, coseismic displacement, rupture duration, rupture velocity, slip rate, rupture width and hypocentre location). A robust fit has been obtained for the majority of the investigated parameters of the reference model (van Dinther et al., 2013a). In particular, the mean of individual source parameters of the numerical model fall systematically within the standard deviation of the same parameter of the analogue models. The largest discrepancies between the two modelling techniques are observed for rupture width and hypocentre location and are attributed to difference in boundary conditions (i.e., the aseismic part of the megathrust doesn't create stress build up in the analogue models) and sampling rate, respectively."

Conclusions:
"We presented an overview of experimental approaches to model earthquakes,seismic cycles and seismotectonic deformation. The processes involved are multiscale posing the challenge to cross time scales from seconds (seismic deformation) to Millions of years (tectonic deformations) both in natural observation as in simulation and experiment. Since natural observations are intrinsically limited in resolution and period of observation, simulation by means of analogue and numerical modelling are key to understanding multiscale processes. An experimental approach to multiscale problems seems most natural because experiments happen in a time and space continuum in contrast to numerical models which need to be discretized."

(…)

"3. Coupling of analogue models with numerical models help to overcome the respective limitations and better exploitation of the respective potentials. For example, numerical models can be used to infer quantities from the experiment that are not directly observable (e.g. details of rupture propagation). On the other hand, experiments can help in validating numerical models by means of testing their predictions. Cross-validation and benchmarking in general should be promoted in the respective communities.

*4. Properly scaled analogue earthquake models may help to improve seismological and geodetic inversion techniques and overcome non-uniqueness of numerical solutions. They provide a large number of well constrained and self-consistent "case studies" which display both natural complexity and variability. Analogue earthquakes may thus serve to minimize the solution space and more adequately constrain e.g. slip variability."*

**D. The review is comprehensive but its utility could be improved by additional presentation of particular configurations, rheology etc that would be adept at capturing particular processes. For example, in the scaling discussion, the appropriate scaling to use in the model depends on the questions of interest. If you are more interested in the directivity and details of rupture evolution you will probably scale the experiment differently then if you are interested in the statistics of thousands of rupture events. Examples of this could be very helpful.**

*We agree that it would be desirable to have more concrete examples and descriptions as in similar reviews (e.g. Dooley and Schreurs, 2012; Graveleau et al, 2012). However, it became clear in an early stage of writing that we cannot provide such granularity in the description of particular setups and results. This is not only because of the large number of papers but also the variety of research questions and approaches. We therefore decided to generalize as much as possible leaving space for individual application of the material presented to the reader.*
*At the same time we provide a rather long reference list of applications to be exploited by the reader once he/she found his/her focus (see also Table 1).*

*Examples of adaptation of scaling to the research question (short vs. long term) are given in current version e.g. by comparing the "long term models" of Rosenau et al. (Chapter 6.1 and 6.5) to the "short-term" models of Corbi et al. experiment on rupture dynamics (chapter 6.3). We consider this sufficient.*

*We added figure 2 and 3 on particular setups; also we describe the ring shear tester used here to generate new data:*
*We describe the setup now in chapter 2.1 and show it in Figure 2 as an example of spring-slider setup.*
*We added:*
*"Several studies which focus in frictional behaviour of granular rock analogue materials (e.g. sand, glassbeads) at low loads (kPa) used a Schulze ring-shear tester (Schulze, 1994, Ritter et., 2016; Klinkmüller et al., 2016; Panien et al., 2006; Lohrmann et al. 2003) which serves here as an example of spring-slider device used to generate analogue earthquakes (Fig. A1). The ring-shear tester consists of a 4 cm high annular shear-cell made of stainless steel holding approximately 0.1 and 1 liter of the sample material. A ring-shaped-lid is placed onto the filled cell. The lid is subjected to a normal force in order to control normal load on the sample. While the cell is rotated, the lid is prevented from rotation by two tie rods connected to a crossbeam. The force necessary to shear the material is measured continuously. To ensure shearing inside the material and prevent slip between the lid and the granular material, the lid has 20 vanes protruding 4 mm into the material.The loading system is compliant enough (~1.3 kN/mm) to generate sticks-slip in a variety of materials at loads below 20 kPa. Results of this setup are presented on several occasions in this paper."*

**E. The implementation of equation (4) is based on the assumption that the Coulomb failure dominates within the brittle regime. Since it is the creation of new faults and not the sliding along existing faults that is the process of interest, the scaled**

parameter should be inherent shear strength rather than cohesion, which describes the strength of existing faults. For dry sand, this distinction is blurry because sand has many surface, grain boundaries, along which to slide and there is no explicit material failure. For this reason, many people have used the term cohesion for scaling of experiments but since this discussion paper goes beyond dry sand, the formulation should be clear that the assumption is failure strength of the material. This also applies to section 4.2.1 on Mohr-Coulomb plasticity. The parameter of interest there should also be inherent shear strength, So, which for dry sand happens to be the same as cohesion.

*The long term scaling as shown here is adopted from sandbox studies. This is why the Smoluchowski Number is defined using the Cohesion here. But it is true that all quantities which share the unit of stress can be used here (including all strengths).*

*Shear strength in the brittle regime is pressure dependent while it is strain rate dependent in the viscous regime. So "inherent shear strength" is not a fixed parameter of our models. In the brittle regime it can only be defined by giving a cohesion (shear strength at zero load) and friction coefficient ((describing the pressure dependency of shear strength) according to the Mohr Coulomb theory.*

*Strain localization in sand follows a qualitatively and quantitatively very similar stress-strain curve compared to brittle failure of intact rock. And it shows Mohr Coulomb failure envelopes characterized by cohesive and pressure-dependent shear strength. While the micromechanics are different, the bulk behavior mimics that of the prototype rather well. This is why we view cohesion in combination with the friction coefficient is the most useful set of parameters to describe brittle deformation, be it frictional failure or sliding. Both failure and sliding can be adequately described using the graphical model known as Mohr Circles.*
*We make these points clearer now.*

*Page 9, Line 18:*
*"Note that all quantities with the unit of stress, in particular all strengths, share the same scaling and are substitutes for cohesion in Eq. (4)."*

*Page 14, Line 26:*
*"While the Mohr-Coulomb criterion originally describes frictional faulting of intact rock, the same graphical method can be applied to describe frictional sliding on pre-existing faults (Byerlee's Law)."*

**F. Please explain why the characteristic length scale for the quasi-static model should be peak slip. The peak slip may a consequence of dynamic processes in the model. If the dynamics aren't properly scaled then the peak slip might not scale regardless of whether the quasi-static regime is properly scaled. For some models the more appropriate length scale for the quasi-static regime would be thickness of the brittle material, which should scale to locking depth of the crustal system.**

*We totally agree. We used peak slip here to illustrate the challenge in observing analogue earthquakes. However, we see that this causes confusion in the context of static vs. dynamic scaling and because it might appear as an a-priori set parameter. But as you say, it is a result of the dynamics and thereby rather a verification criteria for proper dynamic scaling. In fact the length scale is set by the stress scale which in turn is related material*

*properties (cohesion, strength, density). And the length scale is the same in the static and dynamic scaling (which only changes time scale).*

*We deleted the sentence referring to peak slip in the text in order to avoid confusion.*

**G. For the dynamic regime, the scaling of Dc, slip-weakening distance, should be discussed. If the Dc of the material is artificially high, this can change the nature of dynamic rupture. This parameter is challenging to scale within numerical models and should be addressed within the scaling of the dynamic regime. Dc is mentioned in equation 18 but its scaling is not discussed.**

*We thank the reviewer pointing us to this issue. In fact Dc is one of the key material parameters to be chosen according to the length scaling derived in the scaling chapter. However, few constraints exist for scale models. We added what we know about it in chapter 4.2.2.:*

*Page 16, Line 25:*
*"The slip weakening distance Dc is strongly scale dependent: It is in the order of decimetre to meter for natural earthquake slip events and millimeters in rock mechanics experiments (e.g. Hirose and Shimamoto, 2005, and references therein). In studies using rock analogue materials Dc scales down to micrometres consistent with typical length scales derived from applying Eq. (4) (e.g. Mair and Marone, 1999; Rosenau et al., 2009)."*

**H. The discussion of the incompatibility of scaling the Froude and Reynold's numbers is very interesting and relates to the issue that the most important aspect of scaling is to ensure that you scale the processes important for the questions asked. Some scaled models may aim to capture all of the processes acting within the crust but many very useful models will investigate a subset of processes. For most models it will be critical to scale some but maybe not all of the crustal processes. The scaling section of this paper should make note of the importance of matching your scaling to the processes of investigation.**

*The Froude vs. Reynold scaling is a prominent example of the conflict that may arise if you want to "scale everything". While we are friend of rigorous scaling we acknowledge the limits which we illustrate with this example. We think our scaling approach is already quite limited in terms of the crustal processes (brittle and viscous deformation, slow vs. fast deformation) and sets a minimum number of constraints by means of dimensionless numbers. It becomes clear from this paragraph that a dynamic scaling in the viscous is not possible. However, because the earth behaves viscoelastic, i.e. elastic at the timescale of aan earthquake, this limitation might not be a problem.*

**I. The text on rate and state friction within section 4.2.1 overly relies on the textbook of Scholz. As a review paper, this manuscript should cite additional resource. Even Marone, 1998, which is cited in the figure caption, does not appear in the text.**

*RSF theory is rather well established and consensus arrived to the details of RSF that we report here. This is why we restricted ourselves to a short description of RSF theory showing a minimum set of useful equations. We consider citing textbooks (Scholz) and review papers (Marone 1998, Scholz, 1998) along with the original papers (Dieterich,*

*Ruina) as sufficient and least biased. A critical review on RSF is beyond the scope of the paper.*

*We added missing reference to Marone (1998) in the text.*

**K. The paper misses an opportunity in the discussion of seismic versus aseismic faulting to highlight the simple block slider experiments that are used in teaching classrooms. These simple brick on sandpaper (or variations) demonstrations very effectively convey the concepts of stick slip to students and the public. Some apparati include accelerometers, acoustic emissions, force data etc for students to analyze various earthquake properties.**

*We agree that spring slider setups are useful for teaching purposes, but we think they have been exploited scientifically rather to their end. That's why we only refer to them in a historical way.*
*However, we now dedicated a short paragraph in the conclusion on outreach applications.*

*In the conclusions:*
*"4. experimental techniques are a superb method for visualization and teaching complex processes. For example, simple spring-slider experiments equipped with force sensors and accelerometer are easy to realize and provide fascinating hands-on experience on earthquakes."*

**L. There is a rich literature of stick slip experiments with glass beads that is not included in this review (e.g. Savage and Marone 2007 JGR). Since glass beads are analogs for rock they should probably be considered within this review paper.**

*We consider the early papers from this group laying the foundation of glassbead-based analogue earthquake studies (Mair and Marone, 1999; Mair et al. 2002). The Savage and Marone papers (2007 and 2008) focus on dynamic triggering, a topic that we omitted for sake of respecting space limits. These are very interesting papers but dynamic triggering is not yet a topic studied by more than individual groups by means of analogue earthquake modelling. That's the reason why we cut those studies out.*

*No changes in manuscript.*

**M. Section 6.4 may need a different title or become more broad in scope. For many researchers, site effects include the very local effect of the geotechnical layer on ground shaking. For example, some civil engineers care only about the attenuation and amplification within the geotechnical layer.**
*We agree.*
*We changed it to "Ground motion".*
*.*

**Page 2 Line 3:**
**Anthropogenic pumping can also produce earthquakes**
*We agree.*
*We added it.*

**Equation 3.**

**The Ramberg Number doesn't seem to be utilized in this paper. Please explain its utility.**

*It is!*

*Eq. (5) is derived from it and it is used to set the time scale in viscous models. It is applied when discussing the postseismic relaxation in Caniven's model for example.*

*We clarified it in the text:*

*"In the viscous deformation regime, the ratio of gravitation and viscous flow strength is used and has been labelled the Ramberg number"*

*"Note that the Ramberg number has been derived from the Stokes number (Table 2), which characterizes more generally slow (small Reynolds numbers Re << 1, Table 2) flow typical oftectonic applications.To achieve similitude these numbers have to be the same in the model as in the prototype. For a given length scale (usually suitably chosen for handling the model in a lab), Sm and Ra dictate the stress scaling in the brittle and viscous regimes, respectively."*

**Page 9 Line 25:**
**The rate and state parameter Dc has dimension and should be scaled.**
*We agree, see "G" from above.*

**Page 10 Line 9:**
**I may have missed something but I didn't see where Mo\* was defined.**
*Shortly before: "…it is defined as the product of rigidity of the earth, mean coseismic slip and rupture area"*

**End of section 4.1.1.**
**You could point out that in the case where DL equals Ds the sum of the interseismic deformation and the coseismic deformation produce a set function of the tectonic velocity.**
*We agree!*
*New sentence:*
*"In the case where DL equals DS the sum of the interseismic deformation and the coseismic deformation produce a set function of the tectonic velocity."*

**Page 17 Line 1:**
**Please Explain why the Maxwell Model is considered more relevant than the Kelvin-Voigt model.**
*New sentence:*
*"This is because the earth shows, in case of an earthquake stress drop (that can be viewed as large scale creep test), the instantaneous coseismic and transient postseismic deformation characteristic of a slowly relaxing Maxwell body. A Kelvin-Voigt body would show no instantaneous elastic response to sudden loading but a transient."*

**Page 19 Line 23:**
**Can Tchalenko, 1970 GSA Bull V 81 Pm 162501640. A Very cool paper that like Niewland Et al. (2000) Shows the pressure changes associated with faulting.**
*True, but this is a "global" monitoring tool while Niewland is a "local" one.*

*We added Tchalenko and other similar in the Global monitoring techniques section:*

*"Several studies used force sensors to measure the force exerted by a backwall in sandbox experiments of strike-slip (Tchalenko, 1970) and thrusting (e.g., Cruz et al., 2010; Souloumiac et al., 2012; Herbert et al., 2015)."*

**Page 19 Line 27:**
**Paul Young Has also done a lot with AE For precursory failure.**
*True, he is second author of the three Thompson et al papers cited!*

**Page 20 Lines 3-6:**
**What Are the drawbacks and benefits of laser techqniues?**
*The reported high temporal resolution vs. the spatial limit. We reported this already.*

**Page 20 Line 10:**
**Karen Daniels Has some very nice granular experiments using photoelastic materials that should be included.**
*We agree.*
*Page 4:*
*"Shearing of acrylic or polymeric discs (e.g. Daniels and Hayman, 2008; Reber et al., 2014) provided insights into the dynamics of sticks-slip in granular media by means of 2D "see through" experiments."*

*Page 22:*
*"Based on the photoelastic effect, Daniels and Hayman (2008) visualized the dynamics of force chains in sheared granular media undergoing stick-slip."*

**Page 21 Line 13:**
**PIVLAB Is another used by some groups**
*We agree and added:*

*Page 23:*
*"PIVlab (Thielicke and Stamhuis, 2014)"*

**All other technical comments:**
*We agree.*
*We changed the manuscript accordingly.*

---

## Author Comment (AC2) · 1 Mar 2017

**Comments of Reviewer 2 (Anonymous)**
*Replies of Authors*
*Changes in manuscript*

**GENERAL COMMENTS**
**This manuscript provides a review mainly of techniques used in the lab for studying earthquakes via analogue modelling. As someone who is not directly involved in that branch of research, I found it difficult to muster much interest during most of the paper. I am curious about the topic, but I find the review too inwardly focused and disorganized. Many laboratory studies are mentioned, but the review only lists the technical aspects of these studies, ignoring (except in section 6) the insight they (hopefully) provided. To me, the best reviews summarize that insight so that not only the people directly involved in this line of research but also researchers in ancillary fields learn something by reading it.**
**Perhaps the key issue I have with the paper is lack of focus. The title promises a review of experimental modeling "across timescales". I would have expected the paper to address what these time scales are and how the analogue models inform our knowledge of them. I am left wanting. Instead, Section 1 and 2 summarize many of the approaches available to analogue modelling, section 3 introduces scaling, section 4 summarizes rheologies, and section 5 monitoring techniques. All these sections are useful to learn how to build a model, but what do they tell us about the timescales of earthquakes and seismic cycles? They give the tools, but no the results. Section 6 is the only one where results are summarized. The abstract mentions a review of "cornerstones" of development, which actually does describe the content OK (except we don't know why each study mentioned is important, as much of the paper is a just a list of works) but if that's the motivation for the paper, the title is misleading. Section 6 is the closest to what I was expecting in this review. Specific studies are mentioned. In a few cases, details of the setup are included, but, importantly, the results are mentioned.**

*We feel sorry for having called possibly wrong expectations. We indeed intended to provide a review on the modelling tools rather than on the modelling results. The latter is just beyond a single paper as the field of investigations, even when focused on experimental approaches, is very wide and includes various communities (seismology, geodesy, geology) because of the multiscale nature of the process. We here tried (1) to bridge those communities and (2) give an overview of existing approaches. This results in a paper structure which necessarily has a technical focus less than a thematic one. The specific scientific results are critically assessed in dedicated review papers by experts (Scholz, Marone, Rosakis, Lei etc) to which we generally refer in the text for further reading.*

*Following this idea the paper is organized as follows: Intro, Overview, Scaling, Materials, Monitoring, Applications, and Conclusions. Ch 6 (Applications) highlight common results from the different approaches. However, not at the depth requested by the reviewer.*

*The main effort of this paper was categorizing the approaches and their technical aspects and make them visible side-by-side. The section on scaling is actually the first of its kind, same as the discussion of analogue rock rheologies in the given context.*

*We took special care of these two main issues (timescale and results) during revision of the paper which led to changes at various locations in the text.*

*We address the timescales more explicitly now in the abstract, intro and conclusion:*

*Abstract:*
*"Earth deformation is a multiscale process ranging from seconds (earthquake rupture) to Millions of years (tectonic evolutions). Bridging short- and long-term deformation is addressed experimentally for more than a century."*

*Intro:*
*"Deformation of the earth involves timescales ranging from nanoseconds (processes at the rupture tip) to hundreds of Millions of years (Wilson cycle) and all spatial scales from atoms to the earth itself (e.g. Ben-Zion, 2008). Such a multiscale process pose major challenges to observation in nature as the instrumental and historical records are too short to capture a significant amount of the evolution. Simulation is a way to overcome such limitations. However, our knowledge of the physics of earthquakes and earth deformation in general is necessarily incomplete and does not allow to setup realistic numerical scenarios. Non-uniqueness of numerical solutions to typical problems of earth deformation at various time scales (e.g. inversion of rupture kinematics or mantle rheology from co- and postseismic observations, respectively) is another limitation of computer models. Experimental approaches have been traditionally used to address physical problems like earthquakes and seismic cycles and, more recently, bridging the gap between short-term instrumental/historical observations and long-term paleoseismological/geological observations."*

*Conclusions:*
*"We presented an overview of experimental approaches to model earthquakes,seismic cycles and seismotectonic deformation. The processes involved are multiscale posing the challenge to cross time scales from seconds (seismic deformation) to Millions of years (tectonic deformations) both in natural observation as in simulation and experiment. Since natural observations are intrinsically limited in resolution and period of observation, simulation by means of analogue and numerical modelling are key to understanding multiscale processes. An experimental approach to multiscale problems seems most natural because experiments happen in a time and space continuum in contrast to numerical models which need to be discretized.."*

*We added better descriptions of setups including two new Figures which also show results of spring slider and fault block models discussed in the text:*

*e.g. 6.1*

*"King (1991, 1994) who showed that large events tend to roughen the stress distribution while small events smooth. Moreover, he found that large events are dissimilar (i.e. not characteristic) and that rupture nucleation is not were peak slip accumulates. The frequency-size distributions found by King (1991, 1994) have been Gutenberg-Richter-like except for the system-sized events which recur approximately time-predictably."*

[Figure]

Figure 2: Examples of spring-slider model experimental setups and results: (a) Multiple spring-slider setup andexample of "rupture" by King (1991, 1994); (b) Schulze rings-shear tester used to characterize frictional properties of granular materials in this study (after Schulze, 1994); (c) simple shear setup and visualization of force-bridges in granular media by "see-trough" experiments of Daniels and Hayman (2008); (d) simple spring-slider setup used by Varamashvili et al. (2008); (e) Setup of "see through" experiments by Nasuno et al. (1998); ( f) Double-direct shear configuration used to shear rods in various configurations by Knuth and Marone (2007); (g) simple spring-slider setup after Heslot et al. (1994).

[Figure]

*Figure 3: Examples of fault block model experimental setups and results: (a) Brune's foam block model as used in Brune et al. (1993); Anooshehpoor and Brune (2006) and Day et al. (2008); (b) "Schallamach wave" pattern as seen in sliding rubber block experiments by Schallamach (1971); (c) Setup after Archuleta and Brune (1975); (d) Setup and resulting rupture pattern from Latour et al. (2013b); (e) Setup and photoelastic pattern visualizing stress in the vicinity of a crack after De Joussineau et al. (2001); ( f) Setup of Bouissou et al. (1998); (g) Setup and photoelastic fringe pattern associated with rupture in experiments of Rosakis et al. (1998) and Rosakis et al. (1999).*

**There is still vagueness and room for improvement, though. For example, Section 6.5 concludes the description of the Caniven et al. (2015) study with "The model results compare to numerical simulations of strike-slip fault earthquakes", leaving me wondering how they compared (well, I assume, but I can't be sure), to what specific aspects of these simulation the experiments can be compared,**

*In the case you are referring the comparability test are ongoing work and the sentence is a rather general one; we added some detail. We also added a more detailed description of the existing analogue–numerical comparisons:*

*"The model results are comparable to numerical simulations of strike-slip fault earthquakes in terms of seismic moment, slip gradients and postseismic response (e.g. Ben-Zion and Rice, 1997; Lapusta and Rice, 2003, Tullis et al., 2012a, b). "*

*"The analogue models of **Rosenau** et al. (2009, 2010) have been cross-validated using finite element models (Pipping et al., 2016;).The numerical model of Pipping et al. (2016) replicated the laboratory results by means of the general deformation pattern of the wedge through seismic cycles, the recurrence behaviour (recurrence times and periodicity) as well as principal source parameters (slip distribution). Using numerical simulation, the frictional properties of the analogue model material were validated and augmented if not measured physically. Moreover the numerical model provided a highly-resolved image of the rupture dynamics beyond what was observed in the analogue model (see e.g. animation of analogue vs. numerical subduction earthquake cycles in Rosenau et al., 2016). Vice versa, the laboratory example served as an object for testing the general performance of the numerical modelling scheme and for the verification of the effectiveness and reliability of the algorithm for frictional contact modelling introduced there."*

*"The analogue models of **Corbi** et al. (2013) have been tested against numerical models using the finite element technique (van Dinther et al., (2013a, b). An extensive benchmark has been carried out analysing a set of parameters that are characteristic of both the interseismic and coseismic stages (i.e., recurrence time, coseismic displacement, rupture duration, rupture velocity, slip rate, rupture width and hypocentre location). A robust fit has been obtained for the majority of the investigated parameters of the reference model (van Dinther et al., 2013a). In particular, the mean of individual source parameters of the numerical model fall systematically within the standard deviation of the same parameter of the analogue models. The largest discrepancies between the two modelling techniques are observed for rupture width and hypocentre location and are attributed to difference in boundary conditions (i.e., the aseismic part of the megathrust doesn't create stress build up in the analogue models) and sampling rate, respectively."*

**and, crucially, what new insight has been gathered from the experiment. I suppose it's not just a confirmation to earlier studies, but there is no way to tell, based solely on this manuscript. Similar vagueness pervades the paper (e.g. Page 6, line 25).**

*Up to now the experiment is in the validation phase. That's why we cannot provide completely new insights at this stage for this specific case.*

*No changes in manuscript.*

**Finally, I found the paper difficult to read due to imperfections of language. It needs to be thoroughly edited. This may be a stylistic choice, but the authors seem to avoid commas at all cost and that makes many sentences long and confusing. On the other hand, they love "i.e." and "e.g." whereas I find it best to avoid abbreviations. I find many twisted sentences that, although possibly not incorrect from a grammatical standpoint, are certainly not the clearest way to present the information. In writing, as in modeling, simplicity leads to clarity.**

*We agree. As a german-italic-french consortium we are likely not the virtuosi regarding English and happy to receive all the improvements you gave us in the technical comments. We hope the current version has a better style.*

*We revised the manuscript respecting all technical comments.*

**C2**
**SPECIFIC COMMENTS**
**1) The abstract promises to discuss "limits, challenges and links to numerical models".**
**I don't see that in the paper, expect for the occasional statement interspaced**
**with general presentations. It's certainly not a focus of the paper. The stated focus**
**on "scale models that are directly comparable to observational data on short and long**
**timescales" is lost in the more general and occasionally very basic sections on modeling**
**in general, rheology, and monitoring, which are imperfectly linked to observations**
**and models.**

*See our reply in the beginning, the current structure is focused on technical aspects. These aspects form the basis (and therefore are linked) to the observations reported in chapter 6.*

**2) The introduction introduces the "issue" that the time constraint of the earthquake**
**cycle is unknown. How then can it be argued that the analogue systems are properly**
**scaled? Doing this requires that we know and understand the relation between the**
**various timescales (e.g. nucleation stage, repeat time, postseismic duration). The**
**paper doesn't make the point that these relations are well understood, quite on the**
**contrary, whether in the lab or in nature.**

*Do you refer to the sentence:*
*"This raises the issue of a __time constant__ in the earthquake cycle that is far larger than the duration of most scientific observations."?*

*What we meant is that we have only snap shots of stages of the cycle in nature. What we miss are long enough time series to understand the relation across time scales. It is exactly this what analogue models can provide: long time series. Each stage of the cycle may have its own (in places it is known but not necessarily stationary over several seismic cycles) time constant. This is of course an integral part of scaling by means of kinematic similarity.*

**3) The distinction between "fault block" models and "scale" models seems arbitrary**
**to me: fault block models are scaled as well as the "scale" models, even though the**
**scaling may not always be rigorous.**

*It is certainly the rigor in applying scaling to the models which let us differentiate between the two categories. Observations from fault block models can be extrapolated to nature while scale models can be truly upscaled.*

*Fault block models are labscale but not "scaled" in the sense of Hubbert's scaling theory, i.e. showing geometric, kinematic and dynamic similarity. Fault block models do rarely show geometric similarity (instead of representing a certain structural setting at a given geometric scale they represent an arbitrarily oriented and scaled fault interface), typically there is no defined length scale and the deformation is highly exaggerated. Fault block models show no kinematic similarity (which would mean loading rate and rupture speed etc. scale down consistently) and consequently cannot, by definition, be considered dynamically similar. Observations from those models cannot be directly scaled up to nature but only extrapolated qualitatively.*

*We included new examples of category 1 and 2 setups in this Section in order make the difference clearer. We also changed the name of the third category to "seismotectonic scale models" in order to emphasize the multiscale nature and more quantitative applicability of these models.*

*We changed the text in Sect. 2:*
*"Here, we categorize analogue earthquake models into three groups with decreasing level of abstraction and applicability (Fig. 1, Table 1): (1) "Spring-slider models" in which elastic and frictional elements are physically discrete components of the setup (Sect. 2.1). __These models can only be applied conceptually to nature__. (2) "Fault block models" in which two elastic blocks, with similar or different elastic properties, are in frictional contacts (Sect. 2.2). __Observations from these models can be qualitatively extrapolated to nature__. (3) "Seismotectonic scale models" in which a distinct tectonic setting is realistically simulated at small scale and with boundary conditions mimicking as closely as possible the natural prototype (Sect. 2.3). __These models can be directly and often quantitatively upscaled to nature__."*

**The schematics of Figure 1 imply that fault block**
**models are in a strike-slip configuration with elastic blocks only whereas the scale**
**models would be in a thrust configuration with both elastic and viscous layers. I don't**
**see why one would be scaled and not the other.**

*The fault block models are usually not strike slip since this would imply a pressure gradient on the fault. Such a feature is not present in fault block models (which are loaded homogenously) but in scale models (indicated by the g-vector in Figure 1c)*

*See above, we added text and figure 2 and 3 showing examples of the two categories in order make the difference clearer.*

**Elastic moduli and friction properties are relevant in the all the models.**

*True, they are relevant but the elastic properties are not scaled in category 1 and 2 models according to scaling laws outlined in the paper (considering gravity and length scale).*

**Several of the scale models of Figure 2 do not have the kind of layering in Figure 1c and one is not in a thrust setting.**

*Figure 1c is a synthetic example including the sum of features of those models shown in fig 4 in a simplified matter in order to illustrate the main differences between cat 2 and 3 models (e.g. layering, gravity, dimensionless numbers etc.).*

**I agree that there is likely a difference in the rigor of scaling between the models built recently in the authors' labs and earlier efforts, but I don't see them as forming an entirely new category of models. If the classification is based on the complexity of the loading system (rigid blocks, elastic blocks, visco-elasto-plastic blocks) then the name of the proposed categories is misleading.**

*No, it is not the loading system which makes the difference. We think the applicability of scaling laws to these new models makes a distinction which warrants a new category. See reasoning above why we think only cat 3 are scale models.*

*We try to clarify the justification of the new category in the text and two new figure (see above).*

**4) Section 2 is essentially a list of works. It shows that people do different things but doesn't explain why these different approaches were adopted (why this material vs. that material), what problem or question new developments are trying to address, and what we learned as models grew in sophistication. It is as if an architect was describing a monument by listing stones that were used without telling us why there is such a variety of material and what the final building looks like.**

*A main goal of this paper was to give an overview of existing experiments not a critical and comprehensive assessment of results in all three categories. For this we put much effort in listing and categorization. This might not be very attractive for a reader but necessary and useful. A review of approaches and results would need a review paper on each of the categories.*

*Section 2 gives an overview of approaches. Section 3-5 are basic sections, the bricks if you like, (scaling, materials, monitoring) needed to illustrate the general approach. Section 6 then gives the applications and describes key observations gained from the individual approaches, the building if you like.*

*We reworked section 2 adding new examples and figures. Results are reported in section 6.*

**5) I'm amazed that there is summary of what controls frictional sliding more recent than Brace (1972) (page 5, line 9). As much as I like that paper and respect its historical value, it might be good to mention some of the developments from the last 45 years. . .**

*The chapter follows a chronological order starting indeed 45 years ago. The reference you are referring to was meant to summarize only those early experiments. As said in the next sentence we do not aim to review all rock mechanics experiments since.*

**Later on (page 5, line 13), you mention you want to focus on studies using analogue rock materials instead of rock samples. Why?**

*Because the manuscript deals with analogue models, not with rock mechanics (apart from the notion that stick-slip in rock mechanics experiments are viewed as an analogue (analogue mechanism, not model) for earthquakes).*

**And where are the ==results==?**
*Results from the different setups with respect to key thematic areas are reported in section 6.*

**You mention "a large body of work" twice in that paragraph, but only list them. ==What did they see==?**
*One "large body" refers to the rock mechanics experiments that we exclude from our review (in the same sentence).*
*The observations from the other "large body" can be found in section 6 (probably not every result but the general findings).*

**What is the ==key point of these papers== if the context of the present review?**

*They serve as examples for spring slider setup in a wider sense (where, as you say below, elasticity is stored in the loading device, but not in the sample)*

*We describe now the ring shear tester setup in more detail which is a member of this experiment family (see reply to comment 11, new text and figure below).*

**By the way,**
**I don't see why this discussion belongs in the section on spring-slider models, as the**
**==loading machine acts as a deformable loading block (fault-block model)==.**

*No, the key feature of spring-slider setup is the separation of spring and slider. This causes the rupture area to be constant and slip to be homogenously distributed causing slip events to be system-sized and characteristic. This is what happens in deformation rigs where the slider is the sample and the spring are the compliant parts of the loading machine. Block models allow heterogeneous slip to occur, which is the key difference.*

**Also at the end of section 2.2: what did all these studies using blocks of different materials ==see==?**

*Chapter 6 reports key observations, especially in the context of rupture dynamics block models are the main source of observations.*

**6) The issue that a rigid slider distribute stress uniformly (Page 5, line 23-25) is exactly**
**why people developed models with a ==network of springs== (King 1994, Heslot et al., 1994,**
**which were mentioned earlier in the section).**

*Heslot et al. (1994) is a simple spring-slider.*

*We added in order to clarify:*
*"Multiple spring-slider systems (e.g. Burridge-Knopoff, 1967; King, 1991, 1994) aimed at overcoming this limitation and succeeded in generating more complex slip and recurrence pattern (see Sect. 6.1). "*

*and*

*"Fault block models, __beside multiple spring-slider models__, have been developed to circumvent the strong assumption of uniform loading and release inherent __simple__ spring-slider models. (…) This setup also allows small (partial) and large (complete) scale failures to occur __in a less segmented fashion compared to multiple spring sliders__ bearing the potential to generate more realistic frequency size distributions.*

**==What did they see==? What did they learn about earthquakes from these models?**

*Chapter 6.1 reports key observations from spring slider experiments regarding statistics of EQs, especially in the context of eq statistics spring-slider models are the main source of observations. We added in Sect. 6.1:*

*"King (1991, 1994) who showed that large events tend to roughen the stress distribution while small events smooth. Moreover, he found that large events are dissimilar (i.e. not characteristic) and that rupture nucleation is not were peak slip accumulates. The frequency-size distributions found by King (1991, 1994) have been Gutenberg-Richter-like except for the system-sized events which recur approximately time-predictably."*

**7) Page 6, line 6-17:** why is a rigid plate appropriate to model the slab? Slabs are also

elastic. Even though the wedge above the slab is generally softer than the slab due to its elevated temperature, thermal conduction implies that there is no actual temperature jump across the subduction interface. Therefore, the footwall is as deformable as the head wall at least over some length scale.

*Analogue models are simplifications of nature. While it is true that the compliancy contrast across the plate interface is small (< 1 order of magnitude) resulting deformation pattern are asymmetric with much smaller displacements in the footwall. This is very similar to what is shown by the Rosenau models where the lower plate is actually loaded by flexible device. However, we did not discuss this detail here. It was certainly a rather strong boundary condition in the first experiments of Rosenau et al. and Corbi et al. which has been overcome only recently by the models involving a rubber belt or foam slab (Rosenau 3D, Dominguez).*

*We clarified it:*
*"A similar setup has been used by Rosenau et al. (2009) but with a granular, elastoplastic wedge on top of a less compliant conveyer plate or rubber belt (Fig. 4b)."*

**8) I don't understand the** analogy from adaptive time scaling in experiments and adaptive time stepping as a numerical method. The numerical strategy involves changing time step so that the solution becomes more stable or accurate. The solution itself is modified. The physical time scales, lengths scales, and other scales of the modelled system are not modified. The adaptive time scaling does the opposite: the solution is unchanged, but different scales are used when extrapolating different phenomena to natural conditions. This analogy baffles me.

*Possibly the analogy is wrong.*
*We deleted it at the two instances where it was mentioned.*

**9) The section on scaling is important and starts to address the issue of multiple time scales. Once again, unfortunately, it mostly states what is done without presenting many results. In addition, there is quite a bit of** confusion there as some quantities are either incompletely defined, or substitutions occurs without justification. I feel the authors could do a better job linking the non-dimensional numbers and the scaling relations.

*We have thoroughly reworked that section and tried to better define parameters and justify substitutions and relations.*

**To start with,** in Equation 2, please define the measure or component of stress that are you using. Is it an invariant, a shear stress, or a normal stress? The words in the equation say "pressure force", which is weird, as pressure is a force (over unit area) and isotropic, whereas I suspect that shear stress is used here. Neither $\rho*g$ or $\sigma/l$ have units of force. (same issue in equation 3).

*We now use cohesion as used in sandbox modelling. But it can be substituted by any stress or strength measure with the unit Pa. We clarified this in the text (see reply to Reviewer 1).*

**The** definition of Sm in Table 2 is different from Eq. 2,

*Fixed!*

and that table includes a Stokes number whose importance was not discussed in the text.

*Stokes number is referred to in 3.2 in a section on limits of scaling.*

**I am confused, in line 15-16, how Sm and Ra can dictate, among other things, length scaling, when the statement is "for a given length scale".**

*Fixed!*

*Reads now:*

*"To achieve similitude these numbers have to be the same in the model as in the prototype. For a given length scale (usually suitably chosen for handling the model in a lab), Sm and Ra dictate the stress scaling in the brittle and viscous regimes, respectively."*

**Note also that no brittle scaling has been defined. What if the model is not viscous? Can there not be a number equivalent to Ra but using, for example, inertial forces? I see Ra as a special case of Sm when the stress is controlled by viscous processes (\sigma=\eta*v/l). Why are they treated as different numbers?**

*Brittle scaling uses Sm with cohesion (or frictional strength) in the denominator. Inertial forces play a role only in the dynamic regime. Ca can be used to scale the brittle dynamic regime when substituting K for cohesion or frictional strength.*

**Why did you switch from v/l in Eq. 3 to d\epsilon/dt in equation 5?**

*v/l in equation (3) is a shear rate as in the original definition of Ra. deps/dt is strain rate which we consider more general and practical than shear rate. Since both have the same dimension, they are interchangeable.*

**Page 9, line 18: why restrict the scaling to "typically"? How other than with Re would you scale dynamical effects? The final paragraph of Section 3.2 belongs earlier, as that scaling is used in the analogy of moments at the top of page 10.**

*The message is that dynamic viscous effects cannot be scaled in analogue models. However, because of the viscoelastic nature of the earth the coseismic stage is mainly elastic, coseismic viscous deformation might be a neglectable. This is what this paragraph is dedicated to.*

*We delete "typically".*

**10) Section 4 on rheology is written as a level that doesn't help with the topic of the review. It is also not really "historical" as it doesn't describe how ideas and approaches have changed over time, just a portion of current understanding. It would be appropriate for a textbook, but defining all the possible rheologies seems a waste of space.**

*We see this section as a basic contribution to the topic summarizing a minimum set of rheologies. We don't think it's a waste of space.*

**In addition, these definitions are not rigorous. For example, Hooke was referring in 1676**

**to "The power of any springy body is in the same proportion with the extension.", which gives F=kx. It is not equivalent to Eq. 12, which is a differential form that allows for residual strength or strain. The diagram of Fig. 5 shows non-linearity and possibility residual strength, and while this is more realistic than F=kx, it is also not Hooke's law.**

*We changed text and the figure accordingly*
*.*
*"In linear elastic solids, as in springs, elastic strain    is generally linearly related to the applied stress    in the same direction (Fig. 5):*

$$E = \frac{\Delta\sigma 1}{\Delta\epsilon 1},$$
(12)

*which is a differential version of "Hookes's Law"."*

[Figure]

Also, **Byerlee's law** (Fig. 7a) is not a generic linear relation but **refers to specific sets of parameters (those next to the line fits, but not in the label** for \mu = \Delta\sigma_n / \Delta\tau) To save time and space, I will not give details of typos, unclear statements for this section (suffices it to say it needs as much editing work as sections 2 and 3) as I think it first needs to be reworked to focus on what is truly needed to understand the time scales of seismic phenomena in the lab.

*We changed the text and figure accordingly.*

*"While the Mohr-Coulomb criterion originally describes frictional faulting of an intact rock, the same graphical method can be applied to describe frictional sliding on pre-existing faults and retrieve the respective set of parameters ("Byerlee's Law"). Accordingly, fault rocks at very shallow crustal levels ($\sigma n <$ 100 MPa) rocks have virtually no cohesion and a relatively high friction coefficient of 0.85, while at deeper levels rock appears cohesive (C ~ 50 MPa) and has a friction coefficient of ca. 0.6 (Byerlee, 1978)."*

[Figure]

**The section on slip models, for example, is entirely irrelevant.**

*If you mean the section on **crack models and dislocations** (4.1.1.):*
*We think those models are important as they serve as benchmarks for the analogue models (and vice versa).*

*If you mean the section **recurrence models** (slip vs. time predictable) (6.1):*
*Time and slip predictable models are very basic models to be included here.*

*If you mean the section on **rupture models** (pulse vs. crack): same, basics, to be included we think.*

**I do need to point out that unlike what is written at the top of P.17, I find that Burger's body is considered to be more relevant then either the Maxwell or Kelvin-Voigt models in recent studies**

*We do not disagree:*
*P. 17 top sentence says that "Maxwell is more relevant than Kelvin Voigt", below we say "A more elaborate viscoelastic rheology is the Burgers model …"*
*We especially agree that Burgers model is nowadays more often used to model postseismic deformation. But how do we know whether a complex model is better than a simple one? The problem of inverting postseismic deformation is non-unique and there are examples were simple models fit equally well. Applying "Occam's razor" we should favor simpler models (with less parameters) as long as they are qualitatively good enough.*

*We added in conclusion:*
*"New materials remain to be explored. Especially non-linear rheologies both in brittle and viscoelastic regimes will contribute to more realistic analogue models in future. A rigorous material characterization is prerequisite. For example, implementation of Burgers rheology in analogue models including postseismic mantle relaxation appears as a necessary step in near future."*

**and that the presence of multiple time scales of postseismic relaxation was seen in many studies long before Wang et al. (2012), e.g., Savage and Svarc (JGR, 1997), Nishimura et al. (Tectonophysics, 2000), Kenner et al. (JGR 2000) and others.**

*Certainly, but Wang is a useful Review though. Since we do not want to discuss transient rheologies here, and only make the point that such concepts exist we refer to his review. We would like to minimize references here.*

*We now write "(Wang et al. 2012, and references therein)"*

**11) The Schultze ring shear apparatus plays a prominent role in the collection of the mechanical data presented. Yet its description is minimal (page 15, line 21-22). Please describe in more detail what that apparatus is and how it works. Maybe include a schematic of this apparatus?**

*Good point!*

*We describe the setup now in chapter 2.1 and show it in Figure 2 as an example of spring-slider setup.*

*"Several studies which focus in frictional behaviour of granular rock analogue materials (e.g. sand, glassbeads) at low loads (kPa) used a Schulze ring-shear tester (Schulze, 1994, Ritter et., 2016; Klinkmüller et al., 2016; Panien et al., 2006; Lohrmann et al. 2003) which serves here as an example of spring-slider device used to generate analogue earthquakes (Fig. A1). The ring-shear tester consists of a 4 cm high annular shear-cell made of stainless steel holding approximately 0.1 and 1 liter of the sample material. A ring-shaped-lid is placed onto the filled cell. The lid is subjected to a normal force in order to control normal load on the sample. While the cell is rotated, the lid is prevented from rotation by two tie rods connected to a crossbeam. The force necessary to shear the material is measured continuously. To ensure shearing inside the material and prevent slip between the lid and the granular material, the lid has 20 vanes protruding 4 mm into the material.The loading system is compliant enough (~1.3 kN/mm) to generate sticks-slip in a variety of materials at loads below 20 kPa. Results of this setup are presented on several occasions in this paper."*

[Figure]

**12) Section 5 (monitoring techniques) reads like a long list of approaches. As before, I'd like the authors to maybe compare more explicitly what can be learned from using these techniques. Looking back at all these works, what would you recommend using to answer different questions?**

*The idea was to give a short overview of what exists and what observables, at which resolution and coverage, can be retrieved. The categorization according to the coverage (local, regional, global) and the drawing of parallels to seismology and geodesy was meant to get an easy overview about the capabilities of the different techniques. Apart from the text there is Table listing these approaches making it easy to compare. Moreover, most techniques currently in use have dedicated review papers (e.g. Lei and Ma, Adam, Rosakis…) to which we refer. We think this is a convenient way to guide the user finding a suitable monitoring technique.*

**13) I found section 6 to be much better written than the rest of the paper and more useful, in that it details not just what was done, but also what was learned from these experiments. It finally explains something about seismic phenomena and reveals the usefulness of (a few of) the experiments mentioned earlier. There is room from improvement, though. For example, b-values are mentioned page 21 line 30, before the concept was introduced in page 22 line 7.**

*On its first occurrence the term "b-value" is listed along with other parameters which are defined in the following two sentences. We consider this close enough.*

**In page 23, stick-slip is discussed line 18 but defined line 25. My other comments on this section are minor.**

*We now define stick- slip in the introduction:*
*"With the rise of the plate tectonic theory in the 1960s accompanied by thriving of seismology and experimental rock mechanics, stick-slip instabilities (the cyclic slow accumulation and sudden release of stress along frictional interfaces) along pre-existing discontinuities, i.e. tectonic faults, has become the most prominent earthquake mechanism"*

**14) At the end of section 6.4, we are presented again with a technical aspect (how Brune and Anooshepoor excited their models) without being told what they learned in that study.**

*Those models are somehow apart from the modelling approach represented by the rest of the paper. That is why we only give this short note to this specific approach.*

**15) Check the references. Several are missing elements. A few are using all-caps for the journal. Sometimes, the first name appears first (A. Alshibi)**

*Fixed!*

**16) What exactly are the "Nature example" shown in Figure 14? Neither the caption nor the text give us this information.**

*Added in Figure and reference list:*

*"Schmalzle, G., T. Dixon, R. Malservisi, and R. Govers (2006), Strain accumulation across the Carrizo segment of the San Andreas Fault, California: Impact of laterally varying crustal properties, J. Geophys. Res., 111, B05403, doi:10.1029/2005JB003843.*

*Cakir, Z., A. M. Akoglu, S. Belabbes, S. Ergintav, and M. Meghraoui (2005), Creeping along the Ismetpasa section of the North Anatolian fault (Western Turkey): Rate and extent from InSAR, Earth Planet. Sci. Lett.,238,225–234, doi:10.1016/j.epsl.2005.06.044.*

*Peltzer, G., F. Crampé, and G. King (1999), Evidence of nonlinear elasticity of the crust from Mw7.6 Manyi (Tibet) earthquake, Science, 286, 272–276, doi:10.1126/science.286.5438.272.*

*Fialko, Y. (2004), Evidence of fluid-filled upper crust from observations of postseismic deformation due to the 1992 Mw7.3 Landers earthquake, J. Geophys. Res., 109, B08401, doi:10.1029/2004JB002985"*

**All other technical comments:**
*We agree.*
*We changed the manuscript accordingly.*